# REASONING LANGUAGE MODEL INFERENCE SERVING UNVEILED: AN EMPIRICAL STUDY

**Qi Li**[1,2*], **Junpan Wu**[4*], **Xiang Liu**[1*], **Yuxin Wang**[3], **Zeyu Li**[1],
**Yuhan Chen**[1], **Shaohuai Shi**[5], **Zhenheng Tang**[6†], **Xiaowen Chu**[1†]

[1] The Hong Kong University of Science and Technology (Guangzhou)
[2] Shenzhen International Graduate School, Tsinghua University    [3] HKBU
[4] University of Wisconsin-Madison    [5] Harbin Institute of Technology, Shenzhen
[6] The Hong Kong University of Science and Technology
∗ Equal contribution † Corresponding authors
`lqinfdim@163.com`

## ABSTRACT

The reasoning large language model (RLLM) has been proven competitive in solving complex reasoning tasks such as mathematics, coding, compared to LLM. However, the serving performance and behavior of RLLM remains *unexplored*, which may undermine the deployment and utilization of RLLM in real-world scenario. To close this gap, in this paper, we conduct a comprehensive study of RLLM service. We first perform a pilot study on comparing the serving performance between RLLM and traditional LLM and reveal that there are several distinct differences regarding serving behavior: (1) *significant memory usage and fluctuations*; (2) *straggler requests*; (3) *adaptive running time*; (4) *domain preference*. Then we further investigate whether existing inference optimization techniques are valid for RLLM. Our main takeaways are that model weight quantization, KV cache quantization, and speculative decoding can improve service system efficiency with small compromise to RLLM accuracy, while prefix caching may degrade inference serving performance for small RLLM in some scenarios. Lastly, we conduct evaluation under real world workload modeled by the Gamma distribution to verify our findings. Empirical results for real-world workload evaluation across different datasets are *aligned* with our main findings regarding RLLM serving. We hope our work can provide the research community and industry with insights to advance RLLM inference serving. The reproduction details of this work can be found in §F.

## 1 INTRODUCTION

Large language models (LLM) such as GPT4 (Achiam et al., 2023), Claude4 (Anthropic, 2024; 2025), Gemini (Team et al., 2023), Llama (Grattafiori et al., 2024) have emerged as powerful knowledge bases via pre-training. These models, trained on vast Internet-crawled corpora such as C4 (Raffel et al., 2020), PILE (Gao et al., 2020), and guided by scaling law (Kaplan et al., 2020; Rae et al., 2021), have accumulated large-scale knowledge and exhibited remarkable performance on various knowledge-intensive tasks. Despite these advancements, LLMs are criticized for their unsatisfactory capabilities on complex reasoning tasks, e.g., challenging mathematics and programming tasks.

Recently, reasoning large language models (RLLM) like OpenAI o1 (Jaech et al., 2024), DeepSeek R1 (Guo et al., 2025), Qwen-3 (team, 2025) have sparked a growing body of research into *test time scaling* (Snell et al., 2025; Muennighoff et al., 2025) via *long chain-of-thought reasoning* (Wei et al., 2022), significantly improving their mathematical reasoning, coding tasks and knowledge reasoning capabilities, e.g., even a 1.5B open source RLLM can surpass giant cutting-edge LLMs like GPT-4o on math tasks (Guo et al., 2025). Such achievements make it possible to deploy a small to medium RLLM as a powerful assistant to light the burden of workload for the staff of small entities or even for person, democratizing the use of cutting-edge RLLMs. Hence, it is desirable for small entity with *limited GPU resources* to *efficiently* deploy RLLM with an inference engine privately for internal use.

Nevertheless, current LLM serving engines, e.g., vLLM (Kwon et al., 2023), LMDeploy (Contributors, 2023), TensorRT-LLM (NVIDIA, 2023), are initially designed for traditional LLM, other than for RLLM. Though optimization techniques for LLM serving (§2) have been extensively studied, it remains largely *unexplored* whether RLLM exhibits distinct serving characteristics from LLM. If so, directly applying existing LLM serving techniques to RLLM may leave sub-optimal serving performance. Thus, it is natural to ask the following critical research question:

*Is there any distinct difference in serving behaviors between LLM and RLLM?*

To answer the above question, we perform systematic studies of RLLM serving. We first establish the **ASU assessment framework** (§3.2) for assessing RLLM serving. To justify whether there exists a distinct difference in serving behavior between RLLM and LLM, we design a benchmark suite named *ASU-Perf* and conduct a pilot investigation with it on different-scale LLM and RLLM (§4). We found that when requests arrive in batches, the serving behavior of RLLMs *differs significantly* from LLMs, and the *main findings* can be primarily summarized in the following aspects: (1) RLLM exhibits significant KV Cache fluctuations and usage; (2) Long tail distribution of requests running time caused by difficult requests; (3) RLLM solves different difficulty level problems with adaptive running time; (4) RLLM excels LLM on math reasoning while on-par on knowledge-intensive tasks.

To understand RLLM serving further, we first conduct extensive evaluations with various optimization techniques across diverse benchmarks (§5). We find that the model quantization and speculative decoding integrated in the serving engine can improve serving efficiency and performance with only a small compromise on the accuracy of RLLM. KV cache quantization and Prefix caching generally improve throughput and efficiency across almost all evaluated models, with degradation observed only in a few specific cases. For KV cache quantization, we hypothesize the observed degradation is mainly attributed to compatibility issues between specific model architectures and the serving framework (e.g., vLLM). On the other hand, prefix cache introduces performance drops in certain scenarios, especially single-turn dialogue, where the lack of prefix reuse diminishes its advantage and can even result in unnecessary computational overhead, like hash computation.

Lastly, we conduct evaluation (§6) with the same settings under real-world workloads modeled by Gamma distribution to verify our findings with different-scale language models across different domains. Empirical results of real-world workload evaluation indicate that the serving behaviors of RLLM are distinct from the LLM and are *aligned* with our main findings regarding RLLM serving.

We hope our work can both provide the research community and industry with insightful perspectives to help advance studies in efficient RLLM serving. To the best of our knowledge, we are the *first* to dissect the RLLM serving performance. The main contributions of this paper are the following.

- Conceptually, we propose *ASU*, a framework to assess RLLM serving, which considers accuracy of response, RLLM service-provider side metric, and user side performance metrics together (§3).
- Technically, we introduce *ASU-Perf*, a benchmarking suite for evaluating RLLM serving (§3).
- Empirically, we reveal key differences of serving behaviors between RLLM and LLM: *Significant Memory Fluctuations and Usage*, *Straggler Requests*, and *Adaptive Running Time* (§4).
- We conduct extensive experiments on some RLLM serving optimization techniques (§5).
- We empirically validate our findings in real-world workload and verify their generalization (§6).
- We discuss the potential optimization for RLLM specific inference regarding our findings (§8).

## 2 PRELIMINARIES

In this section, we provide preliminaries of RLLM, LLM serving and its metric. For comprehensive introduction of LLM serving optimization and recent advancement, please refer to Appendix E.

**RLLM and LLM.** LLMs have demonstrated remarkable capabilities across various natural language processing tasks. However, standard LLMs often encounter difficulties when faced with complex problems that require multi-step reasoning, planning, and deeper cognitive processes, sometimes referred as "System-2 tasks" (Li et al., 2025d). To address these limitations, RLLMs have emerged, specifically engineered to enhance these deliberative reasoning abilities. A key technique employed by RLLMs is the "long Chain of Thought" (long CoT) prompting strategy (Shao et al., 2024). This

approach encourages the model to generate extended, explicit step-by-step reasoning pathways, breaking down complex problems into more manageable parts. Unlike standard LLMs that might provide more direct or less detailed answers, RLLMs utilizing long CoT can better navigate the intricacies of tasks, leading to more accurate and justifiable solutions by methodically thinking through the problem. This distinction allows RLLMs to tackle challenges in domains like advanced mathematics, intricate logical puzzles, and long-horizon planning more effectively than their conventional counterparts.

**LLM Serving.** To exploit LLM in real-world scenarios, current practice generally delegates the inference procedure as an individual serving service. The design goal of such serving systems is to accommodate inference output to client users with *low latency* and *high throughput* and full use of GPU memories. Unlike the encoder-based language model (Vaswani et al., 2017) like BERT (Devlin et al., 2019), LLM first processes input prompts with intensive computation at the *prefill stage* and then generates output tokens one by one within each iteration at *decoding stage*, which limited by the memory capacity of the hardware. Traditional serving systems process prompts batch by batch, resulting in ineffective memory utilization. Orca (Yu et al., 2022) introduces *continuous batching* schedule at granularity of each token generation iteration to improve throughput of serving system. To handle as much input requests, the memory space for serving system should be efficient yet elaborated managed. Since decoding phase needs to re-use KV values of their prompt tokens which are stored in GPU, vLLM (Kwon et al., 2023), a high performance serving engine, introduces PagedAttention with paged memory fragmentation and sharing mechanism , which alleviates memory fragmentation and enables allocation in demand. Considering the prefill is compute-intensive task, while the decode is memory-intensive task, for further improvement, DistServe (Zhong et al., 2024) disaggregates the prefill and decode phase by assign computation of these two stages to different GPUs, which co-optimizes the resource allocation and parallelism tailored for each phase.

**Serving Performance Metrics.** To measure the performance of serving system, there are multiple metrics can be chosen: (1) Time to first token (TTFT) is the time it takes to process the prompt until generate the first token. It measures how long a user must wait before seeing the model's output; (2) End-to-end request latency (E2E latency) indicates the time it takes from submitting the first token of a request to receiving the last token of response, including the time for queueing and batching and network latencies in real-world scenario; (3) Time between tokens (TBT, a.k.a Intertoken latency, ITL) is the average time between the generation of consecutive tokens in a sequence; (4) Tokens per second (TPS) of system represents the mean of total output tokens number per second , accounting for all the requests happening simultaneously; (5) Requests per second (RPS) is the average number of requests that can be successfully completed by the system in a 1-second period. For More details of LLM benchmarking metrics, please refer to §E.2 and related resource (Vinh et al., 2025; inc, 2024).

## 3 EXPERIMENTAL SETTINGS

In this section, we present experimental setups (§3.1) and the ASU assessment framework (§3.2).

### 3.1 SETUPS

Here, we list necessary experimental setups. For implementation details, please refer to Appendix G.

**Language Models.** We employ 4 different scale models to assess their serving performance and serving behavior. General LLM : Qwen-2.5-Math 7B (Yang et al., 2024b), Qwen-2.5-14B , Qwen-2.5-32B (Yang et al., 2024a), and meta-llama/Llama-3.3-70B-Instruct (Grattafiori et al., 2024) and their long-cot tuned counterparts RLLM: DeepSeek-R1-Distill-Qwen-7B, DeepSeek-R1-Distill-Qwen-14B, DeepSeek-R1-Distill-Qwen-32B , and DeepSeek-R1-Distill-Llama-70B for fair comparison.

**Evaluating Datasets.** We adopt four different widely used datasets to evaluate the performance of RLLM. Since RLLMs are particularly trained for system-2 reasoning tasks (Wei et al., 2022), we mainly perform benchmarking with mathematical problems. We adopt three different difficulty level math reasoning datasets: GSM8K (Cobbe et al., 2021) as easy level, MATH-500 (Hendrycks et al., 2021; Lightman et al., 2023) as medium level, AIME-2024 (Committees, 2024) as the hardest level. To further distinguish are there any differences of serving performance and behaviors for RLLM in reasoning math problem or knowledge-based problem , we also used GPQA (Rein et al., 2024) dataset for knowledge reasoning. More details of these datasets are introduced in §G.1.

**LLM Inference Engine.** We employ 2 most adopted open source LLM inference engines, vLLM and SGLang (Zheng et al., 2024) in evaluation. We use OpenAI compatible API of these engines.

**Evaluation Suite.** We employ *ASU-Perf*, an benchmark suite proposed *by us* for evaluating LLM and RLLM serving performance with different inference engine. We leverage it in all of evaluation.

## 3.2 THE ASU ASSESSMENT FRAMEWORK

The adoption of RLLM hinges on whether their are capable of generating value that outweighs their inference costs (Erol et al., 2025). Assessing this tradeoff requires metrics that account for both performance and serving costs for both service provider and users. For RLLM service providers and users, the performance metrics they care about differ: providers seek to maximize system throughput, while users expect rapid model responses. In addition, it is essential to ensure response accuracy while optimizing RLLM serving system performance as much as possible. Thus, we propose *ASU* (*Accuracy, Service-end, User-end*), a trinity framework for assessing RLLM serving performance by together considering response accuracy, RLLM service provider end and user end. For accuracy metric, we employ evaluation *own metric* for each dataset. For service provider side metrics, we use throughput metric TPS (token per second) . For user-side metrics, we use TTFVT (time to first visible token) , a variant of TTFT , since we assume reasoning tokens of RLLM are invisible to users like commercial RLLM like OpenAI o1, and E2E requests running time as metrics.

In the next section, we will dive into the characteristic of RLLM serving via detailed experiments.

## 4 PILOT INVESTIGATIONS: SERVING LLM V.S. RLLM

In this section, we perform an comprehensive investigation to RLLM and LLM inference serving.

**Experiments.** We involve eight prevailing models in evaluations. For fair comparison, RLLM model we employed is the tuned counterpart of evaluated LLM, e.g., Qwen-2.5-Math-7B and its tuned RLLM counterpart DeepSeek-R1-Distill-Qwen-7B. We conduct evaluation with 7B, 14B, 32B, 70B language models on different inference engines. For comprehensively assessment, we perform evaluation with different token budget and batch size. We use all the datasets described in §3.1.

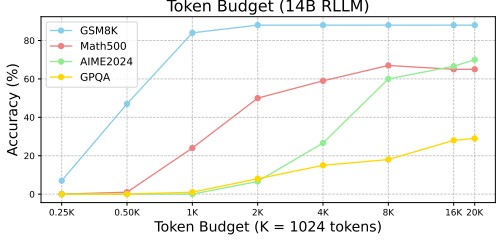 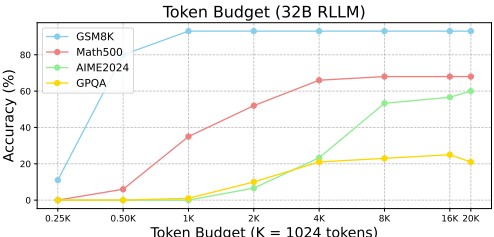

Figure 1: Results of token budget variation across different datasets for 14B and 32B RLLM .

**Main Results.** 1) *Results with Different Token Budget*: Unlike traditional LLMs, RLLMs engage in deliberate reasoning by generating lengthy chains of thought prior to answer, which significantly increases token consumption. However, as existing LLM services are priced based on token usage, this results in substantially higher costs. To justify the impact of token budget for RLLM serving, we conduct evaluation with varying token budget from $0.5K$ to $20K$ across benchmarks. The results are presented in Figure 1. We found that, for the majority of datasets, a token budget of 4096 to 8192 can achieve sufficiently good performance. It is worth noting that, as the token budget increases, the performance of RLLMs on the GPQA and AIME24 datasets declined, which may indicate the *overthinking* problem (Qu et al., 2025) of RLLM. Please refer to §I.1 for full experimental results.

2) *Results with Different Batch Size.* We also explore the impact of different batch sizes on RLLM serving performance with the same experimental setting. We find that increasing the batch size does not affect model accuracy on various datasets. Nevertheless, it reduces the time required for RLLMs to process the same number of requests, and improves throughput metric TPS, but at the cost of increased average TTFVT. Please refer to §I.2 for full results with different batch size.

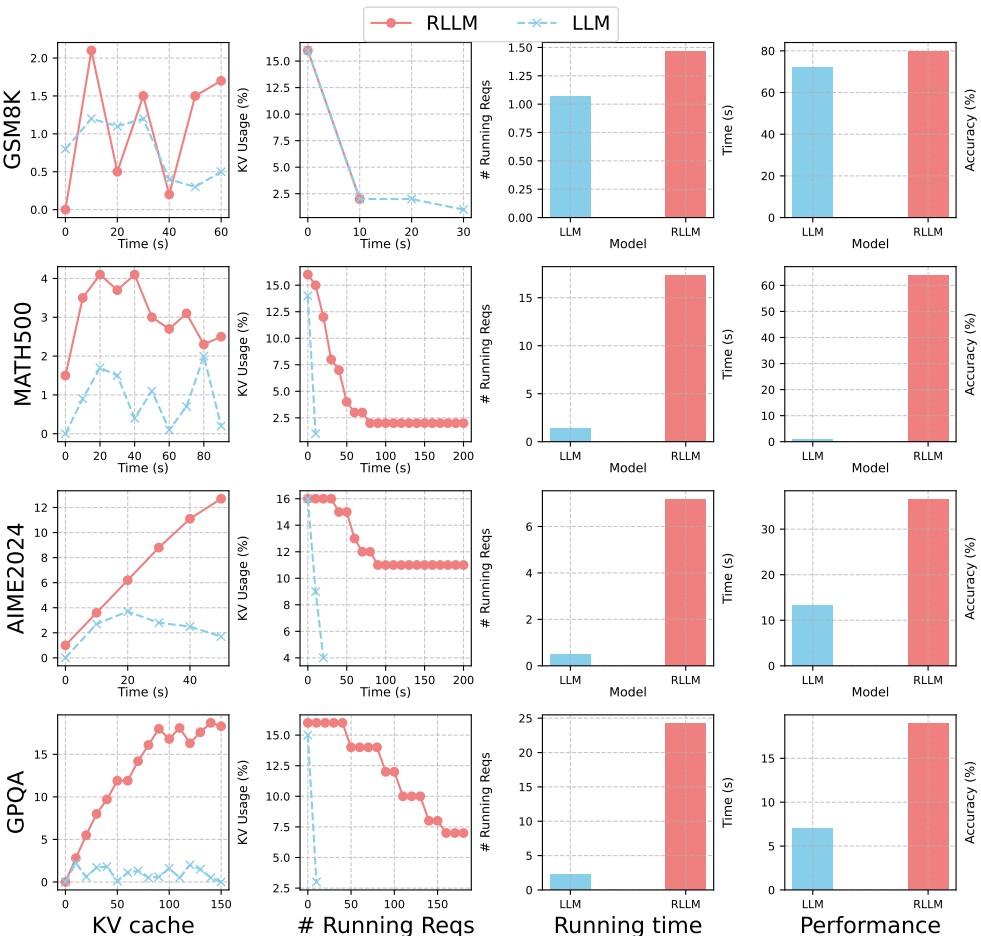

Figure 2: The serving performance and behavior comparison of a batch requests between 7B RLLM and LLM. We can read from this figure that (1) RLLM exhibits significant KV Cache fluctuations than LLM; (2) long tail distribution of requests running time caused by straggler requests; (3) adaptive running time of RLLM; (4) domain preference on math. Please refer to §I.3 for more results.

**Serving Performance and Behaviors.** To investigate RLLM serving behaviors, we analyzed the running logs of the inference serving engine and conducted a visualization of the running traces, as shown in Figure 2. As illustrated, RLLMs achieve much higher accuracy on math datasets than same scale LLM, but a on-par performance on knowledge reasoning such as GPQA. The full results are presented in §I.3. To dissect the difference of serving behavior, we present the running trace in §I.4.

**Main Findings for RLLM Serving Characteristics.** Given the above results in pilot studies, we have the following findings in comparison of RLLM and LLM serving behaviors :

- *Significant Memory Usage and Fluctuations:* We observed significant fluctuations in memory (reserved for KV cache) utilization of inference engine when serving RLLM . In extreme cases, the usage varied dramatically between 3% and 70%, whereas LLMs typically maintain KV cache usage below 3%. We attribute these fluctuations to the excessive length of the reasoning chains generated by RLLMs, which result in high memory consumption. During inference, the engine must retain KV caches for the reasoning chains until the requests are completed, after which they are discarded.

- *Straggler Requests:* When requests arrive at the inference engine in batches, or an RLLM receives multiple requests simultaneously, significant disparities in request difficulty can lead to some requests taking much longer time to complete than others. We denote these slow requests as *straggler requests*. These straggler requests ends either reaching the token budget or finishing the

reasoning process. During this time, only a small number of requests remain running in inference engine, resulting in a noticeable drop in system throughput and hardware utilization. In contrast, LLMs exhibit much smaller variations in execution time for requests within the same batch.

- *Adaptive Running Time of RLLM:* We found that, given the same number of samples with the same batch size (also the same token budget), the runtime of RLLMs varies significantly across different datasets and is strongly correlated with the difficulty of the tasks. In contrast, traditional LLMs exhibit much smaller runtime differences across datasets, with little sensitivity to task difficulty. When the number of samples varies, the runtime of LLMs on each dataset scales approximately linearly with the dataset size, even when there are substantial differences in task difficulty.

- *Domain Preference:* RLLMs and LLMs exhibit significant performance differences on the mathematical reasoning, while on-par on knowledge tasks, which align with existing works.

**Discussion and Analysis for Findings.** Based on the working mechanisms of inference engines we employed in benchmarking, we discuss the reason of why the above revealed phenomena occur.

1) *Straggler requests*: In some mathematical datasets, e.g., Math-500, the difficulty of individual problems varies. We assume that requests arrive in batches, and new requests are sent only after all requests in a batch are completed. As easier problems are answered shortly, the few remaining difficult requests continue running in the engine, leading to the entire batch's runtime being extended by these straggler requests. This situation results in reduced system throughput.

2) *Memory fluctuation and usage*: This issue is caused by the KV Cache management strategy of existing inference engines. Since RLLMs generate more tokens than traditional LLMs, the KV Cache utilization for RLLM is much higher under the same scale model with same precision in the same inference engine. This leads to a rapid increase in KV Cache usage, and since current inference engines discard the KV Cache once completing requests, it results in a sharp drop in cache usage.

3) *Adaptive running time*: RLLM generates varying reasoning chain lengths depending on problem difficulty—more difficulty lead to longer chains and running time. Hence, RLLM's runtime is typically correlated with problem difficulty, while LLMs generally may not be affected by difficulty.

Our findings indicate notable differences in serving RLLMs and LLMs. To enable more effective deployment of RLLMs, we explore some optimization techniques for inference in the next section.

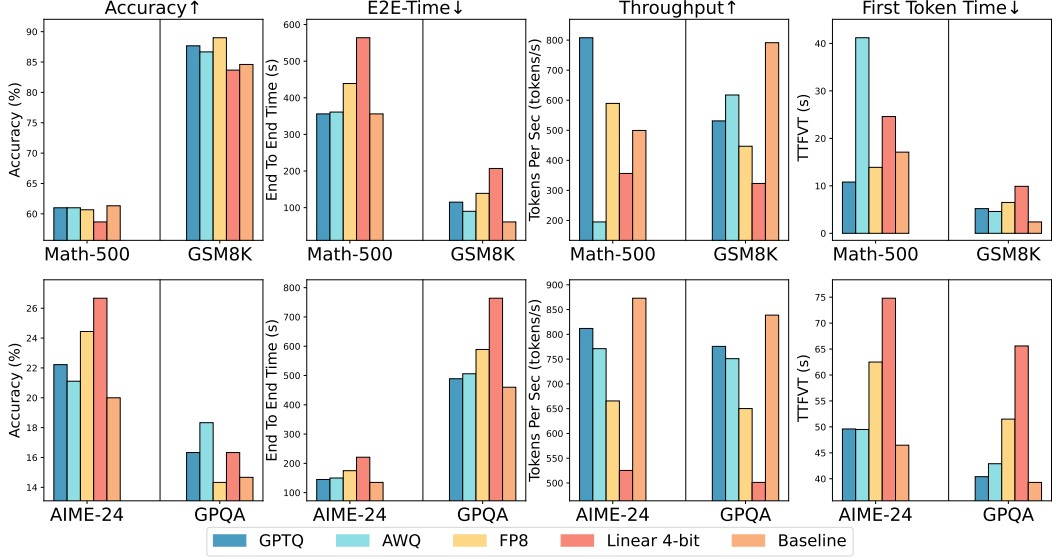

Figure 3: Empirical results of current LLM quantization methods on 7B RLLM. current methods maintain or improve all serving-related metrics with less memory footprint while keep accuracy.

## 5 OBSERVATIONS ON RLLM SERVING OPTIMIZATION

In this section, we take a closer look at the widely adopted techniques for optimizing LLM serving performance. We would like to explore whether these optimizations are still valid for improving

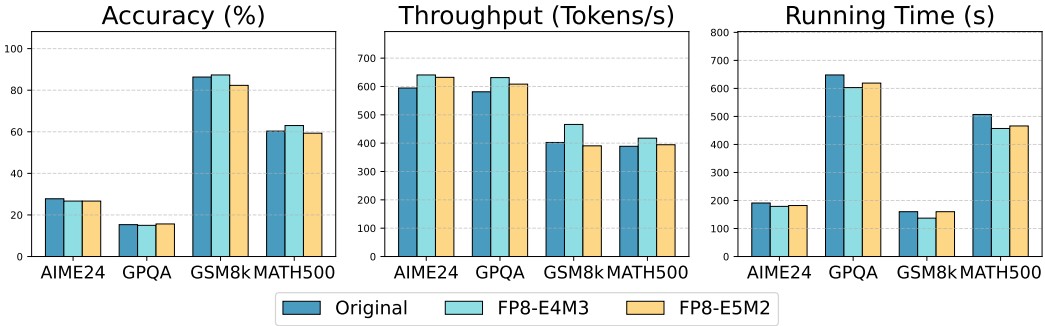

Figure 4: Empirical results for KV cache quantization on 14B model across different datasets.

RLLM serving performance. The prerequisite for assessing these optimization techniques is that they must *preserve* the RLLM's accuracy as much as possible. It holds throughout this section. All of Models we evaluated in section are listed in Table 1. Full experimental results are presented in §J.

## 5.1    IS MODEL WEIGHT QUANTIZATION METHODS EFFECTIVE IN BOOSTING RLLM SERVING?

Model weight quantization (MWQ) refers to the techniques that reduce number of bits for model parameters with the minimal loss in performance. Current LLM quantization methods are mainly fallen into the post-training quantization approaches. For more comprehensive introduction of LLM quantization, please refer to (Zhu et al., 2024) and (Gong et al., 2024).To investigate the impact of model weight quantization, we employ 4 most adopted (also supported by current open source LLM serving engine) quantization methods for LLM: GPTQ (Frantar et al., 2023) (Int4), AWQ (Lin et al., 2024) (4-bit), FP8 (Kuzmin et al., 2022), and Linear 4-bit (Dettmers et al., 2023)(L4) with BitsAndBytes (bitsandbytes foundation, 2022). We conduct experiments on 7B, 14B RLLM.

**Main results.** The evaluation results of quantized 7B RLLM using different quantization methods are presented in Figure 3. GPTQ-IN4 and FP8 quantization preserve the original model performance on most datasets, incurring only a minor degradation of approximately 3% or even perform better, while maintaining or improving all serving-related metrics with less memory footprint. However, GPTQ exhibits a substantial performance drop of around 15–25% on more challenging mathematical tasks such as AIME24. In contrast, AWQ and L4 maintain performance across all datasets but result in a marked reduction in inference efficiency, nearly doubling E2E time and halving throughput. These highlight the limitations of these approaches. The comprehensive results are presented in §J.1.

**Observation 5.1.** MWQ methods exert differing impacts on various metrics of RLLM inference.

## 5.2    COULD KV CACHE QUANTIZATION LEAD TO BETTER RLLM SERVING PERFORMANCE?

As illustrated in (Kwon et al., 2023), to serve traditional LLM, at least 30% of GPU memory is preserved to store KV cache in the generation process. For RLLM, the demand for KV cache storage would be paramount since its much longer output length ( including chain of thought reasoning ), which makes it evitable for efficient management of memory. KV cache quantization emerges as an appealing approach to this end. We employ two KV cache quantization methods natively supported by vLLM: FP8-E5M2, and FP8-E4M3 (vllm project, 2024) for inference serving evaluation.

**Main results.** The results of KV Cache quantization for 14B RLLM are presented in Figure 4. We found that using KV cache quantization effectively accelerates the operation of RLLMs while maintaining performance comparable to the original. Surprisingly, while the 14B or 32B RLLM maintained performance with minimal degradation, the 7B RLLM experienced almost complete performance deterioration. Detailed analysis are shown in §J.2. Furthermore, we observed that KV cache quantization can also improve other metrics such as TTFVT and TPS.

**Observation 5.2.** KV Cache quantization can improve running efficiency for RLLM inference.

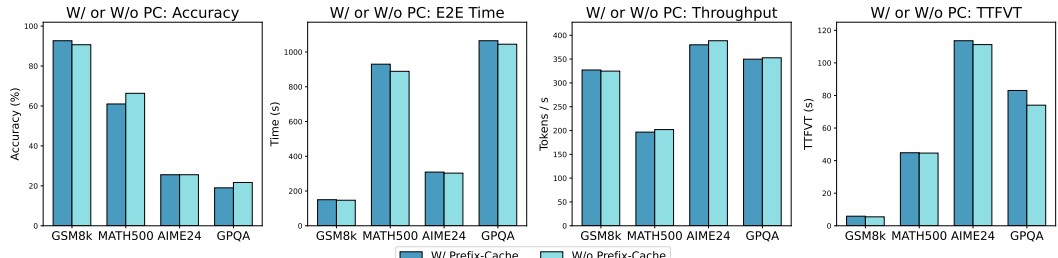

Figure 5: Empirical results of comparison for enabling or disabling prefix caching on 32B RLLM.

## 5.3 IS PREFIX CACHING USEFUL FOR CONTRIBUTING EFFICIENT RLLM SERVING?

Prefix Cache (PC) is a cache optimization policy that reuses computed KV values for prefill stage. By using this technique, new prompts that share the same prefixes (exactly, same prefix tokens) with previous prompts processed by serving systems can reuse these KV cache. This technique is very useful such as long document query or multi-round conversation where requires multiple recomputation of same text. Empirical studies show that the prefix cache can provide a huge performance benefit in such scenarios. Based on hash of input text, we can build an one-to-one mapping to manage mapping relation between logic block and physical block of KV Cache. To evaluate the utility of prefix cache in RLLM serving, we compare the performance of 8 different scale RLLMs across all datasets with or without prefix caching enabled in inference serving engine.

**Main results.** The results of PC evaluation on different datasets are shown in Figure 5. We find that for sufficiently large RLLMs (14B and above), PC significantly improves runtime speed and serving metrics without compromising performance. However, for part of smaller models (less than 8B, both RLLM and LLM), PC may lead to increased latency. This indicates the issue is general to small models, not exclusive to RLLMs. Detailed results and analysis are presented in §J.3.

**Observation 5.3.** PC can accelerate larger RLLMs inference without performance degradation.

## 5.4 DOES SPECULATIVE DECODING HELP TO IMPROVE RLLM SERVING PERFORMANCE?

Speculative decoding (SD) refers to a bunch of approaches that improves inter-token latency in memory-bound LLM inference. The initial speculating sampling usually employs a faster homogeneous LLM as draft model to generate a multiple tokens draft, and then the larger LLM can decide to accept or reject this draft by scoring. The results in (Chen et al., 2023) show that the overhead of draft model is much smaller than larger LLM forwarding, which makes it feasible to be utilized in real world scenario. Recently, many works in speculative decoding (Xia et al., 2024) like n-gram matching (vLLM Team, 2024), MLP speculators (Wertheimer et al., 2024), and Eagle algorithm (Li et al., 2024c; 2025c) are proposed. Despite these advancement, current support and compatibility of speculating decoding for RLLM in serving framework is poor. Given this situation, we only assess n-gram matching algorithm for 7B, 14B and 32B RLLM serving with vLLM iframework. The other experimental settings is keeping the same as the settings in Section 5.1 for fair comparison.

**Main results.** The main results for speculative decoding evaluation of 7B RLLM are listed in Figure 6. See §J.4 for full results. We find that speculative decoding improves the inference serving running time of RLLM across all scales, without degrading model performance on benchmarks. However, speculative decoding significantly reduces throughput and degrades the TTFVT metric.

**Observation 5.4.** SD improves the running time of RLLMs and deteriorates metrics like TPS.

**Summary.** This section suggests that many existing LLM inference optimization techniques can be directly applied to RLLMs seamless. However, surprisingly, some of these techniques have the opposite effect on smaller RLLMs, e.g., 7B. We leave the investigation of this phenomenon to future.

## 6 APPLYING TO REAL WORLD WORKLOAD

In previous section (§4), we have shown that the serving behaviors of RLLM is significantly different from the LLM. However, we assumed that the serving engine receives requests simultaneously in

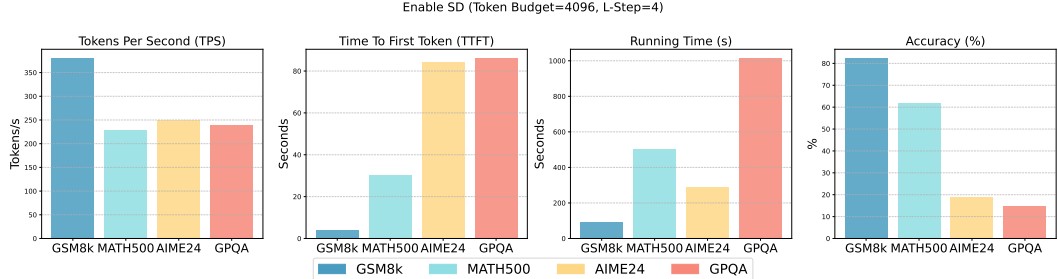

Figure 6: Empirical Results of comparison for enabling or disabling SD on 7B RLLM.

batches, with each new batch arriving only after the system has completed processing the previous one. This assumption may be overly idealized and not fully consistent with real-world conditions. Prior works (Wu et al., 2023; Li et al., 2023; Wang et al., 2025) have shown that, in real-world applications, the burstiness of requests received by the serving engine is typically modeled using the *Gamma distribution*. To validate our insights regarding RLLM serving in §4 under real-world scenarios, we implement a workload generator like BurstGPT-Perf (Wang et al., 2025) that is capable of producing requests following a Gamma distribution in our proposed Serve-Pref suite, enabling the generation of streaming, stochastic, and bursty workloads. We then perform empirical studies with it on various scale language models (7B, 14B, 32B) across different datasets to validate our findings.

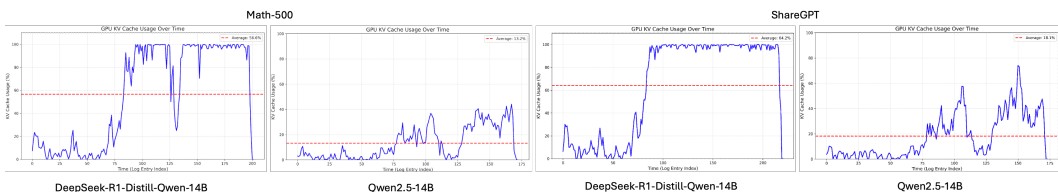

Figure 7: KV cache usage of 14B models under real-world workload across different datasets.

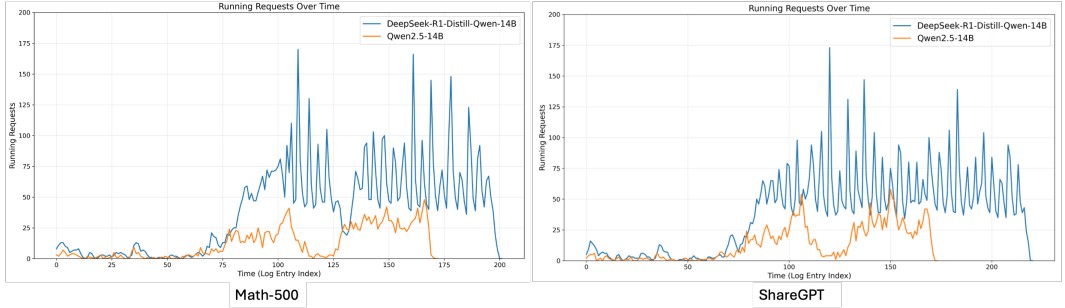

Figure 8: Num of running requests in the inference engine for 14B models under real-world workload.

**Main Results.** As shown in Figure 7, the average KV cache usage rate of RLLM is much higher than LLM. More surprisingly, for RLLM, the utilization of the serving engine's KV cache can remain close to 100% for long periods, forcing some new requests to wait in the waiting queue before running. This may significantly prolong request turnaround time in the serving engine, severely degrading user experience. We attribute the persistently high KV cache utilization to the accumulation of numerous stragglers in the system. The running requests in the engine are also much higher when serving RLLM compared to LLM, as shown in Figure 8. The above phenomena hold consistently across different datasets, demonstrating the generalizability of our findings. These results demonstrate our findings in §4 remain valid under real-world workloads. Please refer to Appendix K for more results.

## 7 RELATED WORK

We introduce necessary related work in this section. Extended related work can be found in §D.

**Reasoning Large Language Models.** Recent advancement in RLLM , such as OpenAI o1 (Jaech et al., 2024) have demonstrated significant improvement in system-2 tasks such as mathematics and programming via test time scaling , which generates long chain of thought (CoT) reasoning text before answer the question. Compared with CoT in traditional LLM, the reasoning process of RLLM have the following characteristics: (1) much longer reasoning process; (2) extensive exploration to unreached logic node; (3) backtrack and reflection; (4) *aha moment*. Since OpenAI's o1 and o3 (OpenAI, 2025) are proprietary, the research community has attempted to replicate their performance. s1 (Muennighoff et al., 2025) try to achieve test time scaling with only $1k$ post-training samples. LIMO (Ye et al., 2025a) exploits only $817$ curated training samples, improving scores from $6.5\%$ to $57.1\%$ on AIME dataset. DeepSeek R1 (Guo et al., 2025) is the first open-source RLLM and achieves on-par performance with OpenAI o1. Followed by (Face, 2025), which aims to fully reproduce R1 by the collaboration of open-source community. Recent cutting edge RLLMs such as QwQ (Team, 2025), Kimi K1.5 (Team et al., 2025), Gemini-2.5-flash (DeepMind, 2025), Seed-think-V1.5 (Seed et al., 2025), Qwen3 (team, 2025) have continually improve the performance of complex reasoning.

**LLM Inference and Serving.** Due to the large scale of LLM, they present considerable challenges in efficient serving, undermining the real world utilities of these models. Numerous works have been proposed to alleviate these problems from 6 different views: (1) model parameter memory optimization: model weight quantization like gptq (Frantar et al., 2023), awq (Lin et al., 2024), FP8 (Kuzmin et al., 2022), model pruning, model parallelism, CPU offloading ; (2) request scheduling: inter-request scheduling, and intra-request scheduling (3) dynamic memory optimization: KV cache quantization (vllm project, 2024), KV cache reuse, eviction, and dropping (Liu et al., 2025c;b); (4) efficient decoding: speculating decoding (Chen et al., 2023) (Li et al., 2024c) (Li et al., 2025c), flash decoding (Tri et al., 2023) ;(5) system optimization: prefill-decoding disaggregation like (Zhong et al., 2024) (Hu et al., 2024a) (Qin et al., 2025); (6) model and algorithm optimization: hard-aware algorithm like flash attention, flash-decoding (Tri et al., 2023), linear attention, mixture of experts; (7) Optimization for LLM-based Agent: Dual-Path (DeepSeek-AI, 2026), Continuum (Li et al., 2025a).

# 8 SUMMARY

In this section, we discuss the findings of above investigations, limitation of current serving engine, and potential direction for optimizing RLLM inference serving. Lastly, we conclude this paper.

**Insights for RLLM Serving.** Building on our empirical observations, we identify several actionable directions for optimizing future RLLM-oriented inference systems. *(1)* RLLMs exhibit higher and more volatile memory usage due to their long chains of thought, calling for finer-grained memory. This includes more adaptive re-prefill or KV reload mechanisms, improved cache lifetime management to smooth CoT boundary fluctuations, and selective KV swapping or offloading to ease memory pressure during decoding. *(2)* Large variations in reasoning length lead to strong straggler effects within batches, where hard queries dominate completion time and reduce tail utilization. To address this, systems should adopt difficulty-aware or runtime-adaptive scheduling that dynamically prioritizes or reorders requests. *(3)* Compared to standard LLMs, RLLMs are significantly more decode-heavy, with runtime determined more by task difficulty than input length. Systems should therefore employ asymmetric prefill–decode resource allocation, dedicating more resources to decode workers, adapting existing PD-disaggregation frameworks to RLLM workloads, and designing allocation policies that explicitly account for task-dependent decoding variance. *(4)* RLLMs generate much longer sequences, increasing bandwidth demand and KV-access intensity. This creates opportunities for hardware–software co-design, including KV-access–optimized accelerators, runtime systems that exploit memory hierarchies more effectively, and architectural refinements to reduce decode-phase latency. Together, these findings highlight the potential for inference syste, specialized for RLLMs.

**Conclusion.** In this work, we systematically investigate the serving performance and behavior of RLLM. We reveal that RLLMs have several different serving behavior compared with LLM, which makes current LLM serving engines struggle to unleash the power of RLLM and fall to reach the optimal performance. Additionally, we further investigate whether existing inference optimization techniques are valid for RLLM. Lastly, we conduct evaluation under real world workload modeled by Gamma distribution, and the results are *aligned* with our main findings regarding RLLM serving.

## ACKNOWLEDGEMENT

This work was partially supported by the National Natural Science Foundation of China under Grant No. 62272122, and Hong Kong CRF grants under Grant No. C7004-22G and C6015-23G.

## ETHICS STATEMENT

We, all the authors of this submission, hereby confirm that we have thoroughly read, fully understood, and explicitly acknowledge the ICLR Code of Ethics. We commit to strictly adhering to its principles and provisions in all aspects of our conference participation, including but not limited to paper submission, the reviewing process, and all discussions.

## REPRODUCIBILITY STATEMENT

Below, we summarize some critical aspects to facilitate reproducible results:

- Datasets. The datasets we used are all publicly accessible, which is introduced in G.1. The website for downloading these data is listed in F.
- Models. We provide the details about our adopted model and hyperparameters in F.
- Environment. All experiments are conducted with multiple runs on NVIDIA RTX4090-24GB GPUs, RTX A6000-48GB GPUs, and NVIDIA A100-PCIE-40GB GPUs with Python 3.11 and PyTorch 2.5.
- Code. Our code is available at https://github.com/lqinfdim/RLMServing.
- Project Homepage. The project homepage is at https://lqinfdim.github.io/project/rllm-serving/index.html.

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

APPENDIX AND SUPPLEMENTARY MATERIAL

## A    USE OF LLMS STATEMENT

We solemnly declare that the originality of ideas, writing, overall methodology, experiments, and other core contributions in this paper are entirely the work of the authors, with no involvement of any LLMs in the research process. LLMs were used solely for grammar checking and language polishing after drafting this submission.

## B    LIMITATION

In this work, we systematically investigate the serving performance of RLLM. Despite our comprehensive and thorough experiments, the evaluation of RLLM serving is limited in some extet due to limited support from the current ecosystem. We hope that future improvements in serving engines will enable broader and more comprehensive evaluations. Additionally, our hardware resources were limited, and we aim to extend our evaluations to a wider range of hardware platforms in the future.

## C    BOARDER IMPACT

In this paper, we systematically investigate the serving performance of RLLM. We hope our work can provide the research community and industry with insightful perspectives to help advance studies in efficient RLLM serving, help to democratize the use of cutting-edge RLLMs for social good.

## D    EXTENDED RELATED WORK

### D.1    REASONING LARGE LANGUAGE MODELS

Recent advancements in RLLM , such as OpenAI o1 (Jaech et al., 2024) have demonstrated significant improvement in system-2 tasks such as mathematics and programming via test time scaling, which generates long chain of thought (CoT) reasoning text before answer the question. Compared with chain-of-thought in traditional LLM, the reasoning process of RLLM has the following characteristics: (1) much longer reasoning process; (2) extensive exploration to unreached logic node; (3) backtrack and reflection; (4) aha moment. Since OpenAI's o1 and o3 (OpenAI, 2025) are proprietary models, the research community has attempted to replicate their performance. s1 (Muennighoff et al., 2025) try to achieve test time scaling with only $1k$ post-training samples. LIMO (Ye et al., 2025a) exploits only $817$ curated training samples, improving scores from $6.5\%$ to $57.1\%$ on AIME dataset. DeepSeek R1 (Guo et al., 2025) is the first open-source RLLM and achieves on-par performance with OpenAI o1. Followed by (Face, 2025), which aims to fully reproduce R1 by the collaboration of open-source community. Recent cutting edge RLLMs such as QwQ (Team, 2025), Kimi K1.5 (Team et al., 2025), Gemini-2.5-flash (DeepMind, 2025), Seed-think-v.15 (Seed et al., 2025), Qwen3 (team, 2025) have continually improve the performance on complex reasoning dataset.

### D.2    LLM INFERENCE AND SERVING

LLM has become a cornerstone of deep learning in recent years, reshaping the landscape of AI research. Due to the large scale of LLM, they present considerable challenges in efficient serving, undermining the real-world utilities of these models. Numerous works have been proposed to alleviate these problems from 6 different views: (1) model parameter memory optimization: model weight quantization like gptq (Frantar et al., 2023), awq (Lin et al., 2024), FP8 (Kuzmin et al., 2022), model pruning, model parallelism, CPU offloading ; (2) request scheduling: inter-request scheduling, and intra-request scheduling (3) dynamic memory optimization: KV cache quantization (vllm project, 2024), KV cache reuse and dropping (Liu et al., 2025c;b;a); (4) efficient decoding: speculating decoding (Chen et al., 2023) (Li et al., 2024c) (Li et al., 2025c), flash decoding (Tri et al., 2023) ;(5) system optimization: prefill-decoding disaggregation architecture like (Zhong et al., 2024) (Hu et al., 2024a) (Qin et al., 2025); (6) model and algorithm optimization: hard-aware algorithm like flash attention (Tri et al., 2023), linear attention, mixture of expert.

Recent advances in LLM inference have yielded a variety of specialized frameworks and serving engines that maximize GPU utilization through optimized kernels and memory strategies. High-

performance libraries such as NVIDIA's FasterTransformer (Shazeer, 2019) and TensorRT-LLM (NVIDIA, 2023), alongside open-source systems like vLLM (Kwon et al., 2023) and SGLang (Zheng et al., 2024), employ different techniques with continuous batching(Yu et al., 2022), speculative decoding(Chen et al., 2023), prefill-decode disaggregation(Zhong et al., 2024) and many other methods, ensuring the GPU pipeline remains saturated. Complementing these efforts are dynamic scheduling and memory management schemes that break large KV caches into reusable blocks and selectively merge or preempt operations, allowing much larger batch sizes with minimal overhead. Equally important are multi-way parallelism and algorithmic innovations that further boost throughput and reduce latency. Large models are commonly deployed across GPUs using tensor parallelism (splitting each layer's computation), pipeline parallelism (partitioning the model into sequential stages), and data parallel replication. Mixture-of-Experts (MoE) architectures extend this by routing tokens to different expert shards via expert parallelism, with communication optimizations to balance load. On the algorithmic side, parameter-efficient methods such as prompt and prefix tuning adapt frozen models via small "soft" prompts, speculative decoding (Chen et al., 2023) uses a lightweight draft model to accelerate token generation, and Simple Test-Time Scaling(Muennighoff et al., 2025) applies budget-forcing at inference to improve reasoning quality.

Together, these system-level designs and algorithm-level approaches form a cohesive ecosystem that drives state-of-the-art performance in efficient LLM serving. Please see survey papers (Li et al., 2024a; Kim et al., 2023; Zhen et al., 2025) for comprehensive introduction (Lazuka et al., 2024).

### D.3 LLM EVALUATION

Recently, with the rapid development of LLM, there is a growing interest in evaluating LLM from different aspects and topics. A holistic evaluation framework of language models is proposed (Liang et al., 2023). Generally, the technical reports like (Yang et al., 2024a; team, 2025; Guo et al., 2025; Liu et al., 2024a) of LLM provides pre-relase comprehensive evaluation results. The quantization methods for LLM are evaluated in (Jin et al., 2024) and (Li et al., 2024b). In (Li et al., 2025b), it evaluates the general abilities of post-edit LLM to assess the utility of existing knowledge editing methods. Work (Lazuka et al., 2024) and (Agrawal et al., 2024) evaluate LLM serving from a new perspective.

### D.4 ECOSYSTEM SUPPORT FOR RLLM SERVING.

The development of LLMs has greatly benefited from the research community and the open-source ecosystem, including open platforms such as Hugging Face, Github, and Modelscope; open-source LLMs like Llama (Grattafiori et al., 2024), Qwen (Yang et al., 2024a), and Deepseek R1; open-source LLM infrastructure such as Deepspeed (Rasley et al., 2020), Megatron-LM (Shoeybi et al., 2019), vLLM (Kwon et al., 2023), OpenRLHF (Hu et al., 2024c), and SGLang (Zheng et al., 2024); various optimization techniques like Flash-Attention (Dao et al., 2022), FlashInfer (Ye et al., 2025b), ZeRO (Rajbhandari et al., 2020), and LMCache (Liu et al., 2024b; Cheng et al., 2024; Yao et al., 2024). The advancement of RLLMs continues this trend. With the open-sourcing of Deepseek R1 (Guo et al., 2025), a large number of open-source RLLMs like Phi-4 reasoning (Abdin et al., 2025), and Llama-Nemotron (Bercovich et al., 2025) have emerged, further promoting the democratization of cutting-edge RLLM technology. Although existing LLM serving systems like vLLM, and SGLang provide some level of support for RLLMs, current support and optimization techniques remain significantly limited. Some techniques do not support RLLMs at all, for instance, Eagle speculative decoding currently lacks compatibility with RLLMs, while others fail to offer targeted optimizations and improvements specific to RLLM characteristics. As RLLMs continue to advance rapidly, we call on the research community and industry to collaborate (Tang et al., 2025; TANG et al., 2025; Dong et al., 2025; Lai et al., 2025) in addressing the issues revealed in this paper.

### D.5 ECOSYSTEM SUPPORT FOR RLLM SERVING.

The development of LLMs has greatly benefited from the research community and the open-source ecosystem, including open platforms such as Hugging Face, open-source LLMs like Llama Grattafiori et al. (2024), Qwen Yang et al. (2024a), open-source LLM infrastructure such as Deepspeed Rasley et al. (2020), Megatron-LM Shoeybi et al. (2019), vLLM Kwon et al. (2023), OpenRLHF Hu et al. (2024c), and SGLang Zheng et al. (2024) and various optimization techniques like Flash-Attention

Dao et al. (2022), FlashInfer Ye et al. (2025b), ZeRO Rajbhandari et al. (2020), and LMCache Liu et al. (2024b); Cheng et al. (2024); Yao et al. (2024). The advancement of RLLMs continues this trend. With the open-sourcing of Deepseek R1 Guo et al. (2025), a large number of open-source RLLMs like Phi-4 reasoning Abdin et al. (2025) , and Llama-Nemotron Bercovich et al. (2025) have emerged, further promoting the democratization of cutting-edge RLLM technology. Although existing LLM serving systems like vLLM, and SGLang provide some level of support for RLLMs, current support and optimization techniques remain significantly limited. Some techniques do not support RLLMs at all, for instance, Eagle speculative decoding currently lacks compatibility with RLLMs, while others fail to offer targeted optimizations and improvements specific to RLLM characteristics. As RLLMs continue to advance rapidly, we call on the research community and industry to collaborate in addressing the issues revealed in this paper.

# E   AN INTRODUCTION TO LLM SERVING

The highly increased development of LLMs' application arise the demand of effectively using LLM serving systems. In this part, we introduce some optimization methods for serving systems and introduce more serving metrics. For more comprehensive introduction, please refer to (Zhen et al., 2025) and (Li et al., 2024a).

## E.1   SERVING PERFORMANCE

Recently, there are a lot of researches focus on optimizing the performance of serving system based on LLM architecture's characteristics and system-level tricks. Current LLMs are mostly using decode-only architecture, making the KV values of former tokens becomes key information for the next token. Hence, the first useful methods is storing all of KV value in memories(particularly in GPUs), this method significantly improve the efficiency of prefill stage. However, this method had already deployed for language models. For LLM, the most important method proposed first is continuous batching(Yu et al., 2022). Continuous batching is processing requests in serving systems in iteration level, compared with former systems process requests in request-level. By using this technique, serving systems don't need to wait until the last request finishes its decoding, but replace requests with new requests once it ends decoding. This method enhance GPU's utilization, reducing waiting time for high-throughput serving systems. Next, considering the difference of prefill and decode that prefill is compute-intense stage which needs more GPU computing resources, while decode is memory-intense stage which needs more GPU memories compared with prefill, Prefill-Decode disaggregation(Zhong et al., 2024) proposed a method that process prefill and decode in different GPUs, fully utilizing GPU resources based on the characteristics of the two phases. Despite this, GPU resources are still not fully utilized because the GPU pre-allocates a portion of GPU space for requests when storing previous KV cache. However, much of this space isn't effectively used, resulting in significant waste (for example, if a request occupies 8 tokens, the GPU allocates 2080 token spaces for decoding this request, but actually only produces 80 tokens, wasting space for 2000 tokens). At the same time, since the GPU allocates and reserves space for requests sequentially, this can lead to memory fragmentation and inefficient resource utilization when requests complete at different times. Paged attention borrows the concept of CPU paging, and in their serving system (vLLM) creates a mapping between virtual addresses and actual GPU addresses through virtual pages(Kwon et al., 2023).

## E.2   SERVING METRIC

With the high demand of deploying customized LLMs for practical utilization, there is a need to measure the cost efficiency of different LLM serving solutions. The cost of serving RLLM depends on how many requests it can handle per second while being responsive to client users and supporting an acceptable level of answer accuracy. To measure the performance of LLM serving system, there are multiple metrics can be chosen: (1) Time to first token (TTFT) is the time it takes to process the prompt until generate the first token. It measures how long a user must wait before seeing the model's output; (2) End-to-end request latency (E2E latency) indicates the time it takes from submitting the first token of a request to receiving the last token of response, including the time for queueing and batching and network latencies in real-world scenario; (3) Time between tokens (TBT, a.k.a Intertoken latency, ITL) is the average time between the generation of consecutive tokens in a sequence; (4) Tokens per second (TPS) of system represents the mean of total output tokens number per second , accounting for all the requests happening simultaneously; (5) Requests per second (RPS) is the average number of requests that can be successfully completed by the system in a 1-second period. In LLM serving systems, there are many metrics evaluating the performance, In this paper, we use metrics for reference that companies and personal users care most while using RLLM. I'll introduce them here for clear understanding.

- **Throughput:** Number of processed requests per second. This is the key metric for users since it directly determines overall system performance.

- **Time to First Token (TTFT):** Time from receiving a request until the first token is generated (i.e., the prefill stage is completed). This reflects how quickly the serving system handles the

prefill stage. Techniques such as continuous batching (Yu et al., 2022) and paged attention (Kwon et al., 2023) were proposed to optimize this metric.

- **Time to First Visible Token (TTFVT):** The time from receiving a request until the first token is actually displayed to the user. This metric is specific to RLLMs because some inference systems hide the internal "thinking" steps and only reveal output once thinking is complete. Since RLLMs often perform a prolonged reasoning chain before producing any visible token, TTFVT is typically much larger than TTFT.

- **Time Between Tokens (TBT):** Average time between generation of consecutive tokens. For RLLMs, both the thinking stage and decoding stage share this metric. Recent algorithm-level optimizations such as S1 (Muennighoff et al., 2025) target TBT. In this paper, TBT reflects the real-time per-token responsiveness of the model during interactive generation, capturing both computational and scheduling overhead.

- **KV Cache Utilization:** Proportion of total memory occupied by the KV cache during model execution. High utilization enables reuse of KV values by subsequent requests, reducing prefill time. However, excessive utilization triggers frequent evictions, degrading performance. Section 4 analyzes KV cache utilization and its impact on overall performance for RLLMs across datasets of varying difficulty.

- **Tokens per Second (TPS):** Total number of tokens generated per second across all active sessions. This combines throughput and per-token speed into one measure of generation capacity.

- **Requests per Second (RPS):** Total number of full-request pipelines completed per second. Unlike throughput (which counts raw requests), RPS tracks end-to-end request handling.

- **Model Initialization Latency:** Total time from service startup—including loading model weights, constructing computation graphs, allocating GPU memory, initializing optimizers, and any warm-up steps—until the system is ready to handle its first request. For MoE models (such as the DeepSeek model used in this paper) with Tensor Parallelism (TP) and Pipeline Parallelism (PP), this also involves partitioning and distributing parameters across multiple GPUs. This metric helps compare how different serving systems optimize model loading and initialization.

- **End-to-End Latency (E2E Latency):** Time from user request submission until receipt of the final token. This metric significantly influences user experience; for enterprises, improving RLLM end-to-end latency is also a critical concern.

## F   IMPLEMENTATION AND REPRODUCTION CHECKLIST

In this section, we would like to provide details for reproducing our experimental results.

### F.1   PROJECT HOMEPAGE

The project homepage is at https://lqinfdim.github.io/project/rllm-serving/index.html.

### F.2   CODE BASE

Our code is available at https://github.com/lqinfdim/RLMServing.

### F.3   RELATED PROJECT

Here, we list all of the projects related to our manuscript.

- vLLM https://github.com/vllm-project/vllm
- SGLang https://github.com/sgl-project/sglang
- LMDeploy https://github.com/InternLM/lmdeploy

### F.4   MODELS

Here, we list all of the model checkpoints used in our experiments.

**RLLM checkpoints:**

- deepseek-ai/DeepSeek-R1-Distill-Qwen-1.5B        https://hf-mirror.com/deepseek-ai/DeepSeek-R1-Distill-Qwen-1.5B
- suayptalha/DeepSeek-R1-Distill-Llama-3B        https://huggingface.co/suayptalha/DeepSeek-R1-Distill-Llama-3B
- microsoft/Phi-4-mini-reasoning        https://huggingface.co/microsoft/Phi-4-mini-reasoning
- deepseek-ai/DeepSeek-R1-Distill-Qwen-7B        https://hf-mirror.com/deepseek-ai/DeepSeek-R1-Distill-Qwen-7B
- deepseek-ai/DeepSeek-R1-Distill-Llama-8B        https://huggingface.co/deepseek-ai/DeepSeek-R1-Distill-Llama-8B
- deepseek-ai/DeepSeek-R1-Distill-Qwen-14B        https://hf-mirror.com/deepseek-ai/DeepSeek-R1-Distill-Qwen-14B
- deepseek-ai/DeepSeek-R1-Distill-Qwen-32B        https://hf-mirror.com/deepseek-ai/DeepSeek-R1-Distill-Qwen-32B
- deepseek-ai/DeepSeek-R1-Distill-Llama-70B        https://hf-mirror.com/deepseek-ai/DeepSeek-R1-Distill-Llama-70B

**LLM checkpoints:**

- Qwen/Qwen2.5-Math-1.5B https://hf-mirror.com/Qwen/Qwen2.5-Math-1.5B
- meta-llama/Llama-3.2-3B-Instruct        https://huggingface.co/meta-llama/Llama-3.2-3B-Instruct
- microsoft/Phi-4-mini-instruct        https://huggingface.co/microsoft/Phi-4-mini-instruct
- Qwen/Qwen2.5-Math-7B https://hf-mirror.com/Qwen/Qwen2.5-Math-7B
- meta-llama/Llama-3.1-8B https://huggingface.co/meta-llama/Llama-3.1-8B

- Qwen/Qwen2.5-14B https://hf-mirror.com/Qwen/Qwen2.5-14B
- Qwen/Qwen2.5-32B https://hf-mirror.com/Qwen/Qwen2.5-32B
- meta-llama/Llama-3.3-70B-Instruct https://hf-mirror.com/meta-llama/Llama-3.3-70B-Instruct

## F.5 DATASETS

Here, we list all of the benchmarking datasets used in our experiments.

- GSM8K https://hf-mirror.com/datasets/openai/gsm8k
- MATH-500 https://hf-mirror.com/datasets/HuggingFaceH4/MATH-500
- AIME-24 https://hf-mirror.com/datasets/HuggingFaceH4/aime_2024
- GPQA https://hf-mirror.com/datasets/Idavidrein/gpqa

## F.6 HYPERPARAMETERS SETTINGS FOR RLLM

The hyperparameters settings for RLLM we employed are as follows:

Batch Size: 8, 16, 32

Dataset Capacity: 100, (AIME24 30)

Temperature: 0.6, Top-p: 0.95, Top-k: 20, Request Timeout: 1200 sec

Experiments Repeat Time: 3

Performance Only Mode: False

Reasoning LLM Mode: True

CoT Visible (for TTFT): False

## F.7 HYPERPARAMETERS SETTINGS FOR LLM

The hyperparameters settings for LLM we employed are as follows:

Batch Size: 8, 16, 32

Dataset Capacity: 100, (AIME24 30)

Temperature: 0.7, Top-p: 0.8, Top-k: 20, Request Timeout: 1200 sec

Experiments Repeat Time: 3

Performance Only Mode: False

Reasoning LLM Mode: False

CoT Visible (for TTFT): False

# G  EXPERIMENTS DETAILS

## G.1  DETAILS EVALUATION DATASETS

We use 4 different datasets in this paper, they are GSM8K, MATH500, AIME24, and GPQA. The details of these datasets are following.

- **GSM8K** (Cobbe et al., 2021): The GSM8K dataset is a large collection of mathematical problem-solving tasks designed for training and evaluating AI models in the context of elementary school-level math. It primarily focuses on grade school math word problems that require multiple steps of reasoning and calculations to solve.
- **MATH500** (Lightman et al., 2023): a challenging dataset consisting of problems from high school math competitions across seven subjects (e.g., Prealgebra, Algebra, Number Theory) and difficulty levels based on AoPS (ranging from 1 to 5). Problems in these competitions range from level 1, the easiest, often found in AMC 8 exams, to level 5, like those in AIME.
- **AIME24** (Committees, 2024):a dataset from the American Invitational Mathematics Examination, which tests math problem solving across multiple areas (e.g. algebra, counting, geometry, number theory, and probability). Because AIME 2024 contains only 30 examples, we don't considered examples of AIME from other years.
- **GPQA** (Rein et al., 2024): a graduate-level dataset consisting of multiple-choice questions in subdomains of physics, chemistry, and biology. For our experiment, we select the highest quality subset, known as GPQA Diamond (composed of 198 questions).

## G.2  RUNNING DEVICE

All of our experiments are running on three devices: a server with 8 RTX A6000 GPUs with 48GB VRAM, a server equipped with 8 RTX 4090 GPUs with 24GB VRAM, and a server with 8 NVIDIA A100-PCIE-40GB GPUs.

## G.3  INFERENCE ENGINE

We use vLLM (Kwon et al., 2023) version 0.8.1 and SGLang (Zheng et al., 2024) version 0.4.6.post1. For evaluation, we use OpenAI compatible API /v1/chat/completions.

## G.4  MODELS

Table 1: Employed Models in this work.

| RLLM Type | RLLM Name | LLM Type | LLM name |
|---|---|---|---|
| 1B RLLM | deepseek-ai/DeepSeek-R1-Distill-Qwen-1.5B | 1B LLM | Qwen/Qwen2.5-Math-1.5B |
| 3B RLLM | suayptalha/DeepSeek-R1-Distill-Llama-3B | 3B LLM | meta-llama/Llama-3.2-3B-Instruct |
| 4B RLLM | microsoft/Phi-4-mini-reasoning | 4B LLM | microsoft/Phi-4-mini-instruct |
| 7B RLLM | deepseek-ai/DeepSeek-R1-Distill-Qwen-7B | 7B LLM | Qwen/Qwen2.5-Math-7B |
| 8B RLLM | deepseek-ai/DeepSeek-R1-Distill-Llama-8B | 8B LLM | meta-llama/Llama-3.1-8B |
| 14B RLLM | deepseek-ai/DeepSeek-R1-Distill-Qwen-14B | 14B LLM | Qwen/Qwen2.5-14B |
| 32B RLLM | deepseek-ai/DeepSeek-R1-Distill-Qwen-32B | 32B LLM | Qwen/Qwen2.5-32B |
| 70B RLLM | deepseek-ai/DeepSeek-R1-Distill-Llama-70B | 70B LLM | meta-llama/Llama-3.3-70B-Instruct |

Table 2: Performance Metrics with PD-disaggregated

| Model | Method | Dataset | Accuracy | Running Time | Token Per Sec | TTFVT | Output Tokens |
|---|---|---|---|---|---|---|---|
| RLLM-7B | w/o PD-disa | GSM8K | 82 | 0:58 | 713.3 | 0.21 | 41789 |
| | w/ PD-disa | GSM8K | 82 | 1:12 | 516.9 | 0.3 | 42052 |
| | w/o PD-disa | MATH500 | 65 | 5:21 | 492.7 | 0.23 | 158431 |
| | w/ PD-disa | MATH500 | 64 | 7:57 | 323.2 | 0.4 | 154318 |
| | w/o PD-disa | AIME2024 | 266 | 2:01 | 935.3 | 0.27 | 112492 |
| | w/ PD-disa | AIME2024 | 23.3 | 2:40 | 711.7 | 0.4 | 113718 |
| | w/o PD-disa | GPQA | 11 | 6:52 | 894.2 | 0.36 | 368377 |
| | w/ PD-disa | GPQA | 20 | 9:19 | 662.0 | 0.6 | 369821 |

## H EXTEND OBSERVATION

### H.1 CAN DISAGGREGATED PREFILLING IMPROVE RLLM SERVING PERFORMANCE ?

As discussed in Section 2 and paper (Zhong et al., 2024), the process of LLM generates responds to a input prompt can be divided into two different phases. The LLM first processes input prompt in the *prefill phase*, which is computation intensive, to generate the first token of response within one iteration. After it , in the memory bounded *decoding phase*, LLM generates token one by one in each iteration until reaching the end token. These two phases have distinct different significance. However, many existing serving system co-locate the prefill and decoding at the same device, which may leads to sub-optimal performance and inter-phase interference as revealed in (Zhong et al., 2024). The disaggregated prefilling architecture was proposed to address this problem. It is first introduced in (Zhong et al., 2024), followed by lines of recent works like (Hu et al., 2024b), (Patel et al., 2024), (Hu et al., 2024a), (Qin et al., 2025), notably improving the TTFT and throughput of system. However, current support for disaggregated prefilling is experimental and only available in vLLM. What's more, the only disaggregated prefilling feature support in vLLM is 1P1D scheme (1 prefilling worker and 1 decoding worker) currently. Hence, we merely perform evaluation with 1P1D on 7B (on two RTX-4090 GPU) and 14B (on two A6000 GPU) models across 4 evaluation datasets.

**Main results.** The results of PD-disaggregation are shown in Table 2. We found that under the 1P1D setup, PD-disaggregation does not improve the serving performance of RLLMs. On the contrary, it leads to a decline in system performance metrics. We find that the performance bottleneck of 1P1D serving for RLLMs lies in decoding, while the devices used for pre-filling are largely idle, which leads to suboptimal performance. Additionally, PD-disaggregation requires KV cache transfer between GPUs, and the communication overhead negatively impacts the serving of RLLMs.

**Observation 6.** PD-disaggregation (1P1D) deteriorates RLLM serving metrics compared to mixed PD. Since nearly half of the computing resources are idle.

## I DETAILED EMPIRICAL RESULTS

### I.1 TOKEN BUDGET FOR PILOT STUDY

Full Figures of token budget exploration are listed in Figure 9.

### I.2 MAIN RESULTS FOR PILOT STUDY

We provide full results of RLLM and LLM serving comparison.

- 7B. RLLM in Table 3, LLM in Table 4
- 14B. RLLM in Table 5, LLM in Table 6
- 32B. RLLM in Table 7, LLM in Table 8
- 70B. RLLM in Table 9, LLM in Table 10

Table 3: Serving Results of RLLM-7B

| Model | BS | Budget | Dataset | Acc. | Running Time | TPS | TTFT | Output Tokens |
|-------|----|--------|---------|------|--------------|-----|------|---------------|
| | | 4096 | GSM8k | 80.67 | 1m53s | 426.87 | 2.4200 | 129629 |
| | | | MATH500 | 61.33 | 9m34 | 302.47 | 16.5500 | 502833 |
| | | | AIME24 | 23.33 | 4m7s | 470.37 | 40.8200 | 340730 |
| | | | GPQA | 15.00 | 13m29s | 477.14 | 35.3600 | 1124673 |
| | 8 | 8192 | GSM8k | 82.33 | 1m46s | 443.95 | 2.3400 | 126155 |
| | | | MATH500 | 64.33 | 14m09 | 240.55 | 20.2900 | 601161 |
| | | | AIME24 | 38.89 | 8m7s | 409.92 | 64.9600 | 592434 |
| | | | GPQA | 26.67 | 26m42s | 410.33 | 65.9200 | 1941071 |
| | | 4096 | GSM8k | 84.30 | 1m1s | 791.40 | 2.3900 | 126451 |
| | | | MATH500 | 59.30 | 5m56s | 499.45 | 17.1000 | 505796 |
| | | | AIME24 | 20.00 | 2m15s | 872.91 | 46.4700 | 346986 |
| | | | GPQA | 14.67 | 7m40s | 838.80 | 39.3300 | 1124379 |
| RLLM-7B | 16 | 8192 | GSM8k | 85.00 | 1m1s | 783.22 | 2.3900 | 125992 |
| | | | MATH500 | 62.30 | 10m29s | 351.14 | 22.2400 | 623861 |
| | | | AIME24 | 37.78 | 4m35s | 736.40 | 73.4100 | 590741 |
| | | | GPQA | 25.33 | 15m21s | 706.20 | 72.5800 | 1921405 |
| | | 4096 | GSM8k | 80.67 | 28s | 1700.96 | 3.7200 | 128915 |
| | | | MATH500 | 60.67 | 2m16s | 1303.67 | 25.1500 | 506862 |
| | | | AIME24 | 18.89 | 1m22s | 1413.73 | 52.9100 | 345118 |
| | | | GPQA | 14.67 | 3m22s | 1929.84 | 62.5700 | 1131700 |
| | 64 | 8192 | GSM8k | 84.67 | 25s | 1872.10 | 3.5800 | 125931 |
| | | | MATH500 | 61.33 | 4m1s | 897.99 | 30.3300 | 624385 |
| | | | AIME24 | 35.56 | 2m46s | 1222.58 | 82.1100 | 600925 |
| | | | GPQA | 28.00 | 6m57s | 1567.52 | 102.1000 | 1936615 |

Table 4: Serving Results of LLM-7B

| Model | BS | Budget | Dataset | Acc. | Running Time | TPS | TTFT | Output Tokens |
|-------|----|--------|---------|------|--------------|-----|------|---------------|
| | | | GSM8k | 69.67 | 1m47s | 394.13 | 0.0600 | 107378 |
| | 8 | 4096 | MATH500 | 3.30 | 3m19s | 343.65 | 0.0713 | 178459 |
| | | | AIME24 | 15.56 | 1m34s | 392.93 | 0.0776 | 101905 |
| | | | GPQA | 3.00 | 3m45s | 324.91 | 0.1366 | 183061 |
| | | | GSM8k | 70.00 | 1m33s | 477.32 | 0.0991 | 115613 |
| LLM-7B | 16 | 4096 | MATH500 | 1.67 | 2m13s | 513.34 | 0.1258 | 178452 |
| | | | AIME24 | 18.89 | 50s | 699.36 | 0.1208 | 95181 |
| | | | GPQA | 0.04 | 2m42s | 495.68 | 0.2114 | 204317 |
| | | | GSM8k | 67.67 | 57s | 762.61 | 0.1698 | 111292 |
| | 32 | 4096 | MATH500 | 1.67 | 1m31s | 748.09 | 0.2006 | 176704 |
| | | | AIME24 | 16.67 | 33s | 1063.23 | 0.1971 | 94296 |
| | | | GPQA | 6.00 | 1m34s | 754.67 | 0.3807 | 175708 |

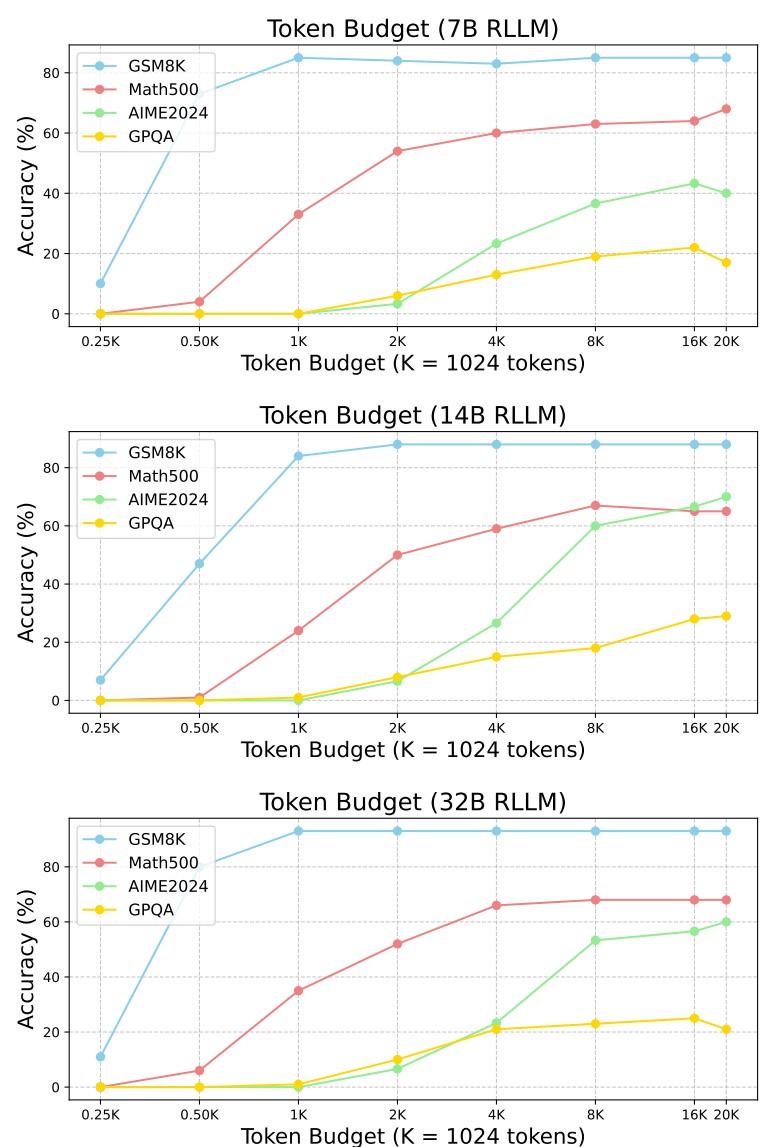

Figure 9: Results of token budget variation across different datasets for different scale RLLM .

## I.3 SERVING BEHAVIORS FOR PILOT STUDY

For better presentation, we provide illustration about 14B and 32B model serving visualization in Figure 10 and 11.

Table 5: Serving Results of RLLM-14B

| Model | BS | Budget | Dataset | Acc. | Running Time | TPS | TTFT | Output Tokens |
|---|---|---|---|---|---|---|---|---|
| RLLM-14B | 8 | 4096 | GSM8k | 87.00 | 3m50s | 280.83 | 5.9731 | 177347 |
| | | | MATH500 | 55.00 | 11m49s | 277.63 | 20.0256 | 566267 |
| | | | AIME24 | 27.78 | 4m42s | 411.29 | 50.6600 | 341998 |
| | | | GPQA | 16.33 | 15m25s | 406.14 | 39.0104 | 1090910 |
| | | 8192 | GSM8k | 87.67 | 3m30s | 303.34 | 6.0476 | 175210 |
| | | | MATH500 | 62.33 | 17m17s | 218.77 | 25.5350 | 656437 |
| | | | AIME24 | 47.78 | 9m18s | 338.41 | 72.4221 | 558966 |
| | | | GPQA | 21.00 | 30m34s | 341.19 | 67.0687 | 1842315 |
| | 16 | 4096 | GSM8k | 86.33 | 2m40s | 402.48 | 7.6400 | 173204 |
| | | | MATH500 | 60.33 | 8m27s | 388.92 | 27.2100 | 563198 |
| | | | AIME24 | 27.78 | 3m11s | 594.52 | 61.7600 | 335082 |
| | | | GPQA | 15.33 | 10m48s | 581.23 | 52.2400 | 1098423 |
| | | 8192 | GSM8k | 86.33 | 3m01s | 382.72 | 8.1400 | 180684 |
| | | | MATH500 | 63.67 | 13m25s | 291.14 | 33.2600 | 669369 |
| | | | AIME24 | 47.78 | 6m28s | 498.79 | 109.3600 | 568311 |
| | | | GPQA | 22.00 | 21m16s | 481.30 | 95.8800 | 1820128 |
| | 32 | 4096 | GSM8k | 85.33 | 1m38s | 619.91 | 10.1400 | 169539 |
| | | | MATH500 | 57.33 | 5m38s | 568.13 | 36.8600 | 554485 |
| | | | AIME24 | 22.22 | 2m20s | 839.99 | 92.3600 | 344929 |
| | | | GPQA | 15.00 | 8m18s | 770.56 | 78.5500 | 1104010 |
| | | 8192 | GSM8k | 86.33 | 2m13s | 486.36 | 11.0500 | 182233 |
| | | | MATH500 | 66.00 | 9m09s | 414.53 | 42.8600 | 654295 |
| | | | AIME24 | 50.00 | 4m22s | 722.93 | 140.4300 | 563372 |
| | | | GPQA | 26.67 | 15m36s | 658.74 | 136.5100 | 1818245 |

Table 6: Serving Results of LLM-14B

| Model | BS | Budget | Dataset | Acc. | Running Time | TPS | TTFT | Output Tokens |
|---|---|---|---|---|---|---|---|---|
| LLM-14B | 8 | 4096 | GSM8k | 74.17 | 1m18s | 317.73 | 0.0603 | 60286 |
| | | | MATH500 | 44.00 | 2m29s | 286.86 | 0.0626 | 100818 |
| | | | AIME24 | 2.22 | 1m47s | 240.82 | 0.0626 | 69874 |
| | | | GPQA | 29.00 | 2m15s | 305.50 | 0.1222 | 87836 |
| | | 8192 | GSM8k | 74.17 | 1m16s | 337.06 | 0.0164 | 59314 |
| | | | MATH500 | 44.67 | 2m23s | 282.50 | 0.0635 | 99443 |
| | | | AIME24 | 3.33 | 1m54s | 252.58 | 0.0619 | 77607 |
| | | | GPQA | 29.33 | 2m09s | 228.90 | 0.1217 | 90576 |
| | 16 | 4096 | GSM8k | 77.33 | 44s | 529.30 | 0.1536 | 55856 |
| | | | MATH500 | 46.00 | 1m57s | 365.86 | 0.0913 | 106980 |
| | | | AIME24 | 3.33 | 1m05s | 388.44 | 0.0913 | 66109 |
| | | | GPQA | 25.33 | 1m40s | 424.31 | 0.2250 | 93954 |
| | | 8192 | GSM8k | 77.33 | 45s | 554.18 | 0.0850 | 57030 |
| | | | MATH500 | 47.00 | 1m53s | 385.42 | 0.0920 | 109010 |
| | | | AIME24 | 4.44 | 58s | 409.73 | 0.9020 | 62213 |
| | | | GPQA | 24.67 | 2m05s | 381.19 | 0.2547 | 97573 |
| | 32 | 4096 | GSM8k | 74.09 | 44s | 615.89 | 0.1350 | 60318 |
| | | | MATH500 | 47.67 | 1m04s | 597.29 | 0.1393 | 94561 |
| | | | AIME24 | 5.56 | 41s | 616.16 | 0.1354 | 68309 |
| | | | GPQA | 28.00 | 1m17s | 403.77 | 0.4789 | 95110 |
| | | 8192 | GSM8k | 74.59 | 52s | 526.30 | 0.1320 | 61833 |
| | | | MATH500 | 46.67 | 1m20s | 523.91 | 0.1385 | 102168 |
| | | | AIME24 | 2.22 | 41s | 620.70 | 0.1445 | 69811 |
| | | | GPQA | 28.67 | 1m19s | 541.19 | 0.4215 | 93025 |
| | 64 | 4096 | GSM8k | 84.00 | 2m15s | 463.17 | 7.3815 | 169635 |
| | | | MATH500 | 59.33 | 7m57s | 406.74 | 25.6479 | 560880 |
| | | | AIME24 | 23.33 | 3m3s | 630.49 | 58.5514 | 340172 |
| | | | GPQA | 17.00 | 10m22s | 607.50 | 50.8592 | 1098715 |
| | | 8192 | GSM8k | 88.00 | 2m39s | 403.58 | 7.2181 | 174708 |
| | | | MATH500 | 68.00 | 12m53s | 299.83 | 32.5928 | 667886 |
| | | | AIME24 | 53.33 | 6m7s | 532.62 | 112.0436 | 577277 |
| | | | GPQA | 25.33 | 20m43s | 504.45 | 86.7966 | 1846931 |

Table 7: Serving Results of RLLM-32B

| Model | BS | Budget | Dataset | Acc. | Running Time | TPS | TTFT | Output Tokens |
|---|---|---|---|---|---|---|---|---|
| RLLM-32B | 8 | 4096 | GSM8k | 91.00 | 4m16s | 192.12 | 5.2715 | 130170 |
| | | | MATH500 | 64.00 | 24m58s | 125.59 | 42.9319 | 537799 |
| | | | AIME24 | 21.11 | 9m12s | 215.58 | 104.0663 | 349099 |
| | | | GPQA | 20.00 | 5m0s | 206.07 | 73.0046 | 1071843 |
| | | 8192 | GSM8k | 90.33 | 4m11s | 194.77 | 5.2037 | 129401 |
| | | | MATH500 | 70.67 | 36m41s | 97.83 | 51.2373 | 621128 |
| | | | AIME24 | 45.56 | 18m36s | 174.28 | 150.4045 | 575207 |
| | | | GPQA | 24.33 | 60m31s | 166.67 | 129.2077 | 1779384 |
| | 16 | 4096 | GSM8k | 90.67 | 2m27s | 324.74 | 5.4900 | 128546 |
| | | | MATH500 | 66.33 | 14m49 | 201.98 | 44.6300 | 517659 |
| | | | AIME24 | 25.56 | 5m03s | 388.52 | 111.2300 | 346308 |
| | | | GPQA | 21.67 | 17m25s | 352.72 | 74.0400 | 1070143 |
| | | 8192 | GSM8k | 92.67 | 2m30s | 324.04 | 5.5700 | 128060 |
| | | | MATH500 | 68.33 | 25m38s | 134.29 | 53.4000 | 597129 |
| | | | AIME24 | 48.89 | 10m22s | 309.03 | 170.1700 | 568936 |
| | | | GPQA | 28.67 | 35m30s | 283.98 | 137.9600 | 1778995 |
| | 32 | 4096 | GSM8k | 91.33 | 1m39s | 490.92 | 6.7103 | 129986 |
| | | | MATH500 | 66.67 | 9m29s | 309.67 | 47.9789 | 504391 |
| | | | AIME24 | 28.89 | 3m01s | 631.69 | 119.5658 | 335582 |
| | | | GPQA | 18.00 | 11m25s | 541.78 | 94.9504 | 1077209 |
| | | 8192 | GSM8k | 92.33 | 1m39s | 494.92 | 6.4570 | 129510 |
| | | | MATH500 | 68.67 | 16m6s | 213.37 | 60.4897 | 593685 |
| | | | AIME24 | 50.00 | 6m06s | 502.69 | 186.3563 | 547959 |
| | | | GPQA | 23.00 | 23m13s | 436.05 | 171.4893 | 1787777 |

Table 8: Serving Results of LLM-32B

| Model | BS | Budget | Dataset | Acc. | Running Time | TPS | TTFT | Output Tokens |
|---|---|---|---|---|---|---|---|---|
| LLM-32B | 8 | 4096 | GSM8k | 60.67 | 7m25s | 87.34 | 0.1271 | 100271 |
| | | | MATH500 | 46.67 | 8m24s | 103.76 | 0.1339 | 131828 |
| | | | AIME24 | 6.67 | 2m43s | 131.04 | 0.1414 | 57729 |
| | | | GPQA | 29.00 | 7m6s | 141.18 | 0.2291 | 144033 |
| | | 8192 | GSM8k | 60.67 | 7m10s | 89.53 | 0.0867 | 98287 |
| | | | MATH500 | 44.00 | 7m42s | 109.18 | 0.0874 | 127011 |
| | | | AIME24 | 4.44 | 3m24s | 127.82 | 0.0899 | 71654 |
| | | | GPQA | 29.00 | 6m18s | 152.47 | 0.1141 | 136446 |
| | 16 | 4096 | GSM8k | 66.33 | 4m03s | 130.10 | 0.1083 | 83076 |
| | | | MATH500 | 49.00 | 4m32s | 170.74 | 0.1104 | 115631 |
| | | | AIME24 | 12.22 | 2m13s | 185.90 | 0.1092 | 67231 |
| | | | GPQA | 31.00 | 4m23s | 210.62 | 0.1544 | 139381 |
| | | 8192 | GSM8k | 62.67 | 4m04s | 126.56 | 0.1107 | 92586 |
| | | | MATH500 | 46.03 | 5m01s | 156.52 | 0.1115 | 122183 |
| | | | AIME24 | 6.67 | 1m53s | 200.03 | 0.1179 | 60391 |
| | | | GPQA | 23.33 | 3m46s | 259.97 | 0.1659 | 136660 |
| | 32 | 4096 | GSM8k | 63.00 | 5m24s | 125.09 | 0.1902 | 96723 |
| | | | MATH500 | 45.00 | 5m44 | 148.90 | 0.2097 | 123313 |
| | | | AIME24 | 7.78 | 1m39s | 211.36 | 0.2206 | 57192 |
| | | | GPQA | 24.33 | 4m06s | 231.08 | 0.3894 | 141199 |
| | | 8192 | GSM8k | 62.15 | 3m59s | 129.75 | 0.1094 | 87157 |
| | | | MATH500 | 44.37 | 5m31s | 148.39 | 0.1109 | 128664 |
| | | | AIME24 | 7.78 | 1m52s | 200.63 | 0.1113 | 63495 |
| | | | GPQA | 25.33 | 4m31s | 246.55 | 0.1872 | 136403 |

Table 9: Serving Results of RLLM-70B

| Model | BS | Budget | Dataset | Acc. | Running Time | TPS | TTFT | Output Tokens |
|---|---|---|---|---|---|---|---|---|
| RLLM-70B | 8 | 4096 | GSM8k | 90.00 | 5m22s | 146.65 | 6.6450 | 125391 |
| | | | MATH500 | 54.67 | 30m29s | 102.48 | 48.4212 | 538966 |
| | | | AIME24 | 32.22 | 11m46s | 162.46 | 108.0681 | 336881 |
| | | | GPQA | 22.67 | 38m11s | 158.29 | 93.9947 | 1052080 |
| | | 8192 | GSM8k | 90.00 | 5m30s | 143.90 | 6.5438 | 125874 |
| | | | MATH500 | 62.00 | 49m48s | 78.31 | 61.8861 | 677161 |
| | | | AIME24 | 54.44 | 23m18s | 135.50 | 192.8731 | 561002 |
| | | | GPQA | 29.67 | 75m38s | 128.03 | 159.9508 | 1702698 |
| | 16 | 4096 | GSM8k | 88.33 | 3m24s | 230.39 | 7.4600 | 123185 |
| | | | MATH500 | 57.00 | 19m45s | 158.60 | 52.3600 | 539666 |
| | | | AIME24 | 26.67 | 7m01s | 277.08 | 140.8100 | 342355 |
| | | | GPQA | 23.00 | 23m35s | 253.66 | 98.0500 | 1042191 |
| | | 8192 | GSM8k | 88.00 | 3m32s | 228.88 | 7.7200 | 125291 |
| | | | MATH500 | 60.67 | 32m27s | 112.85 | 69.6400 | 644132 |
| | | | AIME24 | 55.56 | 13m46s | 221.77 | 197.0300 | 539132 |
| | | | GPQA | 30.67 | 46m01s | 204.78 | 181.9000 | 1659750 |
| | 32 | 4096 | GSM8k | 88.67 | 2m11s | 352.22 | 9.7962 | 122786 |
| | | | MATH500 | 56.67 | 13m0s | 238.82 | 66.2581 | 534918 |
| | | | AIME24 | 25.56 | 4m26s | 438.58 | 184.7638 | 343514 |
| | | | GPQA | 23.00 | 15m57s | 378.78 | 136.1667 | 1052451 |
| | | 8192 | GSM8k | 89.00 | 2m12s | 352.59 | 9.4598 | 123720 |
| | | | MATH500 | 62.33 | 21m36s | 166.26 | 83.2369 | 621237 |
| | | | AIME24 | 51.11 | 8m24s | 360.36 | 246.0471 | 537908 |
| | | | GPQA | 32.00 | 30m31s | 307.04 | 228.6145 | 1650647 |

Table 10: Serving Results of LLM-70B

| Model | BS | Budget | Dataset | Acc. | Running Time | TPS | TTFT | Output Tokens |
|---|---|---|---|---|---|---|---|---|
| LLM-70B | 8 | 4096 | GSM8k | 93.00 | 3m03s | 144.56 | 0.2030 | 62746 |
| | | | MATH500 | 59.33 | 10m38s | 90.67 | 0.2283 | 148624 |
| | | | AIME24 | 30.00 | 5m35s | 107.39 | 0.2461 | 99826 |
| | | | GPQA | 53.33 | 12m18s | 128.53 | 0.4749 | 243123 |
| | | 8192 | GSM8k | 92.67 | 2m58s | 144.42 | 0.1143 | 60838 |
| | | | MATH500 | 59.00 | 13m12s | 78.52 | 0.1197 | 162129 |
| | | | AIME24 | 23.33 | 6m1s | 99.04 | 0.1232 | 98349 |
| | | | GPQA | 51.67 | 11m24s | 131.86 | 0.1847 | 233596 |
| | 16 | 4096 | GSM8k | 91.33 | 1m44s | 243.39 | 0.1525 | 60890 |
| | | | MATH500 | 60.00 | 6m35s | 139.78 | 0.1730 | 144651 |
| | | | AIME24 | 27.78 | 3m06s | 171.31 | 0.1678 | 89789 |
| | | | GPQA | 47.67 | 7m28s | 201.44 | 0.2694 | 231586 |
| | | 8192 | GSM8k | 93.67 | 1m52s | 227.03 | 0.1536 | 61434 |
| | | | MATH500 | 59.00 | 6m38s | 139.61 | 0.1704 | 142362 |
| | | | AIME24 | 27.78 | 5m13s | 124.39 | 0.1715 | 103021 |
| | | | GPQA | 49.00 | 10m26s | 155.10 | 0.2551 | 247291 |
| | 32 | 4096 | GSM8k | 92.33 | 1m17s | 346.24 | 0.6077 | 61987 |
| | | | MATH500 | 60.33 | 5m29s | 176.59 | 0.6520 | 149612 |
| | | | AIME24 | 26.56 | 2m37s | 243.93 | 0.7012 | 106146 |
| | | | GPQA | 51.00 | 5m40s | 268.96 | 1.5823 | 237672 |
| | | 8192 | GSM8k | 93.00 | 1m14s | 357.73 | 0.2325 | 60831 |
| | | | MATH500 | 61.00 | 7m7s | 148.03 | 0.2520 | 164985 |
| | | | AIME24 | 27.78 | 3m23s | 186.51 | 0.2550 | 105595 |
| | | | GPQA | 53.33 | 4m54s | 306.15 | 0.5346 | 233070 |

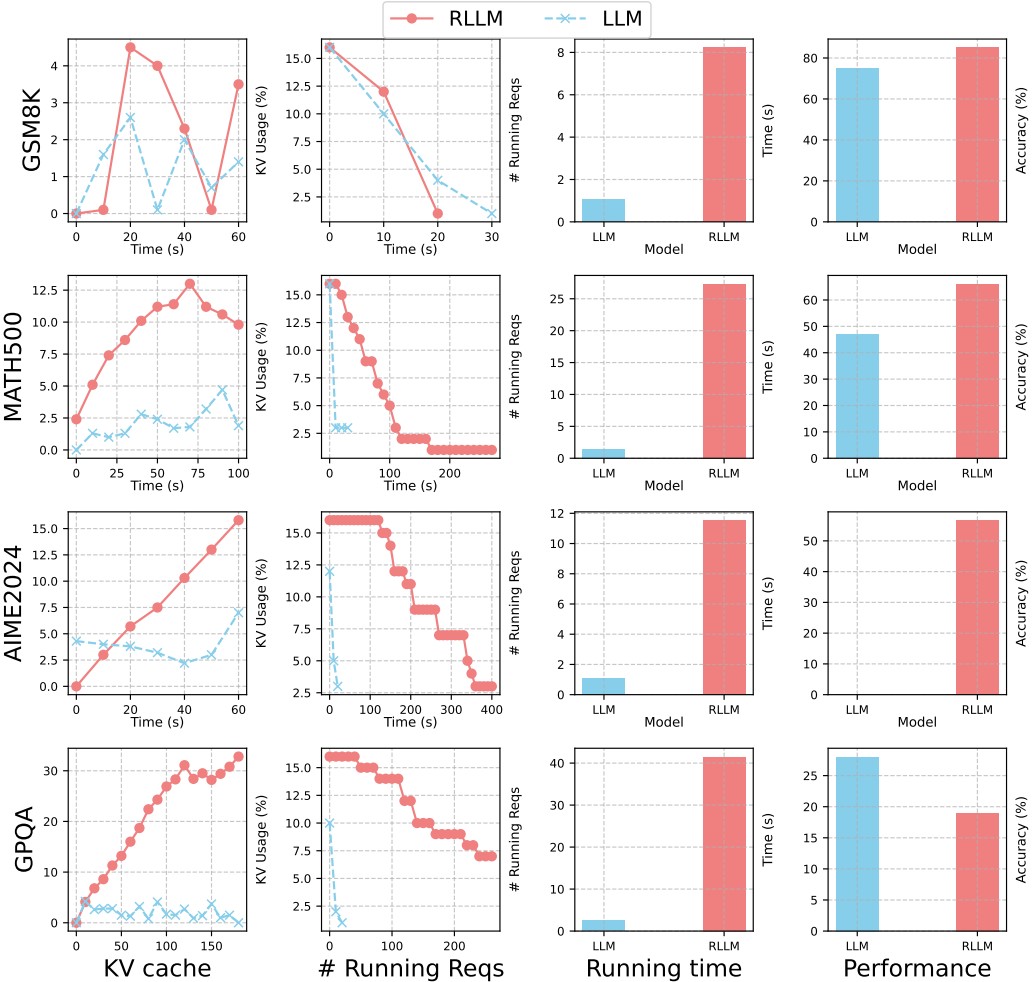

Figure 10: Results of RLLM vs LLM for 14B model size .

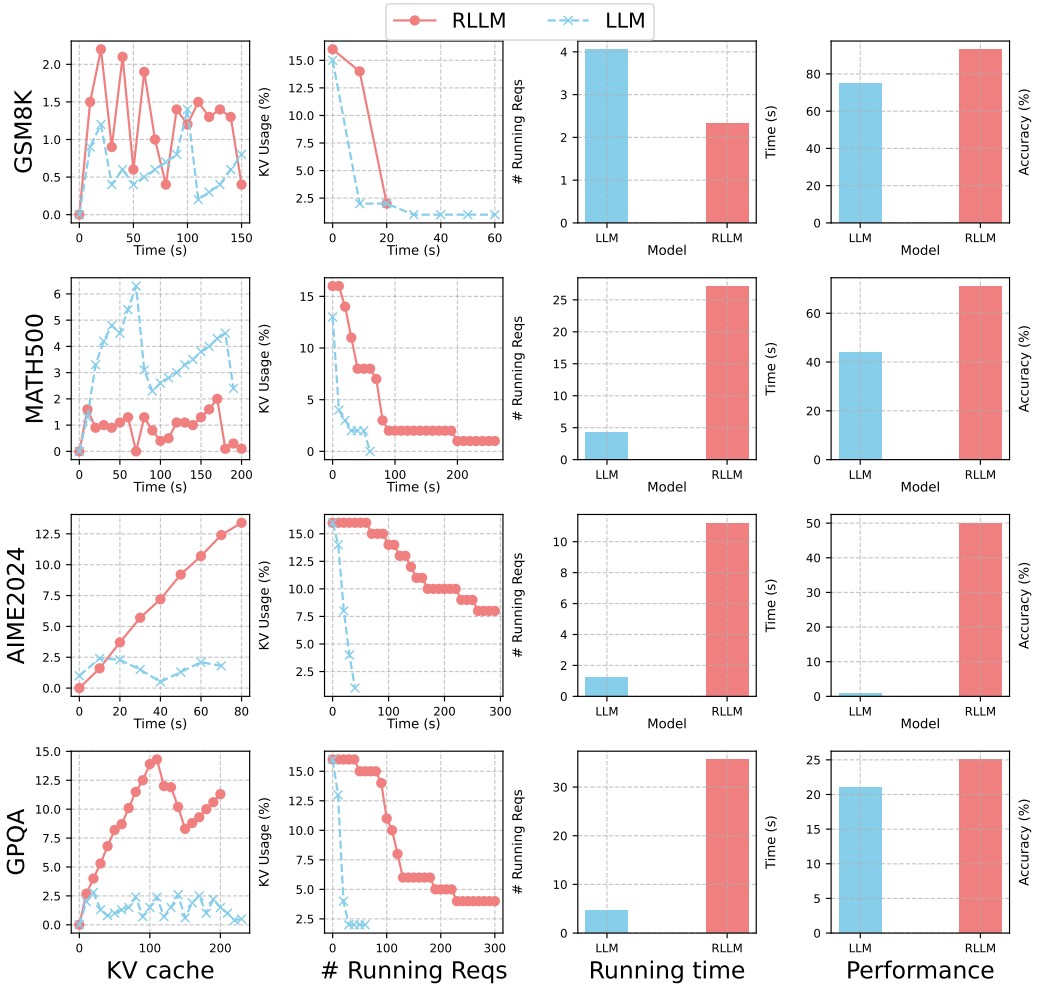

Figure 11: Results of RLLM vs LLM for 32B model size .

I.4   RUNNING TRACES DEMO

INFO:         127.0.0.1:53458 − "POST /v1/chat/completions HTTP/1.1"
   200 OK

INFO 05−10 13:01:49 [loggers.py:80] Avg prompt throughput: 223.0
   tokens/s, Avg generation throughput: 164.5 tokens/s, Running:
   16 reqs, Waiting: 0 reqs, GPU KV cache usage: 1.0\%, Prefix
   cache hit rate: 11.6\%

INFO 05−10 13:01:59 [loggers.py:80] Avg prompt throughput: 0.0
   tokens/s, Avg generation throughput: 650.0 tokens/s, Running:
   14 reqs, Waiting: 0 reqs, GPU KV cache usage: 2.8\%, Prefix
   cache hit rate: 11.6\%

INFO 05−10 13:02:09 [loggers.py:80] Avg prompt throughput: 0.0
   tokens/s, Avg generation throughput: 495.5 tokens/s, Running:
   11 reqs, Waiting: 0 reqs, GPU KV cache usage: 3.9\%, Prefix
   cache hit rate: 11.6\%

INFO 05−10 13:02:19 [loggers.py:80] Avg prompt throughput: 0.0
   tokens/s, Avg generation throughput: 406.1 tokens/s, Running:
   8 reqs, Waiting: 0 reqs, GPU KV cache usage: 4.0\%, Prefix
   cache hit rate: 11.6\%

INFO 05−10 13:02:29 [loggers.py:80] Avg prompt throughput: 0.0
   tokens/s, Avg generation throughput: 320.8 tokens/s, Running:
   8 reqs, Waiting: 0 reqs, GPU KV cache usage: 5.1\%, Prefix
   cache hit rate: 11.6\%

INFO 05−10 13:02:39 [loggers.py:80] Avg prompt throughput: 0.0
   tokens/s, Avg generation throughput: 320.7 tokens/s, Running:
   6 reqs, Waiting: 0 reqs, GPU KV cache usage: 4.9\%, Prefix
   cache hit rate: 11.6\%

INFO 05−10 13:02:49 [loggers.py:80] Avg prompt throughput: 0.0
   tokens/s, Avg generation throughput: 271.3 tokens/s, Running:
   5 reqs, Waiting: 0 reqs, GPU KV cache usage: 4.9\%, Prefix
   cache hit rate: 11.6\%

INFO 05−10 13:02:59 [loggers.py:80] Avg prompt throughput: 0.0
   tokens/s, Avg generation throughput: 180.7 tokens/s, Running:
   4 reqs, Waiting: 0 reqs, GPU KV cache usage: 4.5\%, Prefix
   cache hit rate: 11.6\%

INFO 05−10 13:03:09 [loggers.py:80] Avg prompt throughput: 0.0
   tokens/s, Avg generation throughput: 164.8 tokens/s, Running:
   4 reqs, Waiting: 0 reqs, GPU KV cache usage: 5.0\%, Prefix
   cache hit rate: 11.6\%

INFO 05−10 13:03:19 [loggers.py:80] Avg prompt throughput: 0.0
   tokens/s, Avg generation throughput: 174.4 tokens/s, Running:
   4 reqs, Waiting: 0 reqs, GPU KV cache usage: 5.7\%, Prefix
   cache hit rate: 11.6\%

# J   DETAILED EMPIRICAL RESULTS FOR RLLM SERVING OPTIMIZATION

## J.1   MODEL WEIGHT QUANTIZATION

Full results of model weight quantization with different models are listed in Table 11 and 12.

Table 11: Results of RLLM-7B with Different Quantization Methods

| Model | Method | Dataset | Acc. | Running Time | TPS | TTFT | Output Tokens |
|---|---|---|---|---|---|---|---|
| | | | | Budget-4096 | | | |
| RLLM-7B | GPTQ | GSM8k | 81.67 | 36s | 1258.52 | 1.5015 | 125477 |
| | | MATH500 | 61.00 | 3m34s | 807.99 | 10.8316 | 488868 |
| | | AIME24 | 16.67 | 1m25s | 1383.83 | 28.3623 | 346988 |
| | | GPQA | 16.00 | 4m48s | 1342.53 | 23.6366 | 1127392 |
| | AWQ | GSM8k | 82.67 | 2m36s | 306.10 | 5.7903 | 128626 |
| | | MATH500 | 59.33 | 14m55s | 195.01 | 41.2395 | 501037 |
| | | AIME24 | 21.11 | 5m02s | 390.68 | 104.9071 | 347634 |
| | | GPQA | 14.33 | 17m22s | 371.37 | 89.8281 | 1124314 |
| | FP8 | GSM8k | 82.67 | 58s | 805.44 | 1.8947 | 128016 |
| | | MATH500 | 64.00 | 4m44s | 589.49 | 13.9700 | 479989 |
| | | AIME24 | 25.56 | 1m49s | 1062.28 | 38.5411 | 341103 |
| | | GPQA | 15.00 | 6m13s | 1029.07 | 29.9774 | 1110873 |
| | L4 | GSM8k | 82.67 | 1m21s | 560.40 | 3.5042 | 119344 |
| | | MATH500 | 61.00 | 7m58s | 356.31 | 24.6086 | 486082 |
| | | AIME24 | 20.00 | 3m05s | 630.92 | 60.3252 | 343397 |
| | | GPQA | 15.33 | 10m36s | 599.34 | 51.2468 | 1109222 |
| | | | | Budget-8192 | | | |
| RLLM-7B | GPTQ | GSM8k | 80.33 | 57s | 955.97 | 1.5007 | 135978 |
| | | MATH500 | 64.00 | 5m54s | 591.44 | 12.3735 | 600650 |
| | | AIME24 | 31.11 | 2m59s | 1156.19 | 44.9320 | 618796 |
| | | GPQA | 25.33 | 9m47s | 1131.34 | 44.2137 | 1959609 |
| | AWQ | GSM8k | 80.00 | 2m35s | 306.42 | 5.6545 | 126262 |
| | | MATH500 | 66.67 | 25m03s | 141.57 | 54.4308 | 618240 |
| | | AIME24 | 32.22 | 10m17s | 333.68 | 162.2038 | 610667 |
| | | GPQA | 24.33 | 34m54s | 314.88 | 156.5370 | 1942820 |
| | FP8 | GSM8k | 83.00 | 1m03s | 774.97 | 2.0614 | 129654 |
| | | MATH500 | 63.00 | 7m38s | 455.87 | 17.6713 | 24375 |
| | | AIME24 | 40.00 | 3m42s | 880.38 | 61.7136 | 582089 |
| | | GPQA | 27.67 | 12m40s | 856.35 | 57.6786 | 1915156 |
| | L4 | GSM8k | 80.33 | 1m24s | 562.97 | 3.6269 | 123986 |
| | | MATH500 | 63.67 | 13m01s | 264.82 | 29.4854 | 597768 |
| | | AIME24 | 36.67 | 6m34s | 499.62 | 98.4701 | 586687 |
| | | GPQA | 31.33 | 23m07s | 461.52 | 94.4508 | 1879225 |

## J.2   KV CACHE QUANTIZATION

Full results of KV Cache quantization with different models are listed in Table 13, 14.

We conducted additional experiments to further analyze the phenomenon. Specifically, we evaluated models at 1.5B, 3B, 4B, and 8B scales under with and without KV cache quantization—across RLLMs and LLMs, in order to determine whether model scale influences this metric. Interestingly, performance degradation was observed only in models based on Qwen-2.5-Math. We hypothesize that this issue arises from incompatibility between Qwen-2.5-Math and the current vLLM implementation of KV cache quantization.

Full results w/ & w/o KV Cache quantization for different small scale LLMs and RLLMs are listed in Table 15, 16, 17, 18, 19, 20, 21, 22.

Table 12: Results of RLLM-14B with Different Quantization Methods

| Model | Method | Dataset | Accuracy | Running Time | TPS | TTFT | Output Tokens |
|---|---|---|---|---|---|---|---|
| | | | | Budget-4096 | | | |
| RLLM-14B | GPTQ | GSM8k | 87.67 | 1m55s | 531.24 | 5.2007 | 167860 |
| | | MATH500 | 61.00 | 5m56s | 535.47 | 18.478 | 547482 |
| | | AIME24 | 22.22 | 2m25s | 811.91 | 49.699 | 346219 |
| | | GPQA | 16.33 | 8m09s | 775.78 | 40.463 | 1102007 |
| | AWQ | GSM8k | 86.67 | 1m30s | 617.47 | 4.6092 | 151523 |
| | | MATH500 | 61.00 | 6m01s | 534.74 | 19.725 | 551164 |
| | | AIME24 | 21.11 | 2m30s | 771.16 | 49.545 | 342303 |
| | | GPQA | 18.33 | 8m26s | 750.95 | 42.983 | 1106144 |
| | FP8 | GSM8k | 89.00 | 2m19s | 446.80 | 6.5690 | 170719 |
| | | MATH500 | 60.67 | 7m16s | 447.51 | 24.214 | 560177 |
| | | AIME24 | 24.44 | 2m55s | 665.67 | 62.497 | 343575 |
| | | GPQA | 14.33 | 9m49s | 650.24 | 51.583 | 1113927 |
| | L4 | GSM8k | 83.67 | 3m27s | 323.02 | 9.9777 | 187673 |
| | | MATH500 | 58.67 | 9m24s | 326.66 | 30.626 | 528533 |
| | | AIME24 | 26.67 | 3m41s | 525.41 | 74.833 | 341339 |
| | | GPQA | 16.33 | 12m44s | 501.28 | 65.669 | 1113251 |
| | | | | Budget-8192 | | | |
| RLLM-14B | GPTQ | GSM8k | 84.67 | 1m48s | 569.94 | 5.2287 | 168441 |
| | | MATH500 | 65.33 | 8m49s | 418.08 | 22.0500 | 638458 |
| | | AIME24 | 40.00 | 5m03s | 666.62 | 76.884 | 596659 |
| | | GPQA | 25.00 | 16m14s | 638.98 | 74.9600 | 1831969 |
| | AWQ | GSM8k | 86.67 | 1m48s | 585.55 | 4.7892 | 155465 |
| | | MATH500 | 65.33 | 9m22s | 399.52 | 23.658 | 647840 |
| | | AIME24 | 47.78 | 5m04s | 640.79 | 84.494 | 575835 |
| | | GPQA | 26.33 | 17m02s | 621.02 | 77.938 | 1866185 |
| | FP8 | GSM8k | 86.00 | 2m14s | 467.49 | 6.5689 | 171349 |
| | | MATH500 | 63.33 | 11m40s | 328.62 | 28.621 | 667889 |
| | | AIME24 | 51.11 | 5m43s | 535.96 | 90.392 | 544733 |
| | | GPQA | 27.33 | 19m27s | 529.1 | 84.797 | 1817404 |
| | L4 | GSM8k | 83.33 | 2m47s | 376.01 | 9.3994 | 172748 |
| | | MATH500 | 63.00 | 14m44s | 248.57 | 36.372 | 635662 |
| | | AIME24 | 47.78 | 7m58s | 401.73 | 123.95 | 572091 |
| | | GPQA | 21.33 | 28m55s | 368.01 | 115.88 | 1879105 |

Table 13: Results of RLLM-7B with Different KV Cache Quantization Methods

| Model | Method | Dataset | Acc. | Running Time | TPS | TTFT | Output Tokens |
|---|---|---|---|---|---|---|---|
| | | Budget-4096 | | | | | |
| RLLM-7B | FP8-E4M3 | GSM8k | 9.33 | 7m5s | 678.93 | 3.8153 | 843936 |
| | | MATH500 | 4.33 | 7m26s | 849.23 | 17.1335 | 1113415 |
| | | AIME24 | 0.00 | 2m13s | 936.07 | 23.4623 | 732374 |
| | | GPQA | 0.33 | 16m27s | 831.28 | 73.5032 | 2428386 |
| | FP8-E5M2 | GSM8k | 2.67 | 9m8s | 624.13 | 6.5660 | 1013649 |
| | | MATH500 | 0.33 | 9m28s | 731.99 | 22.0233 | 1223241 |
| | | AIME24 | 0.00 | 2m50s | 733.58 | 71.2345 | 367838 |
| | | GPQA | 0.00 | 9m36s | 739.58 | 36.1577 | 1225641 |
| | | Budget-8192 | | | | | |
| RLLM-7B | FP8-E4M3 | GSM8k | 8.67 | 14m23s | 634.50 | 5.0929 | 1631364 |
| | | MATH500 | 4.00 | 16m01s | 796.17 | 36.4224 | 2271199 |
| | | AIME24 | 0.00 | 4m18s | 855.67 | 50.5708 | 732374 |
| | | GPQA | 0.67 | 7m39s | 914.28 | 33.3658 | 1225852 |
| | FP8-E5M2 | GSM8k | 2.33 | 19m04s | 563.56 | 26.7276 | 1919839 |
| | | MATH500 | 0.33 | 20m02s | 681.93 | 94.8742 | 2438321 |
| | | AIME24 | 0.00 | 5m59s | 682.40 | 91.3258 | 730016 |
| | | GPQA | 0.00 | 20m21s | 678.99 | 102.0499 | 2450781 |

Table 14: Results of RLLM-14B with Different KV Cache Quantization Methods

| Model | Method | Dataset | Acc. | Running Time | TPS | TTFT | Output Tokens |
|---|---|---|---|---|---|---|---|
| | | Budget-4096 | | | | | |
| RLLM-14B | FP8-E4M3 | GSM8k | 87.33 | 2m17s | 466.06 | 7.5905 | 169107 |
| | | MATH500 | 63.00 | 7m37s | 417.60 | 25.1716 | 547992 |
| | | AIME24 | 26.67 | 2m59s | 640.63 | 63.6228 | 338613 |
| | | GPQA | 15.00 | 10m03s | 631.30 | 50.1774 | 1106576 |
| | FP8-E5M2 | GSM8k | 82.33 | 2m40s | 390.53 | 7.6730 | 171445 |
| | | MATH500 | 59.33 | 7m46s | 394.45 | 25.4353 | 527039 |
| | | AIME24 | 26.67 | 3m02s | 632.26 | 62.9118 | 339940 |
| | | GPQA | 15.67 | 10m19s | 608.30 | 49.3198 | 1094547 |
| | | Budget-8192 | | | | | |
| RLLM-14B | FP8-E4M3 | GSM8k | 83.67 | 3m03s | 357.72 | 7.7833 | 180474 |
| | | MATH500 | 66.33 | 12m18s | 305.61 | 30.8091 | 653017 |
| | | AIME24 | 52.22 | 5m53s | 520.69 | 94.5251 | 545376 |
| | | GPQA | 24.00 | 20m23s | 511.37 | 89.8416 | 1838882 |
| | FP8-E5M2 | GSM8k | 85.33 | 2m39s | 401.41 | 7.8384 | 175761 |
| | | MATH500 | 62.67 | 12m48s | 289.84 | 29.0644 | 633207 |
| | | AIME24 | 48.89 | 6m04s | 513.09 | 92.6786 | 553741 |
| | | GPQA | 26.67 | 20m51s | 497.93 | 90.6068 | 1833865 |

Table 15: Results of LLM-1.5B w/ & w/o KV Cache Quantization Method

| Model | Method | Dataset | Accuracy | E2E Time | TPS | TTFT | Output Tokens |
|---|---|---|---|---|---|---|---|
| LLM-1.5B | / | GSM8K | 43 | 229.2 | 983.57 | 0.0464 | 208813 |
| | | MATH-500 | 2.33 | 269.58 | 1362.62 | 0.0383 | 342955 |
| | | AIME-2024 | 3.33 | 77.88 | 1612.82 | 0.0397 | 118053 |
| | | GPQA | 4 | 271.69 | 1469.04 | 0.0543 | 363076 |
| | KV-Quant | GSM8K | 11.3 | 271.26 | 1451.46 | 0.0765 | 377097 |
| | | MATH-500 | 2 | 274.17 | 1585.54 | 0.0862 | 410326 |
| | | AIME-2024 | 0 | 82.58 | 1961.33 | 0.0892 | 154408 |
| | | GPQA | 4.33 | 281.86 | 1585.92 | 0.1199 | 410969 |

Table 16: Results of RLLM-1.5B w/ & w/o KV Cache Quantization Method

| Model | Method | Dataset | Accuracy | E2E Time | TPS | TTFVT | Output Tokens |
|---|---|---|---|---|---|---|---|
| RLLM-1.5B | / | GSM8K | 78.67 | 81.8 | 1746.7 | 0.93 | 126351 |
| | | MATH-500 | 34.67 | 543.2 | 1151.5 | 7.66 | 601202 |
| | | AIME-2024 | 10 | 171.4 | 2100.8 | 16.3 | 352712 |
| | | GPQA | 7 | 602.5 | 1972.3 | 15 | 1152348 |
| | KV-Quant | GSM8K | 1.33 | 645.27 | 1636.84 | 1.77 | 1039590 |
| | | MATH-500 | 1.33 | 683.57 | 1793.17 | 19.5 | 1201382 |
| | | AIME-2024 | 0 | 203.76 | 1845.88 | 1.49 | 368566 |
| | | GPQA | 0.33 | 684.12 | 1823.2 | 15.96 | 1211232 |

Table 17: Results of LLM-3B w/ & w/o KV Cache Quantization Method

| Model | Method | Dataset | Accuracy | E2E Time | TPS | TTFT | Output Tokens |
|---|---|---|---|---|---|---|---|
| LLM-3B | / | GSM8K | 71.33 | 96.09 | 735.03 | 0.1336 | 54009 |
| | | MATH-500 | 42.33 | 403.87 | 450.49 | 0.1572 | 157568 |
| | | AIME-2024 | 2.22 | 243.3 | 626.8 | 0.1746 | 144940 |
| | | GPQA | 25 | 594.2 | 518.77 | 0.3193 | 272210 |
| | KV-Quant | GSM8K | 66.67 | 96.61 | 813.14 | 0.2074 | 53805 |
| | | MATH-500 | 41.67 | 362.41 | 471.08 | 0.2505 | 146349 |
| | | AIME-2024 | 2.22 | 205.04 | 746.91 | 0.2651 | 145582 |
| | | GPQA | 25.67 | 420.76 | 590.33 | 0.395 | 212342 |

Table 18: Results of RLLM-3B w/ & w/o KV Cache Quantization Method

| Model | Method | Dataset | Accuracy | E2E Time | TPS | TTFT | Output Tokens |
|---|---|---|---|---|---|---|---|
| RLLM-3B | / | GSM8K | 70.33 | 174.99 | 646.19 | 0.1236 | 96453 |
| | | MATH-500 | 35 | 493.92 | 494.95 | 0.1565 | 220089 |
| | | AIME-2024 | 4.44 | 237.26 | 627.77 | 0.1762 | 141384 |
| | | GPQA | 11.67 | 729.24 | 640.77 | 0.3535 | 431229 |
| | KV-Quant | GSM8K | 70.33 | 113.69 | 886.52 | 0.162 | 84167 |
| | | MATH-500 | 35.33 | 529.98 | 485.45 | 0.2157 | 232903 |
| | | AIME-2024 | 3.33 | 209.76 | 771.79 | 0.2341 | 154332 |
| | | GPQA | 12 | 632.22 | 731.24 | 0.3161 | 426256 |

Table 19: Results of LLM-4B w/ & w/o KV Cache Quantization Method

| Model | Method | Dataset | Accuracy | E2E Time | TPS | TTFT | Output Tokens |
|---|---|---|---|---|---|---|---|
| LLM-4B | / | GSM8K | 58.33 | 327.2 | 329.3 | 0.0849 | 91124 |
| | | MATH-500 | 20.67 | 825.03 | 444.95 | 0.0863 | 342727 |
| | | AIME-2024 | 0 | 291.15 | 780.29 | 0.1031 | 219620 |
| | | GPQA | 12.33 | 596.03 | 403.09 | 0.1792 | 204211 |
| | KV-Quant | GSM8K | 57 | 120.82 | 665.7 | 0.1722 | 63809 |
| | | MATH-500 | 22.33 | 639.76 | 504.21 | 0.2299 | 298200 |
| | | AIME-2024 | 0 | 253.91 | 799.2 | 0.2492 | 195362 |
| | | GPQA | 12 | 554.87 | 449.49 | 0.4071 | 213366 |

Table 20: Results of RLLM-4B w/ & w/o KV Cache Quantization Method

| Model | Method | Dataset | Accuracy | E2E Time | TPS | TTFT | Output Tokens |
|---|---|---|---|---|---|---|---|
| RLLM-4B | / | GSM8K | 87 | 606.39 | 448.15 | 0.1517 | 255127 |
| | | MATH-500 | 37 | 1143.54 | 513 | 0.3683 | 562267 |
| | | AIME-2024 | 23.33 | 493.63 | 729.9 | 0.1963 | 352738 |
| | | GPQA | 6.67 | 1655.89 | 706.87 | 0.648 | 1134449 |
| | KV-Quant | GSM8K | 84.05 | 617.3 | 489.17 | 0.1948 | 285334 |
| | | MATH-500 | 44.67 | 916.27 | 654.39 | 0.2614 | 575223 |
| | | AIME-2024 | 18.89 | 352.53 | 1026.07 | 0.2811 | 351464 |
| | | GPQA | 8 | 1164.06 | 990.93 | 0.4428 | 1117451 |

Table 21: Results of LLM-8B w/ & w/o KV Cache Quantization Method

| Model | Method | Dataset | Accuracy | E2E Time | TPS | TTFT | Output Tokens |
|---|---|---|---|---|---|---|---|
| LLM-8B | / | GSM8K | 10.67 | 1132.57 | 384.2 | 0.1439 | 418505 |
| | | MATH-500 | 1.67 | 986.09 | 260.4 | 0.1214 | 232406 |
| | | AIME-2024 | 0 | 338.52 | 435.69 | 0.133 | 139929 |
| | | GPQA | 16.94 | 590.85 | 228.47 | 0.2394 | 98948 |
| | KV-Quant | GSM8K | 9 | 1312.4 | 330.81 | 0.4438 | 417527 |
| | | MATH-500 | 3 | 1083.71 | 249.87 | 0.7026 | 246408 |
| | | AIME-2024 | 0 | 369.53 | 310.85 | 0.7762 | 107308 |
| | | GPQA | 12.67 | 726.95 | 200.91 | 1.3001 | 110003 |

Table 22: Results of RLLM-8B w/ & w/o KV Cache Quantization Method

| Model | Method | Dataset | Accuracy | E2E Time | TPS | TTFVT | Output Tokens |
|---|---|---|---|---|---|---|---|
| RLLM-8B | / | GSM8K | 71.67 | 312.38 | 486.23 | 4.1972 | 135267 |
| | | MATH-500 | 46 | 1475.65 | 430.24 | 26.6753 | 610506 |
| | | AIME-2024 | 22.22 | 565.46 | 621.69 | 63.2468 | 343983 |
| | | GPQA | 10.33 | 1904.99 | 610.99 | 50.1914 | 1127883 |
| | KV-Quant | GSM8K | 71.33 | 294.56 | 523.36 | 3.3801 | 136574 |
| | | MATH-500 | 45.33 | 1230.57 | 521.34 | 21.381 | 617166 |
| | | AIME-2024 | 20 | 425.85 | 821.24 | 46.3206 | 342163 |
| | | GPQA | 8.33 | 1467.07 | 803.51 | 42.3785 | 1142755 |

## J.3 PREFIX CACHING

Full results of prefix cache with different models are listed in Table 23, 24, 25, 26.

Table 23: Results of RLLM-7B without Prefix Cache

| Model | Dataset | Acc. | Running Time | TPS | TTFT | Output Tokens |
|---|---|---|---|---|---|---|
| | | | Budget-4096 | | | |
| RLLM-7B | GSM8k | 79.00 | 1m26s | 564.90 | 3.5931 | 130372 |
| | MATH500 | 60.67 | 6m54s | 420.14 | 22.4207 | 498018 |
| | AIME24 | 18.89 | 2m46s | 708.96 | 58.7487 | 349035 |
| | GPQA | 14.67 | 9m24s | 686.08 | 47.7385 | 1124613 |
| | | | Budget-8192 | | | |
| RLLM-7B | GSM8k | 81.33 | 1m14s | 622.53 | 3.4274 | 124776 |
| | MATH500 | 60.33 | 11m02s | 335.91 | 26.4863 | 639141 |
| | AIME24 | 38.89 | 5m26s | 599.82 | 96.2117 | 588055 |
| | GPQA | 27.33 | 18m42s | 583.69 | 89.0785 | 1932823 |

Table 24: Results of RLLM-14B without Prefix Cache

| Model | Dataset | Acc. | Running Time | TPS | TTFT | Output Tokens |
|---|---|---|---|---|---|---|
| | | | Budget-4096 | | | |
| RLLM-14B | GSM8k | 88.00 | 2m31s | 420.23 | 8.2604 | 178175 |
| | MATH500 | 57.67 | 8m17s | 404.26 | 28.1407 | 576686 |
| | AIME24 | 23.33 | 3m12s | 604.58 | 69.2730 | 342347 |
| | GPQA | 13.67 | 10m52s | 587.18 | 56.3880 | 1114920 |
| | | | Budget-8192 | | | |
| RLLM-14B | GSM8k | 87.67 | 2m34s | 412.78 | 8.0207 | 171501 |
| | MATH500 | 62.33 | 12m31s | 292.60 | 31.9481 | 655351 |
| | AIME24 | 48.89 | 6m22s | 502.43 | 107.6285 | 569489 |
| | GPQA | 23.33 | 21m20s | 481.04 | 91.3230 | 1817908 |

Table 25: Results of RLLM-32B without Prefix Cache

| Model | Dataset | Acc. | Running Time | TPS | TTFT | Output Tokens |
|---|---|---|---|---|---|---|
| | | | Budget-4096 | | | |
| RLLM-32B | GSM8k | 92.67 | 2m30s | 327.11 | 5.8756 | 129761 |
| | MATH500 | 61.00 | 15m30s | 196.65 | 44.8259 | 529743 |
| | AIME24 | 25.56 | 5m09s | 379.99 | 113.5958 | 344566 |
| | GPQA | 19.00 | 17m45s | 349.79 | 83.0689 | 1082837 |
| | | | Budget-8192 | | | |
| RLLM-32B | GSM8k | 92.33 | 2m32s | 314.79 | 5.8947 | 130188 |
| | MATH500 | 70.00 | 23m14s | 152.31 | 58.6775 | 615489 |
| | AIME24 | 56.67 | 10m24s | 302.55 | 174.7540 | 559504 |
| | GPQA | 27.67 | 36m01s | 275.68 | 139.8732 | 1752688 |

To gain deeper insight, we evaluated more small models (8B and below) and observed consistent throughput degradation and increased E2E latency for both small LLMs and RLLMs when PC is enabled. In small models, the prefill stage is relatively lightweight, making its computational cost marginal. In addition, our evaluation datasets consist of single-turn conversations, where prefix reuse

Table 26: Results of RLLM-70B without Prefix Cache

| Model | Dataset | Acc. | Running Time | TPS | TTFT | Output Tokens |
|---|---|---|---|---|---|---|
| | | | Budget-4096 | | | |
| RLLM-70B | GSM8k | 90.00 | 3m25s | 228.12 | 8.2671 | 123955 |
| | MATH500 | 54.00 | 20m22s | 153.16 | 53.0075 | 535884 |
| | AIME24 | 31.11 | 7m05s | 269.74 | 136.8779 | 336872 |
| | GPQA | 19.33 | 24m06s | 251.73 | 110.7695 | 1055915 |
| | | | Budget-8192 | | | |
| RLLM-70B | GSM8k | 90.00 | 3m39s | 218.50 | 8.3275 | 126163 |
| | MATH500 | 61.33 | 31m57s | 115.36 | 73.7041 | 634981 |
| | AIME24 | 54.44 | 13m52s | 221.07 | 224.2071 | 544412 |
| | GPQA | 31.33 | 46m37s | 200.15 | 177.5976 | 1644534 |

opportunities are minimal. As a result, PC provides little practical benefit. Instead, the additional system-level overhead introduced by PC, such as hashing, block lookup, metadata management, and cache maintenance, can dominate the execution time. These factors cause the overhead to outweigh the potential savings from skipping prefill, ultimately leading to reduced throughput and increased E2E latency.

Full results of RLLMs and LLMs w/ & w/o prefix cache are listed in Table 27, 28, 29, 30, 31, 32, 33, 34.

Table 27: Results of LLM-1.5B w/ & w/o Prefix Cache

| Model | Method | Dataset | Accuracy | E2E Time | TPS | TTFT | Output Tokens |
|---|---|---|---|---|---|---|---|
| LLM-1.5B | w/ Prefix | GSM8K | 43 | 229.2 | 983.57 | 0.0464 | 208813 |
| | | MATH-500 | 2.33 | 269.58 | 1362.62 | 0.0383 | 342955 |
| | | AIME-2024 | 3.33 | 77.88 | 1612.82 | 0.0397 | 118053 |
| | | GPQA | 4 | 271.69 | 1469.04 | 0.0543 | 363076 |
| | w/o Prefix | GSM8K | 34.67 | 239.65 | 1005.01 | 0.0578 | 224224 |
| | | MATH-500 | 5.67 | 268.4 | 1321.34 | 0.058 | 330269 |
| | | AIME-2024 | 0 | 77.77 | 1714.1 | 0.0607 | 125741 |
| | | GPQA | 5.67 | 264.41 | 1450.04 | 0.0829 | 347353 |

Table 28: Results of RLLM-1.5B w/ & w/o Prefix Cache

| Model | Method | Dataset | Accuracy | E2E Time | TPS | TTFVT | Output Tokens |
|---|---|---|---|---|---|---|---|
| RLLM-1.5B | w/ Prefix | GSM8K | 78.67 | 81.8 | 1746.7 | 0.93 | 126351 |
| | | MATH-500 | 34.67 | 543.2 | 1151.5 | 7.66 | 601202 |
| | | AIME-2024 | 10 | 171.4 | 2100.8 | 16.3 | 352712 |
| | | GPQA | 7 | 602.5 | 1972.3 | 15 | 1152348 |
| | w/o Prefix | GSM8K | 75.33 | 84.51 | 1715.38 | 0.9577 | 128352 |
| | | MATH-500 | 42 | 541.61 | 1130.27 | 8.009 | 587791 |
| | | AIME-2024 | 16.67 | 170.91 | 2096.83 | 19.2384 | 350815 |
| | | GPQA | 5.67 | 600.57 | 1974.81 | 13.645 | 1149970 |

## J.4 SPECULATIVE DECODING

The visualization for 7B model SD is in Figure 6. Full results of speculative decoding evaluation with different models are listed in this subsection. For RLLM, results are presneted in Table 35, 36, 37, 38.

For LLM, results are presented in Table 39 (7B), 40 (32B).

Table 29: Results of LLM-3B w/ & w/o Prefix Cache

| Model | Method | Dataset | Accuracy | E2E Time | TPS | TTFT | Output Tokens |
|---|---|---|---|---|---|---|---|
| LLM-3B | w/ Prefix | GSM8K | 71.33 | 96.09 | 735.03 | 0.1336 | 54009 |
| | | MATH-500 | 42.33 | 403.87 | 450.49 | 0.1572 | 157568 |
| | | AIME-2024 | 2.22 | 243.3 | 626.8 | 0.1746 | 144940 |
| | | GPQA | 25 | 594.2 | 518.77 | 0.3193 | 272210 |
| | w/o Prefix | GSM8K | 68.67 | 86.37 | 823.13 | 0.1674 | 54468 |
| | | MATH-500 | 44 | 338.06 | 528.22 | 0.2129 | 154197 |
| | | AIME-2024 | 6.67 | 188.33 | 717.12 | 0.2192 | 127496 |
| | | GPQA | 26 | 480.12 | 556.83 | 0.3029 | 231298 |

Table 30: Results of RLLM-3B w/ & w/o Prefix Cache

| Model | Method | Dataset | Accuracy | E2E Time | TPS | TTFT | Output Tokens |
|---|---|---|---|---|---|---|---|
| RLLM-3B | w/ Prefix | GSM8K | 70.33 | 174.99 | 646.19 | 0.1236 | 96453 |
| | | MATH-500 | 35 | 493.92 | 494.95 | 0.1565 | 220089 |
| | | AIME-2024 | 4.44 | 237.26 | 627.77 | 0.1762 | 141384 |
| | | GPQA | 11.67 | 729.24 | 640.77 | 0.3535 | 431229 |
| | w/o Prefix | GSM8K | 74.67 | 172.62 | 646.13 | 0.1317 | 94910 |
| | | MATH-500 | 34.67 | 450.28 | 509.68 | 0.1981 | 205122 |
| | | AIME-2024 | 6.67 | 203.48 | 825.89 | 0.2102 | 160493 |
| | | GPQA | 7.33 | 601.27 | 736.82 | 0.2872 | 406986 |

Table 31: Results of LLM-4B w/ & w/o Prefix Cache

| Model | Method | Dataset | Accuracy | E2E Time | TPS | TTFT | Output Tokens |
|---|---|---|---|---|---|---|---|
| LLM-4B | w/ Prefix | GSM8K | 58.33 | 327.2 | 329.3 | 0.0849 | 91124 |
| | | MATH-500 | 20.67 | 825.03 | 444.95 | 0.0863 | 342727 |
| | | AIME-2024 | 0 | 291.15 | 780.29 | 0.1031 | 219620 |
| | | GPQA | 12.33 | 596.03 | 403.09 | 0.1792 | 204211 |
| | w/o Prefix | GSM8K | 57.67 | 184 | 450.33 | 0.301 | 66238 |
| | | MATH-500 | 22.33 | 787.06 | 428.68 | 0.481 | 313024 |
| | | AIME-2024 | 1.11 | 339.31 | 508.5 | 0.5265 | 164978 |
| | | GPQA | 14 | 627.28 | 379.1 | 0.7312 | 201756 |

Table 32: Results of RLLM-4B w/ & w/o Prefix Cache

| Model | Method | Dataset | Accuracy | E2E Time | TPS | TTFT | Output Tokens |
|---|---|---|---|---|---|---|---|
| RLLM-4B | w/ Prefix | GSM8K | 87 | 606.39 | 448.15 | 0.1517 | 255127 |
| | | MATH-500 | 37 | 1143.54 | 513 | 0.3683 | 562267 |
| | | AIME-2024 | 23.33 | 493.63 | 729.9 | 0.1963 | 352738 |
| | | GPQA | 6.67 | 1655.89 | 706.87 | 0.648 | 1134449 |
| | w/o Prefix | GSM8K | 84.33 | 643.99 | 437.34 | 0.3544 | 265022 |
| | | MATH-500 | 39 | 1158.26 | 502.03 | 0.5133 | 557106 |
| | | AIME-2024 | 26.67 | 496.64 | 716.82 | 0.5513 | 348441 |
| | | GPQA | 8.33 | 1667.94 | 699.65 | 0.7679 | 1130926 |

Table 33: Results of LLM-8B w/ & w/o Prefix Cache

| Model | Method | Dataset | Accuracy | E2E Time | TPS | TTFT | Output Tokens |
|---|---|---|---|---|---|---|---|
| LLM-8B | w/ Prefix | GSM8K | 10.67 | 1132.57 | 384.2 | 0.1439 | 418505 |
| | | MATH-500 | 1.67 | 986.09 | 260.4 | 0.1214 | 232406 |
| | | AIME-2024 | 0 | 338.52 | 435.69 | 0.133 | 139929 |
| | | GPQA | 16.94 | 590.85 | 228.47 | 0.2394 | 98948 |
| | w/o Prefix | GSM8K | 11 | 1342.13 | 344.13 | 0.4413 | 445235 |
| | | MATH-500 | 1.67 | 1035.8 | 227.53 | 0.6856 | 211306 |
| | | AIME-2024 | 1.11 | 366.74 | 302.76 | 0.7516 | 103473 |
| | | GPQA | 12.62 | 657.74 | 198.44 | 1.051 | 94475 |

Table 34: Results of RLLM-8B w/ & w/o Prefix Cache

| Model | Method | Dataset | Accuracy | E2E Time | TPS | TTFVT | Output Tokens |
|---|---|---|---|---|---|---|---|
| RLLM-8B | w/ Prefix | GSM8K | 71.67 | 312.38 | 486.23 | 4.1972 | 135267 |
| | | MATH-500 | 46 | 1475.65 | 430.24 | 26.6753 | 610506 |
| | | AIME-2024 | 22.22 | 565.46 | 621.69 | 63.2468 | 343983 |
| | | GPQA | 10.33 | 1904.99 | 610.99 | 50.1914 | 1127883 |
| | w/o Prefix | GSM8K | 71.33 | 264.27 | 572.27 | 3.2143 | 134612 |
| | | MATH-500 | 48 | 1195.59 | 527.13 | 20.2698 | 605859 |
| | | AIME-2024 | 13.33 | 428.79 | 847.52 | 53.3377 | 355845 |
| | | GPQA | 8 | 1455.82 | 807.04 | 40.7557 | 1138860 |

Table 35: Results of RLLM-7B with Different Speculative Decoding Methods

| Model | Budget | Dataset | Acc. | Running Time | TPS | TTFT | Output Tokens |
|---|---|---|---|---|---|---|---|
| | | | | L-Step: 2 | | | |
| RLLM-7B | 4096 | GSM8k | 83.33 | 1m21s | 411.47 | 3.8268 | 84044 |
| | | MATH500 | 63.67 | 8m26s | 241.67 | 28.9989 | 342214 |
| | | AIME24 | 18.89 | 4m52s | 266.43 | 91.1752 | 226340 |
| | | GPQA | 14.33 | 16m04s | 261.39 | 73.0445 | 720304 |
| | 8192 | GSM8k | 82.67 | 1m21s | 412.96 | 3.7715 | 83798 |
| | | MATH500 | 61.33 | 13m50s | 176.53 | 34.7626 | 415561 |
| | | AIME24 | 37.78 | 11m13s | 192.58 | 156.9481 | 383225 |
| | | GPQA | 26.67 | 37m16s | 187.68 | 161.3714 | 1223433 |
| | | | | L-Step: 4 | | | |
| RLLM-7B | 4096 | GSM8k | 82.33 | 1m32s | 380.14 | 3.9015 | 83812 |
| | | MATH500 | 62.00 | 8m26s | 228.30 | 30.2225 | 325135 |
| | | AIME24 | 18.89 | 4m49s | 250.19 | 84.2456 | 213773 |
| | | GPQA | 15.00 | 16m54s | 240.27 | 86.0878 | 699813 |
| | 8192 | GSM8k | 83.67 | 1m17s | 418.62 | 4.0124 | 83857 |
| | | MATH500 | 65.00 | 13m05s | 178.65 | 36.4870 | 400672 |
| | | AIME24 | 37.78 | 11m13s | 180.60 | 151.0329 | 356258 |
| | | GPQA | 28.33 | 37m11s | 172.07 | 152.6671 | 1119974 |
| | | | | L-Step: 8 | | | |
| RLLM-7B | 4096 | GSM8k | 85.67 | 1m28s | 378.44 | 4.0057 | 84219 |
| | | MATH500 | 61.00 | 8m35s | 228.01 | 30.6229 | 328151 |
| | | AIME24 | 21.11 | 4m57s | 248.74 | 92.2684 | 215501 |
| | | GPQA | 14.67 | 16m25s | 244.13 | 78.9886 | 685904 |
| | 8192 | GSM8k | 82.33 | 1m26s | 388.10 | 4.0512 | 83643 |
| | | MATH500 | 60.33 | 13m15s | 174.36 | 36.0669 | 392872 |
| | | AIME24 | 38.89 | 10m36s | 184.35 | 146.6946 | 344680 |
| | | GPQA | 23.33 | 36m48s | 180.27 | 171.9770 | 1158190 |

Table 36: Results of RLLM-14B with Speculative Decoding Method

| Model | Budget | L-Step | Dataset | Accuracy | Running Time | TPS | TTFT | Output Tokens |
|---|---|---|---|---|---|---|---|---|
| RLLM-14B | 4096 | 4 | GSM8k | 88.00 | 3m3s | 238.27 | 11.1347 | 113595 |
| | | | MATH500 | 59.33 | 11m52s | 187.51 | 45.6259 | 373475 |
| | | | AIME24 | 24.44 | 6m09s | 199.02 | 126.3789 | 217784 |
| | | | GPQA | 15.67 | 20m42s | 196.47 | 105.7676 | 696919 |
| | 8192 | 4 | GSM8k | 85.00 | 2m47s | 255.10 | 11.5077 | 112893 |
| | | | MATH500 | 62.67 | 17m22s | 148.10 | 52.2801 | 433701 |
| | | | AIME24 | 52.22 | 12m17s | 151.99 | 194.4668 | 337745 |
| | | | GPQA | 23.33 | 43m14s | 146.90 | 182.6785 | 1099344 |

Table 37: Results of RLLM-32B with Speculative Decoding Method

| Model | Budget | L-Step | Dataset | Accuracy | Running Time | TPS | TTFT | Output Tokens |
|---|---|---|---|---|---|---|---|---|
| RLLM-32B | 4096 | 4 | GSM8k | 90.33 | 2m36s | 218.41 | 7.2955 | 84809 |
| | | | MATH500 | 62.33 | 15m45s | 125.10 | 57.7919 | 333363 |
| | | | AIME24 | 25.56 | 8m12s | 148.29 | 157.7257 | 211331 |
| | | | GPQA | 16.67 | 26m29s | 147.23 | 117.7640 | 665233 |
| | 8192 | 4 | GSM8k | 92.00 | 2m32s | 222.34 | 7.3433 | 84595 |
| | | | MATH500 | 68.67 | 26m6s | 84.75 | 77.4161 | 376796 |
| | | | AIME24 | 47.78 | 18m56s | 99.98 | 240.3189 | 331180 |
| | | | GPQA | 24.00 | 56m02s | 105.96 | 216.8525 | 1050802 |

Table 38: Results of RLLM-70B with Speculative Decoding Method

| Model | Budget | L-Step | Dataset | Accuracy | Running Time | TPS | TTFT | Output Tokens |
|---|---|---|---|---|---|---|---|---|
| RLLM-70B | 4096 | 4 | GSM8k | 88.67 | 3m30s | 157.68 | 10.0978 | 84118 |
| | | | MATH500 | 56.67 | 22m01s | 98.28 | 72.7944 | 367031 |
| | | | AIME24 | 32.22 | 10m25s | 121.87 | 195.0349 | 221532 |
| | | | GPQA | 23.33 | 33m40s | 117.38 | 159.7265 | 678418 |
| | 8192 | 4 | GSM8k | 90.33 | 3m33s | 158.49 | 10.2136 | 84802 |
| | | | MATH500 | 59.00 | 31m32s | 77.02 | 86.9171 | 413304 |
| | | | AIME24 | 51.11 | 21m15s | 94.05 | 298.7859 | 349228 |
| | | | GPQA | 34.33 | 66m23s | 90.79 | 278.9286 | 1049010 |

Table 39: Results of LLM-7B with Different Speculative Decoding Methods

| Model | Budget | L-Step | Dataset | Accuracy | Running Time | TPS | TTFT | Output Tokens |
|---|---|---|---|---|---|---|---|---|
| LLM-7B | 4096 | 2 | GSM8k | 68.67 | 2m07s | 262.59 | 0.2300 | 83610 |
| | | | MATH500 | 2.33 | 2m40s | 299.47 | 0.3243 | 119825 |
| | | | AIME24 | 15.56 | 1m10s | 321.47 | 0.3449 | 60913 |
| | | | GPQA | 6.33 | 2m29s | 282.74 | 0.5113 | 90423 |
| | | 4 | GSM8k | 66.82 | 1m47s | 304.82 | 0.2263 | 80208 |
| | | | MATH500 | 0.67 | 2m32s | 309.60 | 0.3160 | 115604 |
| | | | AIME24 | 18.89 | 1m02s | 342.77 | 0.3234 | 54837 |
| | | | GPQA | 3.67 | 2m24s | 297.60 | 0.5061 | 36045 |
| | | 8 | GSM8k | 69.33 | 1m59s | 268.42 | 0.2257 | 79618 |
| | | | MATH500 | 2.00 | 2m54s | 277.62 | 0.3213 | 121082 |
| | | | AIME24 | 21.11 | 1m06s | 313.01 | 0.3505 | 54877 |
| | | | GPQA | 3.00 | 2m27s | 275.03 | 0.5067 | 85956 |

Table 40: Results of LLM-32B with Speculative Decoding Method

| Model | Budget | L-Step | Dataset | Accuracy | Running Time | TPS | TTFT | Output Tokens |
|---|---|---|---|---|---|---|---|---|
| LLM-32B | 4096 | 4 | GSM8k | 59.67 | 4m07s | 104.92 | 0.3605 | 61457 |
| | | | MATH500 | 45.33 | 3m57s | 142.52 | 0.5117 | 77136 |
| | | | AIME24 | 6.67 | 1m19s | 170.42 | 0.5746 | 32802 |
| | | | GPQA | 23.67 | 3m12s | 199.91 | 0.9070 | 79295 |
| | 8192 | 4 | GSM8k | 63.00 | 3m33s | 108.95 | 0.3581 | 53146 |
| | | | MATH500 | 45.67 | 3m33s | 146.57 | 0.5238 | 69419 |
| | | | AIME24 | 3.33 | 1m14s | 182.46 | 0.5354 | 31939 |
| | | | GPQA | 24.67 | 3m04s | 204.56 | 0.8977 | 77214 |

## K  EXTENDED RESULTS FOR REAL WORLD BENCHMARKING

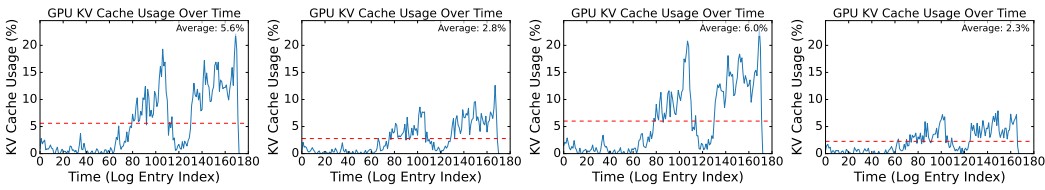

Figure 12: KV cache usage of 7B models under real-world workload across different datasets.

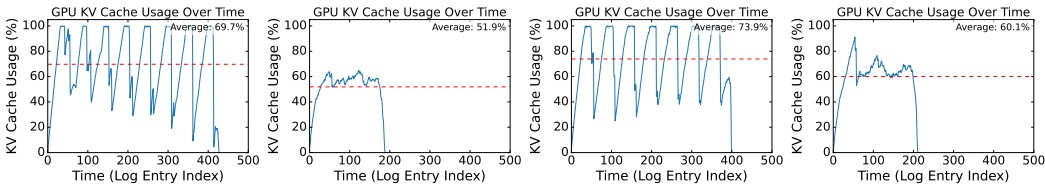

Figure 13: KV cache usage of 32B models under real-world workload across different datasets.

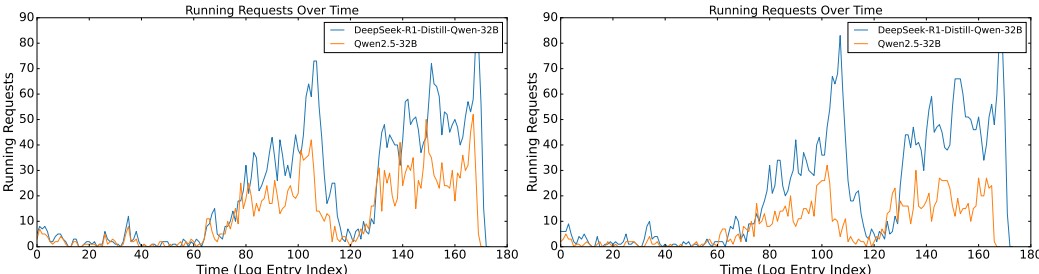

Figure 14: Num of running requests in the inference engine for 7B models under real-world workload.

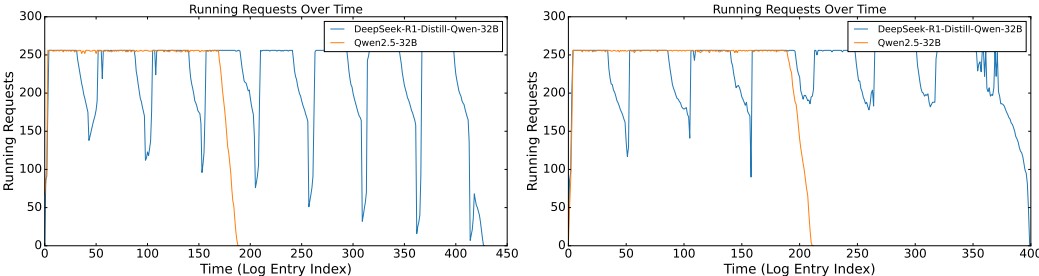

Figure 15: Num of running requests in the inference engine for 32B models under real-world workload.

