# OpenReview forum: "Reasoning Language Model Inference Serving Unveiled: An Empirical Study"
_ICLR.cc/2026/Conference — ICLR 2026 Poster_

### Official Review · Reviewer_ewEa · 2025-10-27

**Soundness:** 2
**Presentation:** 3
**Contribution:** 2
**Rating:** 4
**Confidence:** 3

**Summary:**

This paper presents an in-depth empirical study of inference serving for Reasoning Large Language Models, i.e. language models augmented or fine-tuned for complex multi-step reasoning. The authors introduce a new evaluation framework called ASU and a benchmarking suite named ASU-Perf for systematically measuring RLLM serving performance. Using these tools, the paper compares the serving behavior of RLLMs to that of standard LLMs across multiple model scales and tasks. The study finds distinct differences in how RLLMs perform under inference workloads, notably: (1) substantially higher and more volatile memory usage due to long reasoning chains  (2) the presence of “straggler” requests that take significantly longer than others in batched processing (3) an adaptive running time phenomenon where RLLMs spend more time on harder queries (4) a domain-specific performance gap, with RLLMs markedly outperforming same-size LLMs on math reasoning tasks but only matching their performance on knowledge-intensive queries. In addition to characterizing these behaviors, the paper evaluates common LLM serving optimizations on RLLMs. It reports that techniques like model quantization and speculative decoding can significantly improve throughput and latency for RLLM inference with only minimal accuracy loss. In contrast, methods such as prefix caching and KV-cache quantization do not consistently help RLLMs and can even degrade performance or accuracy for smaller models. Finally, the authors simulate a real-world workload to validate their findings under realistic conditions. The results confirm that RLLMs exhibit distinct serving behavior compared to normal LLMs even with irregular, bursty traffic, and the observed performance characteristics and trade-offs remain consistent. Overall, the paper’s contributions include exposing the unique challenges of serving reasoning-oriented LMs and providing insights and tooling to guide future research and engineering in efficient RLLM deployment.

**Strengths:**

Novel Problem: The paper tackles how to efficiently serve reasoning-augmented LLMs, which has not been systematically studied before. As RLLMs are increasingly relevant for complex tasks, understanding their inference behavior has significant practical and scientific value. This novelty in focusing on RLLM serving performance makes the contribution unique and valuable.

Introduction of Benchmarking Tools: Beyond experiments, the paper provides concrete tools to the community. The ASU framework unifies how to assess an LLM service from multiple perspectives rather than just single metrics. Along with this, the ASU-Perf benchmark suite is introduced for evaluating RLLM serving performance in a standardized way. These contributions are likely to be useful for researchers and practitioners working on LLM infrastructure, as they offer a way to consistently measure improvements and compare approaches on RLLM workloads.

Clear Identification of Key Findings: The paper does a good job of distilling its empirical observations into a set of clear findings, which have practical implications. For example, it highlights that RLLM inference can cause extreme memory fluctuation due to long chain-of-thought. This insight warns that existing serving systems might need to be adapted to avoid OOM issues when deploying RLLMs. Another useful finding is the “straggler request” problem: when processing a batch of queries, if one query requires an especially long reasoning chain , it will significantly lag behind others and occupy the resources, reducing overall throughput while it finishes. The study’s visualization of long-tail latency distribution and identification of this bottleneck is valuable for anyone designing scheduling or batching algorithms. The authors also notice an “overthinking” phenomenon that beyond a certain token budget, adding more chain-of-thought steps can hurt performance on some datasets. This is an intriguing insight that RLLMs might sometimes generate excessively long reasoning that isn’t beneficial, indicating a need for controlling reasoning length.

**Weaknesses:**

Certain Findings Are Expected: A few of the observed differences between RLLMs and standard LLMs are intuitive given the nature of chain-of-thought reasoning. For instance, the fact that generating a lengthy reasoning chain consumes more tokens and thus more memory and time is not surprising. RLLMs, by design, use more tokens to arrive at an answer, so higher memory usage and occasional long latencies are to be expected. The paper strengthens these points with data, which is good; however, a skeptical view is that some results confirm known intuitions rather than reveal completely unforeseen phenomena. This could be critiqued for reinforcing obvious points although it does add value by measuring the extent. A stronger theoretical or analytical exploration of why these phenomena occur would further strengthen the contribution.

Lacks guiding solutions: The paper does an excellent job in identifying and describing problems, but is deficient in proposing or evaluating solutions. Regarding the straggler request issue, the authors provide a clear diagnosis but do not explore potential mitigation strategies, such as preemption, gang scheduling, or other advanced scheduling techniques known in the system literature. This limits the constructive contribution of the paper.

The analysis is superficial: the paper reports a surprising finding that certain optimizations reduce the performance of 7B RLLMs. However, the analysis stops here. The paper should, at the very least, propose a plausible hypothesis for such an interesting and unexpected result. Is it because small models have a weaker ability to handle quantization noise in long reasoning chains? Or is it due to some architectural issue? The lack of deeper exploration is a missed opportunity.

**Questions:**

The paper identifies straggler requests as a key issue in RLLM services. Given this finding, have the authors considered or experimented with any scheduling strategies other than the default schedulers in vLLM/SGLang, such as preemption of long-running requests or adopting an approximate shortest job first strategy, to mitigate this problem?

The author found that certain optimizations can reduce the performance of 7B models. Could you provide a specific hypothesis to explain this phenomenon?

The proposed ASU framework is an excellent conceptual tool. Could you elaborate on the specific metrics selected for the server-side and user-side components in your ASU-Perf suite, and explain why they are most critical for the dedicated evaluation of RLLM?

---

> ### Author Response · Authors · 2025-11-25
> **Author Rebuttal  Part (1/8)**
>
> Dear Reviewer ewEa:
>
> We would like to express our deep appreciation to you for the exceptionally thorough, expert, and constructive evaluation of our work. Your detailed analysis and well-informed comments demonstrate a high level of technical insight, and we sincerely value the time and effort you invested. We have carefully examined every point you raised, and we address each of them with the utmost seriousness and clarity in the responses below.
>
> ----
>
> > **Q1**:  Certain Findings Are Expected: A few of the observed differences between RLLMs and standard LLMs are intuitive given the nature of chain-of-thought reasoning. For instance, the fact that generating a lengthy reasoning chain consumes more tokens and thus more memory and time is not surprising. RLLMs, by design, use more tokens to arrive at an answer, so higher memory usage and occasional long latencies are to be expected. The paper strengthens these points with data, which is good; however, a skeptical view is that some results confirm known intuitions rather than reveal completely unforeseen phenomena. This could be critiqued for reinforcing obvious points although it does add value by measuring the extent. A stronger theoretical or analytical exploration of why these phenomena occur would further strengthen the contribution.
>
> Thank you for the insightful comments. We fully agree that some behavioral characteristics of RLLMs—such as increased token usage, higher memory consumption, and longer latency due to extended chain-of-thought (CoT) generation—are indeed intuitive.
>
> However, we believe the core contribution of this paper lies in two aspects:
>
> 1.  Moving beyond qualitative intuition to provide, to the best of our knowledge, the first systematic characterization and analysis of the serving behavior of RLLMs within inference engines.
> 2.  Discussing potential techniques for efficient RLLM serving under resource constraints, offering practical guidance for both the research community and engineering practice.
>
> More specifically, our analysis does not stop at surface-level observations, but further investigates the underlying mechanisms:
>
> 1.  **Mechanism of memory fluctuation and high memory usage.**
>     We show that the pronounced memory oscillation fluctuations mainly arises from the interaction between long RLLM reasoning chains and the memory-management strategies of inference engines. Unlike standard LLMs, which exhibit relatively smooth memory usage patterns, RLLMs release large segments of KV cache abruptly at the end of a CoT reasoning, creating “pulse-like” memory pressure that challenges existing allocation strategies.
> 2.  **Explanation of the straggler effect (in latency / throughput).**
>     Because RLLM reasoning lengths vary significantly, the hardest queries in a batch become slowest stragglers. By quantifying task difficulty via inference-step length, we explain why batch latency in RLLM settings is dominated by these hardest cases, which in turn leads to substantial GPU idle time toward the end of each batch and motivates the design of improved scheduling algorithms.
> 3.  **Counterintuitive observations about optimization techniques (e.g., prefix caching).**\
>     We observe that certain optimizations designed for standard LLMs, such as prefix caching, are not always effective on smaller RLLMs (e.g., 7B). Our analysis suggests that this is because RLLMs often have short prompts but long, low-reuse CoT sequences, so the overhead of cache lookup can offset or outweigh the benefits of cache hits.
>
> In summary, although some of our findings are consistent with common intuition, our systematic analysis reveals **distinct serving behavior patterns of RLLMs** that are not yet well understood in the community. These insights provide a foundational basis for designing inference engines with more efficient memory management and scheduling strategies for RLLM workloads.

---

> ### Author Response · Authors · 2025-11-25
> **Author Rebuttal Part (2/8)**
>
> > **Q2**:  Lacks guiding solutions: The paper does an excellent job in identifying and describing problems, but is deficient in proposing or evaluating solutions. Regarding the straggler request issue, the authors provide a clear diagnosis but do not explore potential mitigation strategies, such as preemption, gang scheduling, or other advanced scheduling techniques known in the system literature. This limits the constructive contribution of the paper.
>
> Thank you for the valuable feedback. We agree that, beyond identifying the unique serving challenges introduced by RLLMs, discussing potential mitigation strategies can further deepen the community’s understanding of these phenomena. Below, we clarify: (i) the scope and primary contribution of this work, and (ii) several system-level optimization may be inspired by our observations.
>
> **Scope of the current work.**\
> This paper focuses on analyzing *how* RLLMs behave under the current inference engines (e.g., vLLM v1), which are tailored for LLM, and *how* their behaviors differ from standard LLMs. Our experiments reveal that, due to the lack of RLLM-specific optimizations in existing engines, long reasoning chains lead to pronounced tail latency and straggler effects. Making these issues explicit is one of the key messages we aim to deliver. As the reviewer points out, designing specialized scheduling or preemption mechanisms is indeed an important and natural next step, and we are actively exploring these directions in follow-up work.
>
> **Potential system-level optimization directions.**\
> Based on RLLM-specific characteristics and the behaviors observed in our study, we believe several classes of techniques merit systematic investigation:
>
> *   **Recompute vs. Swap–based preemption.**\
>     Current engines typically rely on recompute-based preemption to avoid explicit memory-management overhead. However, under RLLM workloads with much longer reasoning chains, re-computation becomes a larger portion of total latency. This suggests that *selective KV-cache swapping *may be a promising alternative for RLLM-oriented inference.
> *   **Length-aware scheduling (e.g., approximate SJF).**\
>     Building on prior LLM serving work, we believe that leveraging early signals—such as logits from the first generated token—to predict the eventual reasoning length may be enable approximate Shortest-Job-First or related policies, potentially reducing overall turnaround time and alleviating queue congestion.
> *   **Token-budget–based preemption and multi-level queues.**\
>     Treating “generated token count” as an analogue to CPU time slices enables token-budget–based preemption. Incorporating token counts as aging/degradation signals in multi-level feedback queues (MLFQ) could help maintain responsiveness for short queries while gently throttling extremely long reasoning chains.
> *   **Long-running request isolation in multi-instance deployment.**\
>     In multi-instance settings, isolating unusually long-running requests into dedicated instances or resource pools—using lightweight redirection—may reduce interference with short requests and improve overall quality of service.
>
> We hope that these directions provide initial inspiration for future research efforts on efficient RLLM serving.

---

> ### Author Response · Authors · 2025-11-25
> **Author Rebuttal Part (3/8)**
>
> ### **Q3 (Part 1/5)**
>
> > Q3: The analysis is superficial: the paper reports a surprising finding that certain optimizations reduce the performance of 7B RLLMs. However, the analysis stops here. The paper should, at the very least, propose a plausible hypothesis for such an interesting and unexpected result. Is it because small models have a weaker ability to handle quantization noise in long reasoning chains? Or is it due to some architectural issue? The lack of deeper exploration is a missed opportunity.
>
>
>
> Thank you for raising this important point. We agree that providing a deeper analysis would strengthen the paper, and following the reviewer’s suggestion, we conducted additional controlled experiments across a broader range of model sizes and architectures (listed in Table-1) to understand why small RLLMs respond poorly to KV-cache quantization and prefix caching.
>
> | RLLM  Type |              RLLM Name              | LLM Type |          LLM name          |
> | :-------------: | :---------------------------------------: | :------------: | :------------------------------: |
> |     1B RLLM     | deepseek-ai/DeepSeek-R1-Distill-Qwen-1.5B |     1B LLM     |      Qwen/Qwen2.5-Math-1.5B      |
> |     3B RLLM     |  suayptalha/DeepSeek-R1-Distill-Llama-3B  |     3B LLM     | meta-llama/Llama-3.2-3B-Instruct |
> |     4B RLLM     |       microsoft/Phi-4-mini-reasoning      |     4B LLM     |   microsoft/Phi-4-mini-instruct  |
> |     8B RLLM     |  deepseek-ai/DeepSeek-R1-Distill-Llama-8B |     8B LLM     |      meta-llama/Llama-3.1-8B     |
>
> Table-1 Employed models in our rebuttal
>
> ----
>
> Our expanded analysis leads to several plausible and experimentally supported hypotheses:
>
> ### **1. Why KV-Cache Quantization Harms Small RLLMs ？**
>
> In Observation 5.2, we reported that enabling KV-cache quantization causes drastic accuracy drops for small RLLMs (e.g., 7B), whereas large models remain stable. Our new experiments across 1.5B, 3B, 4B, and 8B models reveal the following insights （Table 2 - Table 9）:
>
> #### **(a) High sensitivity of small models to quantization noise**
>
> Across all tested RLLM and LLM, larger models demonstrate robust performance under KV-cache quantization, but smaller models show substantial variance or collapse.
>
> We hypothesize that this stems from small models having:
>
> *   Lower parameter redundancy,
> *   Higher reliance on precise attention scores, and
> *   Weaker ability to tolerate perturbations in multi-step CoT reasoning.
>
> This makes their reasoning chains particularly susceptible to small errors introduced by low-precision KV states.
>
> #### **(b) Architecture–engine compatibility issues**
>
> Performance collapse is highly **architecture-dependent**:
>
> *   Failures are concentrated in **Qwen-2.5-Math** 1.5B and 7B models.
> *   Other small models (e.g., 3B, 4B) remain relatively stable after quantization.
>
> This suggests that part of the degradation arises from **incomplete model support** for specific architectures in the current serving engine (e.g. vLLM), rather than from model scale.
>
> #### **(c) Quantization appears to change reasoning behavior**
>
> We also observed that KV-cache quantization substantially changes the **number of outputed tokens** (i.e., reasoning chain length) for 1B/3B/4B RLLM models (also 4B LLM). This indicates that quantization noise may disrupt the model’s **step-by-step reasoning trajectory**, which aligns with the hypothesis that long CoT requires high-precision attention dynamics.
>
> #### **(d) Small RLLMs are affected far more than small LLMs**
>
> Interestingly, the accuracy degradation is significantly larger for \*\*small RLLMs \*\*than for **small non-reasoning LLMs**, suggesting that long CoT reasoning is inherently more sensitive to KV-cache perturbations.
>
>
> **Summary:**  Our deeper analysis indicates that the severe degradation observed in small RLLMs under KV-cache quantization arises from the combination of three factors: 1) their low parameter redundancy makes them highly sensitive to quantization noise, 2) their long chain-of-thought reasoning requires precise attention dynamics, and 3) architecture–kernel mismatches in the current vLLM implementation further amplify these vulnerabilities. Therefore,  we explicitly recommend applying KV-cache quantization to small RLLMs only with great caution.
>
> ----
> **The Table-2 to Table-9 is in the following part.**
>
>
> ----

---

> ### Author Response · Authors · 2025-11-25
> **Author Rebuttal Part(4/8)**
>
> ### **Q3 Part (2/5)**
> ----
>
>
> |   Model   |  Method  |  Dataset  | Accuracy | E2E Time |   TPS   | TTFVT | Output Tokens |
> | :-------: | :------: | :-------: | :------: | :------: | :-----: | :---: | :-----------: |
> | 1.5B RLLM |     /    |   GSM8K   |   78.67  |   81.8   |  1746.7 |  0.93 |     126351    |
> | 1.5B RLLM |     /    |  MATH-500 |   34.67  |   543.2  |  1151.5 |  7.66 |     601202    |
> | 1.5B RLLM |     /    | AIME-2024 |   10.00  |   171.4  |  2100.8 |  16.3 |     352712    |
> | 1.5B RLLM |     /    |    GPQA   |   7.00   |   602.5  |  1972.3 |  15.0 |    1152348    |
> | 1.5B RLLM | KV-Quant |   GSM8K   |   1.33   |  645.27  | 1636.84 |  1.77 |    1039590    |
> | 1.5B RLLM | KV-Quant |  MATH-500 |   1.33   |  683.57  | 1793.17 | 19.50 |    1201382    |
> | 1.5B RLLM | KV-Quant | AIME-2024 |   0.00   |  203.76  | 1845.88 |  1.49 |     368566    |
> | 1.5B RLLM | KV-Quant |    GPQA   |   0.33   |  684.12  | 1823.20 | 15.96 |    1211232    |
>
> **Table-2 1B RLLM KV Cache Quantization**
>
> |  Model  |  Method  |  Dataset  | Accuracy | E2E Time |   TPS  |  TTFT  | Output Tokens |
> | :-----: | :------: | :-------: | :------: | :------: | :----: | :----: | :-----------: |
> | 3B RLLM |     /    |   GSM8K   |   70.33  |  174.99  | 646.19 | 0.1236 |     96453     |
> | 3B RLLM |     /    |  MATH-500 |   35.00  |  493.92  | 494.95 | 0.1565 |     220089    |
> | 3B RLLM |     /    | AIME-2024 |   4.44   |  237.26  | 627.77 | 0.1762 |     141384    |
> | 3B RLLM |     /    |    GPQA   |   11.67  |  729.24  | 640.77 | 0.3535 |     431229    |
> | 3B RLLM | KV-Quant |   GSM8K   |   70.33  |  113.69  | 886.52 | 0.1620 |     84167     |
> | 3B RLLM | KV-Quant |  MATH-500 |   35.33  |  529.98  | 485.45 | 0.2157 |     232903    |
> | 3B RLLM | KV-Quant | AIME-2024 |   3.33   |  209.76  | 771.79 | 0.2341 |     154332    |
> | 3B RLLM | KV-Quant |    GPQA   |   12.00  |  632.22  | 731.24 | 0.3161 |     426256    |
>
> **Table-3 3B RLLM KV Cache Quantization**
>
> |  Model  |  Method  |  Dataset  | Accuracy | E2E Time |   TPS   |  TTFT  | Output Tokens |
> | :-----: | :------: | :-------: | :------: | :------: | :-----: | :----: | :-----------: |
> | 4B RLLM |     /    |   GSM8K   |   87.00  |  606.39  |  448.15 | 0.1517 |     255127    |
> | 4B RLLM |     /    |  MATH-500 |   37.00  |  1143.54 |  513.00 | 0.3683 |     562267    |
> | 4B RLLM |     /    | AIME-2024 |   23.33  |  493.63  |  729.90 | 0.1963 |     352738    |
> | 4B RLLM |     /    |    GPQA   |   6.67   |  1655.89 |  706.87 | 0.6480 |    1134449    |
> | 4B RLLM | KV-Quant |   GSM8K   |   84.05  |  617.30  |  489.17 | 0.1948 |     285334    |
> | 4B RLLM | KV-Quant |  MATH-500 |   44.67  |  916.27  |  654.39 | 0.2614 |     575223    |
> | 4B RLLM | KV-Quant | AIME-2024 |   18.89  |  352.53  | 1026.07 | 0.2811 |     351464    |
> | 4B RLLM | KV-Quant |    GPQA   |   8.00   |  1164.06 |  990.93 | 0.4428 |    1117451    |
>
> **Table-4 4B RLLM KV Cache Quantization**
>
> |  Model  |  Method  |  Dataset  | Accuracy | E2E Time |   TPS  |  TTFVT  | Output Tokens |
> | :-----: | :------: | :-------: | :------: | :------: | :----: | :-----: | :-----------: |
> | 8B RLLM |     /    |   GSM8K   |   71.67  |  312.38  | 486.23 |  4.1972 |     135267    |
> | 8B RLLM |     /    |  MATH-500 |   46.00  |  1475.65 | 430.24 | 26.6753 |     610506    |
> | 8B RLLM |     /    | AIME-2024 |   22.22  |  565.46  | 621.69 | 63.2468 |     343983    |
> | 8B RLLM |     /    |    GPQA   |   10.33  |  1904.99 | 610.99 | 50.1914 |    1127883    |
> | 8B RLLM | KV-Quant |   GSM8K   |   71.33  |  294.56  | 523.36 |  3.3801 |     136574    |
> | 8B RLLM | KV-Quant |  MATH-500 |   45.33  |  1230.57 | 521.34 | 21.3810 |     617166    |
> | 8B RLLM | KV-Quant | AIME-2024 |   20.00  |  425.85  | 821.24 | 46.3206 |     342163    |
> | 8B RLLM | KV-Quant |    GPQA   |   8.33   |  1467.07 | 803.51 | 42.3785 |    1142755    |
>
> **Table-5 8B RLLM KV Cache Quantization**
>
> |   Model  |  Method  |  Dataset  | Accuracy | E2E Time |   TPS   |  TTFT  | Output Tokens |
> | :------: | :------: | :-------: | :------: | :------: | :-----: | :----: | :-----------: |
> | 1.5B LLM |     /    |   GSM8K   |   43.00  |  229.20  |  983.57 | 0.0464 |     208813    |
> | 1.5B LLM |     /    |  MATH-500 |   2.33   |  269.58  | 1362.62 | 0.0383 |     342955    |
> | 1.5B LLM |     /    | AIME-2024 |   3.33   |   77.88  | 1612.82 | 0.0397 |     118053    |
> | 1.5B LLM |     /    |    GPQA   |   4.00   |  271.69  | 1469.04 | 0.0543 |     363076    |
> | 1.5B LLM | KV-Quant |   GSM8K   |   11.30  |  271.26  | 1451.46 | 0.0765 |     377097    |
> | 1.5B LLM | KV-Quant |  MATH-500 |   2.00   |  274.17  | 1585.54 | 0.0862 |     410326    |
> | 1.5B LLM | KV-Quant | AIME-2024 |   0.00   |   82.58  | 1961.33 | 0.0892 |     154408    |
> | 1.5B LLM | KV-Quant |    GPQA   |   4.33   |  281.86  | 1585.92 | 0.1199 |     410969    |
>
> **Table-6 1B LLM KV Cache Quantization**

---

> ### Author Response · Authors · 2025-11-25
> **Author Rebuttal Part(5/8)**
>
> ### **Q3 (Part 3/5)**
>
> ----
>
> |  Model |  Method  |  Dataset  | Accuracy | E2E Time |   TPS  |  TTFT  | Output Tokens |
> | :----: | :------: | :-------: | :------: | :------: | :----: | :----: | :-----------: |
> | 3B LLM |     /    |   GSM8K   |   71.33  |   96.09  | 735.03 | 0.1336 |     54009     |
> | 3B LLM |     /    |  MATH-500 |   42.33  |  403.87  | 450.49 | 0.1572 |     157568    |
> | 3B LLM |     /    | AIME-2024 |   2.22   |  243.30  | 626.80 | 0.1746 |     144940    |
> | 3B LLM |     /    |    GPQA   |   25.00  |  594.20  | 518.77 | 0.3193 |     272210    |
> | 3B LLM | KV-Quant |   GSM8K   |   66.67  |   96.61  | 813.14 | 0.2074 |     53805     |
> | 3B LLM | KV-Quant |  MATH-500 |   41.67  |  362.41  | 471.08 | 0.2505 |     146349    |
> | 3B LLM | KV-Quant | AIME-2024 |   2.22   |  205.04  | 746.91 | 0.2651 |     145582    |
> | 3B LLM | KV-Quant |    GPQA   |   25.67  |  420.76  | 590.33 | 0.3950 |     212342    |
>
> **Table-7 3B RLLM KV Cache Quantization**
>
> |  Model |  Method  |  Dataset  | Accuracy | E2E Time |   TPS  |  TTFT  | Output Tokens |
> | :----: | :------: | :-------: | :------: | :------: | :----: | :----: | :-----------: |
> | 4B LLM |     /    |   GSM8K   |   58.33  |  327.20  | 329.30 | 0.0849 |     91124     |
> | 4B LLM |     /    |  MATH-500 |   20.67  |  825.03  | 444.95 | 0.0863 |     342727    |
> | 4B LLM |     /    | AIME-2024 |   0.00   |  291.15  | 780.29 | 0.1031 |     219620    |
> | 4B LLM |     /    |    GPQA   |   12.33  |  596.03  | 403.09 | 0.1792 |     204211    |
> | 4B LLM | KV-Quant |   GSM8K   |   57.00  |  120.82  | 665.70 | 0.1722 |     63809     |
> | 4B LLM | KV-Quant |  MATH-500 |   22.33  |  639.76  | 504.21 | 0.2299 |     298200    |
> | 4B LLM | KV-Quant | AIME-2024 |   0.00   |  253.91  | 799.20 | 0.2492 |     195362    |
> | 4B LLM | KV-Quant |    GPQA   |   12.00  |  554.87  | 449.49 | 0.4071 |     213366    |
>
> **Table-8 4B RLLM KV Cache Quantization**
>
> |  Model |  Method  |  Dataset  | Accuracy | E2E Time |   TPS  |  TTFT  | Output Tokens |
> | :----: | :------: | :-------: | :------: | :------: | :----: | :----: | :-----------: |
> | 8B LLM |     /    |   GSM8K   |   10.67  |  1132.57 | 384.20 | 0.1439 |     418505    |
> | 8B LLM |     /    |  MATH-500 |   1.67   |  986.09  | 260.40 | 0.1214 |     232406    |
> | 8B LLM |     /    | AIME-2024 |   0.00   |  338.52  | 435.69 | 0.1330 |     139929    |
> | 8B LLM |     /    |    GPQA   |   16.94  |  590.85  | 228.47 | 0.2394 |     98948     |
> | 8B LLM | KV-Quant |   GSM8K   |   9.00   |  1312.40 | 330.81 | 0.4438 |     417527    |
> | 8B LLM | KV-Quant |  MATH-500 |   3.00   |  1083.71 | 249.87 | 0.7026 |     246408    |
> | 8B LLM | KV-Quant | AIME-2024 |   0.00   |  369.53  | 310.85 | 0.7762 |     107308    |
> | 8B LLM | KV-Quant |    GPQA   |   12.67  |  726.95  | 200.91 | 1.3001 |     110003    |
>
> **Table-9 8B RLLM KV Cache Quantization**
>
> ----
>
> ## **2. Why Prefix Cache (PC) Slows Down Small RLLMs ?**
>
> Our paper reports that prefix caching improves throughput for large RLLMs but unexpectedly degrades efficiency for small ones. We performed additional analysis and now provide a  system-level explanation based on the overhead–benefit trade-off.
>
> **1. Overhead outweighs benefits for small language models**
>
> Prefix caching introduces non-trivial runtime overhead, including:
> + Hashing and prefix matching
> + Cache block lookup
> + Metadata and memory management
>
> For large models, the prefill phase is computationally expensive. The savings from skipping prefill are much larger than these overheads, so PC provides net performance gains.
> For small models (e.g., 7B):
> The prefill phase is extremely lightweight (very few FLOPs).
> The PC overhead becomes comparable to—or even larger than—the actual prefill computation cost.
> As a result, the net effect is performance degradation.
>
> **2. RLLM workloads are decode-dominated (Long CoT)**
>
> A key property of RLLMs is their very long chain-of-thought (CoT) generation.
> RLLM inference time is dominated by the decode phase, which is memory-bound.
> Prefix caching only accelerates prefill, not decode.
> The decode bottleneck remains unchanged, while PC adds extra GPU-side memory and metadata operations.
> Thus, PC increases system complexity without accelerating the dominant computation stage.
>
> **3. Low cache hit rate in our workload**
>
> As noted in the experimental setup:
> Most benchmark queries are single-turn prompts.
> Prefix reuse is therefore low.
> In such low hit-rate scenarios, the cost of maintaining the cache exceeds the benefits.
>
> ### **Conclusion**
>
> Prefix caching helps large RLLMs because their prefill cost is large and benefits outweigh overhead. For small RLLMs, however, PC adds overhead but does not meaningfully reduce runtime, and may even worsen it due to increased memory-management complexity.
> Therefore, unless the workload has high prefix-reuse (e.g., multi-turn dialogs or shared/system prompts), we recommend disabling prefix caching for small RLLMs.
>
> ----
>
> **Table-10 to 17 are in the following.**
>
>
> ----

---

> ### Author Response · Authors · 2025-11-25
> **Author Rebuttal Part (6/8)**
>
> ### **Q3 Part (4/5)**
>
> ----
>
>
>
> |   Model   |   Method   |  Dataset  | Accuracy | E2E Time |   TPS   |  TTFVT  | Output Tokens |
> | :-------: | :--------: | :-------: | :------: | :------: | :-----: | :-----: | :-----------: |
> | 1.5B RLLM |      /     |   GSM8K   |   78.67  |   81.8   |  1746.7 |   0.93  |     126351    |
> | 1.5B RLLM |      /     |  MATH-500 |   34.67  |   543.2  |  1151.5 |   7.66  |     601202    |
> | 1.5B RLLM |      /     | AIME-2024 |   10.00  |   171.4  |  2100.8 |   16.3  |     352712    |
> | 1.5B RLLM |      /     |    GPQA   |   7.00   |   602.5  |  1972.3 |   15.0  |    1152348    |
> | 1.5B RLLM | w/o Prefix |   GSM8K   |   75.33  |   84.51  | 1715.38 |  0.9577 |     128352    |
> | 1.5B RLLM | w/o Prefix |  MATH-500 |   42.00  |  541.61  | 1130.27 |  8.0090 |     587791    |
> | 1.5B RLLM | w/o Prefix | AIME-2024 |   16.67  |  170.91  | 2096.83 | 19.2384 |     350815    |
> | 1.5B RLLM | w/o Prefix |    GPQA   |   5.67   |  600.57  | 1974.81 | 13.6450 |    1149970    |
>
> **Table-10 1B RLLM without prefix caching**
>
> |  Model  |   Method   |  Dataset  | Accuracy | E2E Time |   TPS  |  TTFT  | Output Tokens |
> | :-----: | :--------: | :-------: | :------: | :------: | :----: | :----: | :-----------: |
> | 3B RLLM |      /     |   GSM8K   |   70.33  |  174.99  | 646.19 | 0.1236 |     96453     |
> | 3B RLLM |      /     |  MATH-500 |   35.00  |  493.92  | 494.95 | 0.1565 |     220089    |
> | 3B RLLM |      /     | AIME-2024 |   4.44   |  237.26  | 627.77 | 0.1762 |     141384    |
> | 3B RLLM |      /     |    GPQA   |   11.67  |  729.24  | 640.77 | 0.3535 |     431229    |
> | 3B RLLM | w/o Prefix |   GSM8K   |   74.67  |  172.62  | 646.13 | 0.1317 |     94910     |
> | 3B RLLM | w/o Prefix |  MATH-500 |   34.67  |  450.28  | 509.68 | 0.1981 |     205122    |
> | 3B RLLM | w/o Prefix | AIME-2024 |   6.67   |  203.48  | 825.89 | 0.2102 |     160493    |
> | 3B RLLM | w/o Prefix |    GPQA   |   7.33   |  601.27  | 736.82 | 0.2872 |     406986    |
>
> **Table-11 3B RLLM without prefix caching**
>
> |  Model  |   Method   |  Dataset  | Accuracy | E2E Time |   TPS  |  TTFT  | Output Tokens |
> | :-----: | :--------: | :-------: | :------: | :------: | :----: | :----: | :-----------: |
> | 4B RLLM |      /     |   GSM8K   |   87.00  |  606.39  | 448.15 | 0.1517 |     255127    |
> | 4B RLLM |      /     |  MATH-500 |   37.00  |  1143.54 | 513.00 | 0.3683 |     562267    |
> | 4B RLLM |      /     | AIME-2024 |   23.33  |  493.63  | 729.90 | 0.1963 |     352738    |
> | 4B RLLM |      /     |    GPQA   |   6.67   |  1655.89 | 706.87 | 0.6480 |    1134449    |
> | 4B RLLM | w/o Prefix |   GSM8K   |   84.33  |  643.99  | 437.34 | 0.3544 |     265022    |
> | 4B RLLM | w/o Prefix |  MATH-500 |   39.00  |  1158.26 | 502.03 | 0.5133 |     557106    |
> | 4B RLLM | w/o Prefix | AIME-2024 |   26.67  |  496.64  | 716.82 | 0.5513 |     348441    |
> | 4B RLLM | w/o Prefix |    GPQA   |   8.33   |  1667.94 | 699.65 | 0.7679 |    1130926    |
>
> **Table-12 4B RLLM without prefix caching**
>
> |  Model  |   Method   |  Dataset  | Accuracy | E2E Time |   TPS  |  TTFVT  | Output Tokens |
> | :-----: | :--------: | :-------: | :------: | :------: | :----: | :-----: | :-----------: |
> | 8B RLLM |      /     |   GSM8K   |   71.67  |  312.38  | 486.23 |  4.1972 |     135267    |
> | 8B RLLM |      /     |  MATH-500 |   46.00  |  1475.65 | 430.24 | 26.6753 |     610506    |
> | 8B RLLM |      /     | AIME-2024 |   22.22  |  565.46  | 621.69 | 63.2468 |     343983    |
> | 8B RLLM |      /     |    GPQA   |   10.33  |  1904.99 | 610.99 | 50.1914 |    1127883    |
> | 8B RLLM | w/o Prefix |   GSM8K   |   71.33  |  264.27  | 572.27 |  3.2143 |     134612    |
> | 8B RLLM | w/o Prefix |  MATH-500 |   48.00  |  1195.59 | 527.13 | 20.2698 |     605859    |
> | 8B RLLM | w/o Prefix | AIME-2024 |   13.33  |  428.79  | 847.52 | 53.3377 |     355845    |
> | 8B RLLM | w/o Prefix |    GPQA   |   8.00   |  1455.82 | 807.04 | 40.7557 |    1138860    |
>
> **Table-13 8B RLLM without prefix caching**
>
> |   Model  |   Method   |  Dataset  | Accuracy | E2E Time |   TPS   |  TTFT  | Output Tokens |
> | :------: | :--------: | :-------: | :------: | :------: | :-----: | :----: | :-----------: |
> | 1.5B LLM |      /     |   GSM8K   |   43.00  |  229.20  |  983.57 | 0.0464 |     208813    |
> | 1.5B LLM |      /     |  MATH-500 |   2.33   |  269.58  | 1362.62 | 0.0383 |     342955    |
> | 1.5B LLM |      /     | AIME-2024 |   3.33   |   77.88  | 1612.82 | 0.0397 |     118053    |
> | 1.5B LLM |      /     |    GPQA   |   4.00   |  271.69  | 1469.04 | 0.0543 |     363076    |
> | 1.5B LLM | w/o Prefix |   GSM8K   |   34.67  |  239.65  | 1005.01 | 0.0578 |     224224    |
> | 1.5B LLM | w/o Prefix |  MATH-500 |   5.67   |  268.40  | 1321.34 | 0.0580 |     330269    |
> | 1.5B LLM | w/o Prefix | AIME-2024 |    0.0   |   77.77  | 1714.10 | 0.0607 |     125741    |
> | 1.5B LLM | w/o Prefix |    GPQA   |   5.67   |  264.41  | 1450.04 | 0.0829 |     347353    |
>
> **Table-14 1B LLM without prefix caching**

---

> ### Author Response · Authors · 2025-11-25
> **Author Rebuttal Part (7/8)**
>
> ### **Q3 Part (5/5)**
>
> |  Model |   Method   |  Dataset  | Accuracy | E2E Time |   TPS  |  TTFT  | Output Tokens |
> | :----: | :--------: | :-------: | :------: | :------: | :----: | :----: | :-----------: |
> | 3B LLM |      /     |   GSM8K   |   71.33  |   96.09  | 735.03 | 0.1336 |     54009     |
> | 3B LLM |      /     |  MATH-500 |   42.33  |  403.87  | 450.49 | 0.1572 |     157568    |
> | 3B LLM |      /     | AIME-2024 |   2.22   |  243.30  | 626.80 | 0.1746 |     144940    |
> | 3B LLM |      /     |    GPQA   |   25.00  |  594.20  | 518.77 | 0.3193 |     272210    |
> | 3B LLM | w/o Prefix |   GSM8K   |   68.67  |   86.37  | 823.13 | 0.1674 |     54468     |
> | 3B LLM | w/o Prefix |  MATH-500 |   44.00  |  338.06  | 528.22 | 0.2129 |     154197    |
> | 3B LLM | w/o Prefix | AIME-2024 |   6.67   |  188.33  | 717.12 | 0.2192 |     127496    |
> | 3B LLM | w/o Prefix |    GPQA   |   26.00  |  480.12  | 556.83 | 0.3029 |     231298    |
>
> **Table-15 3B LLM without prefix caching**
>
> |  Model |   Method   |  Dataset  | Accuracy | E2E Time |   TPS  |  TTFT  | Output Tokens |
> | :----: | :--------: | :-------: | :------: | :------: | :----: | :----: | :-----------: |
> | 4B LLM |      /     |   GSM8K   |   58.33  |  327.20  | 329.30 | 0.0849 |     91124     |
> | 4B LLM |      /     |  MATH-500 |   20.67  |  825.03  | 444.95 | 0.0863 |     342727    |
> | 4B LLM |      /     | AIME-2024 |   0.00   |  291.15  | 780.29 | 0.1031 |     219620    |
> | 4B LLM |      /     |    GPQA   |   12.33  |  596.03  | 403.09 | 0.1792 |     204211    |
> | 4B LLM | w/o Prefix |   GSM8K   |   57.67  |  184.00  | 450.33 | 0.3010 |     66238     |
> | 4B LLM | w/o Prefix |  MATH-500 |   22.33  |  787.06  | 428.68 | 0.4810 |     313024    |
> | 4B LLM | w/o Prefix | AIME-2024 |   1.11   |  339.31  | 508.50 | 0.5265 |     164978    |
> | 4B LLM | w/o Prefix |    GPQA   |   14.00  |  627.28  | 379.10 | 0.7312 |     201756    |
>
> **Table-16 4B LLM without prefix caching**
>
> |  Model |   Method   |  Dataset  | Accuracy | E2E Time |   TPS  |  TTFT  | Output Tokens |
> | :----: | :--------: | :-------: | :------: | :------: | :----: | :----: | :-----------: |
> | 8B LLM |      /     |   GSM8K   |   10.67  |  1132.57 | 384.20 | 0.1439 |     418505    |
> | 8B LLM |      /     |  MATH-500 |   1.67   |  986.09  | 260.40 | 0.1214 |     232406    |
> | 8B LLM |      /     | AIME-2024 |   0.00   |  338.52  | 435.69 | 0.1330 |     139929    |
> | 8B LLM |      /     |    GPQA   |   16.94  |  590.85  | 228.47 | 0.2394 |     98948     |
> | 8B LLM | w/o Prefix |   GSM8K   |   11.00  |  1342.13 | 344.13 | 0.4413 |     445235    |
> | 8B LLM | w/o Prefix |  MATH-500 |   1.67   |  1035.80 | 227.53 | 0.6856 |     211306    |
> | 8B LLM | w/o Prefix | AIME-2024 |   1.11   |  366.74  | 302.76 | 0.7516 |     103473    |
> | 8B LLM | w/o Prefix |    GPQA   |   12.62  |  657.74  | 198.44 | 1.0510 |     94475     |
>
> **Table-17 6B LLM without prefix caching**
>
> ----
>
> > **Q4:** The paper identifies straggler requests as a key issue in RLLM services. Given this finding, have the authors considered or experimented with any scheduling strategies other than the default schedulers in vLLM/SGLang, such as preemption of long-running requests or adopting an approximate shortest job first strategy, to mitigate this problem?
>
> See Q2.
>
> > **Q5:** Q5: The author found that certain optimizations can reduce the performance of 7B models. Could you provide a specific hypothesis to explain this phenomenon?
>
> See Q3.

---

> ### Author Response · Authors · 2025-11-25
> **Author Rebuttal Part (8/8)**
>
> > Q6: The proposed ASU framework is an excellent conceptual tool. Could you elaborate on the specific metrics selected for the server-side and user-side components in your ASU-Perf suite, and explain why they are most critical for the dedicated evaluation of RLLM?
>
> Thank you for the thoughtful question. In the ASU framework, our goal is to ensure that a model not only achieves sufficient task performance (**Accuracy**) but also satisfies the distinct needs of **service providers** and **end users**. Accordingly, **ASU-Perf** incorporates metrics across three dimensions—Accuracy, Service-end, and User-end—to capture the practical trade-offs of deploying RLLMs in real systems.
>
> ### 1. **Accuracy: The prerequisite for adoption RLLM models**
>
> As discussed in Section 3.2, for RLLMs, complex reasoning capabilities (e.g., mathematics and coding) are the primary value users look for. Thus, **Accuracy** is the most fundamental and irreplaceable metric in ASU. Users will only consider adopting or paying for an RLLM if its reasoning performance meets their expectations.
>
> ### 2. **Service-end Metrics: Resource efficiency for providers**
>
> Service providers aim to maximize throughput under constrained hardware resources. We therefore select two core metrics:
>
> *   **TPS (Tokens per Second)**\
>     TPS measures the overall processing capacity of the system. For RLLMs—whose extended chain-of-thought significantly increases the number of generated tokens—TPS directly reflects the computational load placed on the inference engine.
> *   **E2E Latency (End-to-End time)**\
>     This captures the total time from receiving a request to delivering the final answer, including startup, prefill, decode, and scheduling overheads. Since RLLM reasoning time varies widely with task difficulty (as shown in our “straggler” and “adaptive running time” analyses in §4), E2E latency is essential for reflecting system efficiency under realistic workloads.
>
> ### 3. **User-end Metrics: Reflecting real user experience**
>
> For end users, responsiveness and cost are the key aspects of perceived quality. ASU-Perf therefore includes:
>
> *   **TTFVT (Time to First Visible Token)**\
>     In commercial RLLMs (e.g., OpenAI o1), the internal chain-of-thought is hidden from users. Thus, we extend TTFT to **TTFVT**, which measures when the user first sees *visible* model output. While users may tolerate some “thinking time,” they primarily care about when the final output begins to appear, making TTFVT a more faithful measure of interactive experience.
> *   **Number of Output Tokens**\
>     Since RLLMs tend to produce long CoT traces and lengthy answers, output token count directly affects user cost (especially under token-based billing). Excessive token generation increases usage cost and may harm user retention, making this an important user-side metric.
> *   **E2E Completion Time**\
>     This measures the total time from request submission to receiving the full answer. Because RLLM runtime fluctuates significantly with task difficulty, E2E completion time best captures the user’s actual perception of responsiveness.
>
> ----
>
> We hope the above responses satisfactorily address your concerns, and we are happy to clarify any questions.

---

### Official Review · Reviewer_soWx · 2025-10-29

**Soundness:** 3
**Presentation:** 3
**Contribution:** 3
**Rating:** 6
**Confidence:** 3

**Summary:**

This paper presents an empirical study on the serving performance of reasoning large language models (RLLMs), highlighting how their serving behaviors differ from those of traditional LLMs. It also investigates inference optimization techniques, such as quantization and KV caching, and examines whether these methods provide measurable benefits when serving LLMs.

Their main contributions include:
1. ASU, a framework for assessing RLLM serving performance, along with ASU-Perf, its corresponding benchmark suite.
2. An empirical investigation into the key differences in serving behaviors between RLLMs and traditional LLMs.
3. An empirical study on the effectiveness of serving optimization techniques (e.g., quantization and KV caching) when applied to RLLMs.

**Strengths:**

1. The proposed benchmark and framework for evaluating RLLM serving are valuable contributions, especially as RLLMs become increasingly prevalent.
2. The empirical study is extensive, offering interesting observations on both serving behaviors and serving optimization techniques for RLLMs. These findings should be useful for researchers and practitioners aiming to improve systems for serving LLMs.
3 The serving performance of reasoning models remains under-explored, and this paper helps fill that gap.

**Weaknesses:**

This paper primarily presents empirical observations without offering much in-depth analysis. The authors treat the models largely as black boxes, running benchmark experiments and reporting results without providing deeper insights or interpretations.

Some of the evaluations, while interesting, have been explored in prior work, such as comparisons between RLLMs and LLMs that do not focus on the serving aspect. It would strengthen the paper to narrow the scope and focus more clearly on serving-related issues, which would make the key messages and contributions more distinct.

**Questions:**

1. In the experiments analyzing the impact of different batch sizes, could you clarify why batch size would affect model accuracy at all? Are there any interactions between computations across examples within a batch? While batch size certainly influences serving performance, it should not directly affect accuracy.
2. Could you provide an explanation for why AWQ and L4 could worsen end-to-end latency and throughput?
3. In Figure 5 (first subplot), how does the prefix cache affect accuracy? My understanding is that prefix caching should not influence the model’s output at all.

---

> ### Author Response · Authors · 2025-11-25
> **Author Rebuttal Part (1/4)**
>
> Reviewer soWx:
>
> We are grateful for your careful assessment and helpful critiques. We have reflected on all feedback and respond to each concern in detail in the sections that follow.
>
> ----
>
> > Q1: This paper primarily presents empirical observations without offering much in-depth analysis. The authors treat the models largely as black boxes, running benchmark experiments and reporting results without providing deeper insights or interpretations.
>
> Thank you for the insightful comment. We understand the concern that the paper may appear to focus on empirical observations without sufficiently deep analysis. While our work is positioned as the *first empirical study* on RLLM serving behavior—aimed at filling an existing gap—we emphasize that our contribution goes beyond presenting data: we also analyze the *system-level mechanisms* underlying the observed phenomena. We respond to your concern from three perspectives.
>
> ***
>
> ### **1. Methodological rationale in serving-related research**
>
> In inference serving research, treating the model as a **black-box workload** is a widely accepted methodology. It allows system researchers to identify bottlenecks—such as memory fragmentation or scheduling inefficiencies—*without modifying model internals*. Our goal is to inform system design (e.g., scheduling, memory management), not model architecture.
>
> Even so, we do not limit ourselves to black-box measurements. We further examine how **RLLM-specific properties**, especially long chain-of-thought (CoT) reasoning, fundamentally change load characteristics compared to standard LLMs.\
> For example, Section 4 shows that **RLLM runtime strongly correlates with problem difficulty**, unlike standard LLMs whose runtime mainly scales with input length. This is more than an empirical fact—it indicates that **future RLLM schedulers must incorporate difficulty-aware signals instead of relying solely on length predictors**, a system-level insight absent in previous work.
>
> ***
>
> ### **2. The causal mechanisms behind core observations**
>
> Several sections of the paper already provide **root-cause analysis**:
>
> *   **Memory fluctuation.**\
>     We show that extreme KV-cache oscillation (3% → 70%) is caused by the “generate-and-discard” nature of long CoT reasoning. This demonstrates that current memory managers, which retain many states irrelevant to future steps, are **poorly aligned with RLLM behavior**, motivating *reasoning-aware memory management* algorithms.
> *   **Straggler effects.**\
>     We reveal that tail latency is not random noise, but arises from large variance in the number of reasoning steps required for hard problems. This insight implies that **continuous batching must be redesigned** for workloads with extreme long-tail distributions.
>
> Thus, many of our observations are accompanied by explanations of *why* they arise and what they imply for serving systems.
>
> ***
>
> ### **3. Deeper explanations of optimization failures—and improvements in the revised version**
>
> We appreciate the reviewer’s suggestion to strengthen the interpretability of Section 5. In the revised version, we will expand our analysis to more explicitly link model behaviors with optimization outcomes.
>
> Examples include:
>
> *   **Why small models fail under KV-cache quantization.**\
>     Small RLLMs (e.g., 7B) lack parameter redundancy and rely heavily on precise attention scores to maintain long reasoning chains. Quantization noise disrupts these chains, causing performance collapse—while larger models (14B+) can tolerate such noise.
> *   **Why prefix caching hurts small RLLMs.**\
>     For small models, the cost of cache hashing, lookup, and metadata management can exceed the savings from skipping prefill.\
>     Additionally, RLLMs’ long CoT makes decode the dominant bottleneck—prefix caching does not optimize decode but increases memory pressure, sometimes worsening fragmentation or triggering more swaps.
>
> These analyses move beyond empirical results and provide **actionable interpretations** that motivate future system optimizations.

---

> ### Author Response · Authors · 2025-11-25
> **Author Rebuttal Part (2/4)**
>
> >  Q2: Some of the evaluations, while interesting, have been explored in prior work, such as comparisons between RLLMs and LLMs that do not focus on the serving aspect. It would strengthen the paper to narrow the scope and focus more clearly on serving-related issues, which would make the key messages and contributions more distinct.
>
> Thank you for the constructive suggestion. We fully agree that narrowing the scope and placing stronger emphasis on **serving-related issues** will make the paper’s contributions clearer and more impactful. Our goal is not to replicate existing comparisons between RLLMs and LLMs, but to reveal the *system-level implications* of deploying RLLMs in real inference environments.

---

> ### Author Response · Authors · 2025-11-25
> **Author Rebuttal Part (3/4)**
>
> > Q3: In the experiments analyzing the impact of different batch sizes, could you clarify why batch size would affect model accuracy at all? Are there any interactions between computations across examples within a batch? While batch size certainly influences serving performance, it should not directly affect accuracy.
>
> Thank you for the careful observation. We fully agree with your point: **in theory, batch size should not affect model accuracy**, as each sample in a batch is processed independently.
>
> To address the discrepancy seen in our reported results, we clarify the following:
>
> ### **1. No computational interaction within a batch**
>
> We have verified that the inference engine strictly isolates the computation of each sample within a batch. There is **no parameter sharing or cross-example interference**, so batch size has *no direct influence* on the correctness of the model outputs.
>
> ### **2. The minor accuracy fluctuations arise from non-deterministic decoding**
>
> The small differences in accuracy across batch sizes are caused by randomness in the decoding strategy—not by batching itself.
>
> To follow standard RLLM evaluation protocols (consistent with industrial practice \[1]\[2]), we use **non-greedy decoding** with:
>
> *   Temperature = 0.6
> *   Top-p = 0.95
> *   Top-k = 20
>
> This introduces **sampling noise**, meaning that different runs or different sample orderings can produce slightly different outputs.
>
> > Q4: Could you provide an explanation for why AWQ and L4 could worsen end-to-end latency and throughput?
>
> Thank you for raising this important technical question. While quantization *theoretically* improves performance by reducing memory bandwidth and compute cost, we indeed observed increased latency and reduced throughput for AWQ and L4. We attribute this to limitations in the current inference engine implementation, rather than the quantization algorithms themselves.
>
> ### **1. Kernel optimization differences**
>
> Our inspection of the vLLM codebase shows that:
>
> *   **GPTQ-INT4** benefits from a highly optimized kernel (`gptq_marlin`), which efficiently exploits low-bit computation.
> *   **AWQ and L4**, however, **lack similarly optimized kernels**, leading to suboptimal performance.
>
> Thus, GPTQ-INT4 enjoys mature engineering support, while AWQ/L4 do not—yet.
>
> ### **2. Overheads exceed theoretical gains**
>
> For AWQ and L4 in their current implementations:
>
> *   Additional overheads such as **dequantization**,
> *   Less efficient memory layouts, and
> *   Non-fused operations
>
> often outweigh the savings from reduced memory traffic.
>
> This is consistent with the runtime logs observed during experiments. In particular, we found explicit warnings in vLLM log of our experiments such as:
>
> ```markdown
> [config.py:662] awq quantization is not fully optimized yet.
> The speed can be slower than non-quantized models.
> ```
>
> ### **Conclusion**
>
> The performance regressions of AWQ and L4 arise from the **current inference engine eco-system**, not from inherent flaws in the quantization methods.

---

> ### Author Response · Authors · 2025-11-25
> **Author Rebuttal Part (4/4)**
>
> > Q5: In Figure 5 (first subplot), how does the prefix cache affect accuracy? My understanding is that prefix caching should not influence the model’s output at all.
>
>
> Thank you for pointing this out. We fully agree with your understanding: **prefix caching should not influence model accuracy**, since it only reuses previously computed KV states and does *not* alter the forward-pass computation. Under **greedy decoding**, the outputs would be identical.
>
> The small accuracy variations observed in Figure 5 are not caused by prefix caching. Instead, they arise from **stochastic sampling** in our evaluation setup:
>
> ### **1. Experimental decoding setup**
>
> To follow standard RLLM evaluation protocols (consistent with industrial practice \[1]\[2]), we use **non-greedy decoding** with:
>
> *   Temperature = 0.6
> *   Top-p = 0.95
> *   Top-k = 20
>
> This decoding strategy is **non-deterministic**, and thus inherently introduces randomness.
>
> ### **2. Statistical fluctuation in outputs**
>
> Because of this stochastic sampling:
>
> *   Different runs may produce slightly different answers during to the multiple reasoning path.
> *   Different batch compositions or execution orders can also lead to small output variations.
>
> This results in **minor statistical fluctuations** in accuracy, even when the underlying computation is unchanged.
>
> ### **Conclusion**
>
> The small differences seen in Figure 5 stem from **sampling randomness**, \*not \*from prefix caching affecting the model’s output. We will clarify this explicitly in the revised manuscript.
>
> ----
>
> We hope the above responses satisfactorily address your concerns, and we are happy to clarify any  questions.

---

### Official Review · Reviewer_d8ud · 2025-11-01

**Soundness:** 3
**Presentation:** 3
**Contribution:** 2
**Rating:** 4
**Confidence:** 4

**Summary:**

This paper presents an empirical study on Reasoning Large Language Model (RLLM) inference serving. The authors compare RLLMs (e.g., DeepSeek-R1-Distill-Qwen) with traditional LLMs under various workloads and inference engines (vLLM, SGLang). They identify distinct serving behaviors—such as high memory fluctuation, straggler requests, adaptive running time, and domain preference—and evaluate how existing serving optimizations (quantization, speculative decoding, prefix caching, KV cache quantization) affect RLLM serving efficiency. The paper further validates these findings under real-world workloads modeled by Gamma distribution.

**Strengths:**

Novel and relevant empirical perspective.
This is one of the first systematic studies on the serving characteristics of reasoning-oriented LLMs, which are becoming increasingly important in practice. The empirical exploration fills a gap between model-level reasoning research and system-level inference efficiency.

Comprehensive experimental coverage.
The paper evaluates multiple model scales (7B–70B), reasoning datasets (GSM8K, MATH500, AIME24, GPQA), and optimization techniques (quantization, speculative decoding, prefix caching). This breadth enhances the generalizability of observations.

Insightful findings for system designers.
The work identifies several counterintuitive behaviors including: (1) significant memory usage
and fluctuations; (2) straggler requests; (3) adaptive running time; (4) domain preference.

**Weaknesses:**

Lack of explanation for partial observations.
Several empirical findings are not sufficiently explained. For instance, the paper reports that prefix caching provides little or even negative benefit for 7B reasoning models, but no analysis is given on why this happens. Without such interpretation, the results remain descriptive rather than insightful.

Missing comparison with standard LLMs in Section 5.
Section 5 focuses on evaluating the effectiveness of several optimization techniques—such as quantization, speculative decoding, prefix caching, and KV-cache quantization—on reasoning LLMs. However, it does not include any baseline results for LLMs under the same experimental setup. Without this comparison, it is difficult to determine whether the reported behaviors (e.g., prefix caching being ineffective for 7B models, speculative decoding offering limited speedup, or quantization showing inconsistent gains) are unique challenges of reasoning models or common limitations of large language models in general.

Unclear implications for system optimization.
While the paper presents many interesting empirical patterns, it does not clearly explain how these observations could guide future optimization of RLLM inference systems. The paper stops at observation without turning the results into actionable guidance for improving RLLM

**Questions:**

Same as weakness

---

> ### Author Response · Authors · 2025-11-25
> **Author Rebuttal Part (1/8)**
>
> Dear Reviewer d8ud:
>
> We thank you for dedicating their time to review our submission and for providing meaningful and constructive comments. We take these suggestions seriously and address them comprehensively below.
>
> ----
>
> > Q1 : Lack of explanation for partial observations. Several empirical findings are not sufficiently explained. For instance, the paper reports that prefix caching provides little or even negative benefit for 7B reasoning models, but no analysis is given on why this happens. Without such interpretation, the results remain descriptive rather than insightful.
>
> Thank you for raising this important point. We agree that providing a deeper analysis would strengthen the paper, and following the reviewer’s suggestion, we conducted additional controlled experiments across a broader range of model sizes and architectures (listed in Table-1) to understand why small RLLMs respond poorly to KV-cache quantization and prefix caching.
>
> | RLLM  Type |              RLLM Name              | LLM Type |          LLM name          |
> | :-------------: | :---------------------------------------: | :------------: | :------------------------------: |
> |     1B RLLM     | deepseek-ai/DeepSeek-R1-Distill-Qwen-1.5B |     1B LLM     |      Qwen/Qwen2.5-Math-1.5B      |
> |     3B RLLM     |  suayptalha/DeepSeek-R1-Distill-Llama-3B  |     3B LLM     | meta-llama/Llama-3.2-3B-Instruct |
> |     4B RLLM     |       microsoft/Phi-4-mini-reasoning      |     4B LLM     |   microsoft/Phi-4-mini-instruct  |
> |     8B RLLM     |  deepseek-ai/DeepSeek-R1-Distill-Llama-8B |     8B LLM     |      meta-llama/Llama-3.1-8B     |
>
> Table-1 Employed models in our rebuttal
>
> ----
>
> Our expanded analysis leads to several plausible and experimentally supported hypotheses:
>
> ### **1. Why KV-Cache Quantization Harms Small RLLMs ？**
>
> In Observation 5.2, we reported that enabling KV-cache quantization causes drastic accuracy drops for small RLLMs (e.g., 7B), whereas large models remain stable. Our new experiments across 1.5B, 3B, 4B, and 8B models reveal the following insights （Table 2 - Table 9）:
>
> #### **(a) High sensitivity of small models to quantization noise**
>
> Across all tested RLLM and LLM, larger models demonstrate robust performance under KV-cache quantization, but smaller models show substantial variance or collapse.
>
> We hypothesize that this stems from small models having:
>
> *   Lower parameter redundancy,
> *   Higher reliance on precise attention scores, and
> *   Weaker ability to tolerate perturbations in multi-step CoT reasoning.
>
> This makes their reasoning chains particularly susceptible to small errors introduced by low-precision KV states.
>
> #### **(b) Architecture–engine compatibility issues**
>
> Performance collapse is highly **architecture-dependent**:
>
> *   Failures are concentrated in **Qwen-2.5-Math** 1.5B and 7B models.
> *   Other small models (e.g., 3B, 4B) remain relatively stable after quantization.
>
> This suggests that part of the degradation arises from **incomplete model support** for specific architectures in the current serving engine (e.g. vLLM), rather than from model scale.
>
> #### **(c) Quantization appears to change reasoning behavior**
>
> We also observed that KV-cache quantization substantially changes the **number of outputed tokens** (i.e., reasoning chain length) for 1B/3B/4B RLLM models (also 4B LLM). This indicates that quantization noise may disrupt the model’s ** step-by-step reasoning trajectory**, which aligns with the hypothesis that long CoT requires high-precision attention dynamics.
>
> #### **(d) Small RLLMs are affected far more than small LLMs**
>
> Interestingly, the accuracy degradation is significantly larger for \*\*small RLLMs \*\*than for **small non-reasoning LLMs**, suggesting that long CoT reasoning is inherently more sensitive to KV-cache perturbations.
>
>
> **Summary:**  Our deeper analysis indicates that the severe degradation observed in small RLLMs under KV-cache quantization arises from the combination of three factors: 1) their low parameter redundancy makes them highly sensitive to quantization noise, 2) their long chain-of-thought reasoning requires precise attention dynamics, and 3) architecture–kernel mismatches in the current vLLM implementation further amplify these vulnerabilities. Therefore,  we explicitly recommend applying KV-cache quantization to small RLLMs only with great caution.
>
> ----
> **The Table-2 to Table-9 is in the following part.**
>
>
> ----

---

> ### Author Response · Authors · 2025-11-25
> **Author Rebuttal Part (2/8)**
>
> |   Model   |  Method  |  Dataset  | Accuracy | E2E Time |   TPS   | TTFVT | Output Tokens |
> | :-------: | :------: | :-------: | :------: | :------: | :-----: | :---: | :-----------: |
> | 1.5B RLLM |     /    |   GSM8K   |   78.67  |   81.8   |  1746.7 |  0.93 |     126351    |
> | 1.5B RLLM |     /    |  MATH-500 |   34.67  |   543.2  |  1151.5 |  7.66 |     601202    |
> | 1.5B RLLM |     /    | AIME-2024 |   10.00  |   171.4  |  2100.8 |  16.3 |     352712    |
> | 1.5B RLLM |     /    |    GPQA   |   7.00   |   602.5  |  1972.3 |  15.0 |    1152348    |
> | 1.5B RLLM | KV-Quant |   GSM8K   |   1.33   |  645.27  | 1636.84 |  1.77 |    1039590    |
> | 1.5B RLLM | KV-Quant |  MATH-500 |   1.33   |  683.57  | 1793.17 | 19.50 |    1201382    |
> | 1.5B RLLM | KV-Quant | AIME-2024 |   0.00   |  203.76  | 1845.88 |  1.49 |     368566    |
> | 1.5B RLLM | KV-Quant |    GPQA   |   0.33   |  684.12  | 1823.20 | 15.96 |    1211232    |
>
> **Table-2 1B RLLM KV Cache Quantization**
>
> |  Model  |  Method  |  Dataset  | Accuracy | E2E Time |   TPS  |  TTFT  | Output Tokens |
> | :-----: | :------: | :-------: | :------: | :------: | :----: | :----: | :-----------: |
> | 3B RLLM |     /    |   GSM8K   |   70.33  |  174.99  | 646.19 | 0.1236 |     96453     |
> | 3B RLLM |     /    |  MATH-500 |   35.00  |  493.92  | 494.95 | 0.1565 |     220089    |
> | 3B RLLM |     /    | AIME-2024 |   4.44   |  237.26  | 627.77 | 0.1762 |     141384    |
> | 3B RLLM |     /    |    GPQA   |   11.67  |  729.24  | 640.77 | 0.3535 |     431229    |
> | 3B RLLM | KV-Quant |   GSM8K   |   70.33  |  113.69  | 886.52 | 0.1620 |     84167     |
> | 3B RLLM | KV-Quant |  MATH-500 |   35.33  |  529.98  | 485.45 | 0.2157 |     232903    |
> | 3B RLLM | KV-Quant | AIME-2024 |   3.33   |  209.76  | 771.79 | 0.2341 |     154332    |
> | 3B RLLM | KV-Quant |    GPQA   |   12.00  |  632.22  | 731.24 | 0.3161 |     426256    |
>
> **Table-3 3B RLLM KV Cache Quantization**
>
> |  Model  |  Method  |  Dataset  | Accuracy | E2E Time |   TPS   |  TTFT  | Output Tokens |
> | :-----: | :------: | :-------: | :------: | :------: | :-----: | :----: | :-----------: |
> | 4B RLLM |     /    |   GSM8K   |   87.00  |  606.39  |  448.15 | 0.1517 |     255127    |
> | 4B RLLM |     /    |  MATH-500 |   37.00  |  1143.54 |  513.00 | 0.3683 |     562267    |
> | 4B RLLM |     /    | AIME-2024 |   23.33  |  493.63  |  729.90 | 0.1963 |     352738    |
> | 4B RLLM |     /    |    GPQA   |   6.67   |  1655.89 |  706.87 | 0.6480 |    1134449    |
> | 4B RLLM | KV-Quant |   GSM8K   |   84.05  |  617.30  |  489.17 | 0.1948 |     285334    |
> | 4B RLLM | KV-Quant |  MATH-500 |   44.67  |  916.27  |  654.39 | 0.2614 |     575223    |
> | 4B RLLM | KV-Quant | AIME-2024 |   18.89  |  352.53  | 1026.07 | 0.2811 |     351464    |
> | 4B RLLM | KV-Quant |    GPQA   |   8.00   |  1164.06 |  990.93 | 0.4428 |    1117451    |
>
> **Table-4 4B RLLM KV Cache Quantization**
>
> |  Model  |  Method  |  Dataset  | Accuracy | E2E Time |   TPS  |  TTFVT  | Output Tokens |
> | :-----: | :------: | :-------: | :------: | :------: | :----: | :-----: | :-----------: |
> | 8B RLLM |     /    |   GSM8K   |   71.67  |  312.38  | 486.23 |  4.1972 |     135267    |
> | 8B RLLM |     /    |  MATH-500 |   46.00  |  1475.65 | 430.24 | 26.6753 |     610506    |
> | 8B RLLM |     /    | AIME-2024 |   22.22  |  565.46  | 621.69 | 63.2468 |     343983    |
> | 8B RLLM |     /    |    GPQA   |   10.33  |  1904.99 | 610.99 | 50.1914 |    1127883    |
> | 8B RLLM | KV-Quant |   GSM8K   |   71.33  |  294.56  | 523.36 |  3.3801 |     136574    |
> | 8B RLLM | KV-Quant |  MATH-500 |   45.33  |  1230.57 | 521.34 | 21.3810 |     617166    |
> | 8B RLLM | KV-Quant | AIME-2024 |   20.00  |  425.85  | 821.24 | 46.3206 |     342163    |
> | 8B RLLM | KV-Quant |    GPQA   |   8.33   |  1467.07 | 803.51 | 42.3785 |    1142755    |
>
> **Table-5 8B RLLM KV Cache Quantization**
>
> |   Model  |  Method  |  Dataset  | Accuracy | E2E Time |   TPS   |  TTFT  | Output Tokens |
> | :------: | :------: | :-------: | :------: | :------: | :-----: | :----: | :-----------: |
> | 1.5B LLM |     /    |   GSM8K   |   43.00  |  229.20  |  983.57 | 0.0464 |     208813    |
> | 1.5B LLM |     /    |  MATH-500 |   2.33   |  269.58  | 1362.62 | 0.0383 |     342955    |
> | 1.5B LLM |     /    | AIME-2024 |   3.33   |   77.88  | 1612.82 | 0.0397 |     118053    |
> | 1.5B LLM |     /    |    GPQA   |   4.00   |  271.69  | 1469.04 | 0.0543 |     363076    |
> | 1.5B LLM | KV-Quant |   GSM8K   |   11.30  |  271.26  | 1451.46 | 0.0765 |     377097    |
> | 1.5B LLM | KV-Quant |  MATH-500 |   2.00   |  274.17  | 1585.54 | 0.0862 |     410326    |
> | 1.5B LLM | KV-Quant | AIME-2024 |   0.00   |   82.58  | 1961.33 | 0.0892 |     154408    |
> | 1.5B LLM | KV-Quant |    GPQA   |   4.33   |  281.86  | 1585.92 | 0.1199 |     410969    |
>
> **Table-6 1B LLM KV Cache Quantization**

---

> ### Author Response · Authors · 2025-11-25
> **Author Rebuttal Part (3/8)**
>
> |  Model |  Method  |  Dataset  | Accuracy | E2E Time |   TPS  |  TTFT  | Output Tokens |
> | :----: | :------: | :-------: | :------: | :------: | :----: | :----: | :-----------: |
> | 3B LLM |     /    |   GSM8K   |   71.33  |   96.09  | 735.03 | 0.1336 |     54009     |
> | 3B LLM |     /    |  MATH-500 |   42.33  |  403.87  | 450.49 | 0.1572 |     157568    |
> | 3B LLM |     /    | AIME-2024 |   2.22   |  243.30  | 626.80 | 0.1746 |     144940    |
> | 3B LLM |     /    |    GPQA   |   25.00  |  594.20  | 518.77 | 0.3193 |     272210    |
> | 3B LLM | KV-Quant |   GSM8K   |   66.67  |   96.61  | 813.14 | 0.2074 |     53805     |
> | 3B LLM | KV-Quant |  MATH-500 |   41.67  |  362.41  | 471.08 | 0.2505 |     146349    |
> | 3B LLM | KV-Quant | AIME-2024 |   2.22   |  205.04  | 746.91 | 0.2651 |     145582    |
> | 3B LLM | KV-Quant |    GPQA   |   25.67  |  420.76  | 590.33 | 0.3950 |     212342    |
>
> **Table-7 3B RLLM KV Cache Quantization**
>
> |  Model |  Method  |  Dataset  | Accuracy | E2E Time |   TPS  |  TTFT  | Output Tokens |
> | :----: | :------: | :-------: | :------: | :------: | :----: | :----: | :-----------: |
> | 4B LLM |     /    |   GSM8K   |   58.33  |  327.20  | 329.30 | 0.0849 |     91124     |
> | 4B LLM |     /    |  MATH-500 |   20.67  |  825.03  | 444.95 | 0.0863 |     342727    |
> | 4B LLM |     /    | AIME-2024 |   0.00   |  291.15  | 780.29 | 0.1031 |     219620    |
> | 4B LLM |     /    |    GPQA   |   12.33  |  596.03  | 403.09 | 0.1792 |     204211    |
> | 4B LLM | KV-Quant |   GSM8K   |   57.00  |  120.82  | 665.70 | 0.1722 |     63809     |
> | 4B LLM | KV-Quant |  MATH-500 |   22.33  |  639.76  | 504.21 | 0.2299 |     298200    |
> | 4B LLM | KV-Quant | AIME-2024 |   0.00   |  253.91  | 799.20 | 0.2492 |     195362    |
> | 4B LLM | KV-Quant |    GPQA   |   12.00  |  554.87  | 449.49 | 0.4071 |     213366    |
>
> **Table-8 4B RLLM KV Cache Quantization**
>
> |  Model |  Method  |  Dataset  | Accuracy | E2E Time |   TPS  |  TTFT  | Output Tokens |
> | :----: | :------: | :-------: | :------: | :------: | :----: | :----: | :-----------: |
> | 8B LLM |     /    |   GSM8K   |   10.67  |  1132.57 | 384.20 | 0.1439 |     418505    |
> | 8B LLM |     /    |  MATH-500 |   1.67   |  986.09  | 260.40 | 0.1214 |     232406    |
> | 8B LLM |     /    | AIME-2024 |   0.00   |  338.52  | 435.69 | 0.1330 |     139929    |
> | 8B LLM |     /    |    GPQA   |   16.94  |  590.85  | 228.47 | 0.2394 |     98948     |
> | 8B LLM | KV-Quant |   GSM8K   |   9.00   |  1312.40 | 330.81 | 0.4438 |     417527    |
> | 8B LLM | KV-Quant |  MATH-500 |   3.00   |  1083.71 | 249.87 | 0.7026 |     246408    |
> | 8B LLM | KV-Quant | AIME-2024 |   0.00   |  369.53  | 310.85 | 0.7762 |     107308    |
> | 8B LLM | KV-Quant |    GPQA   |   12.67  |  726.95  | 200.91 | 1.3001 |     110003    |
>
> **Table-9 8B RLLM KV Cache Quantization**
>
> ## **2. Why Prefix Cache (PC) Slows Down Small RLLMs ?**
>
> Our paper reports that prefix caching improves throughput for large RLLMs but unexpectedly degrades efficiency for small ones. We performed additional analysis and now provide a  system-level explanation based on the overhead–benefit trade-off.
>
> **1. Overhead outweighs benefits for small language models**
>
> Prefix caching introduces non-trivial runtime overhead, including:
> + Hashing and prefix matching
> + Cache block lookup
> + Metadata and memory management
>
> For large models, the prefill phase is computationally expensive. The savings from skipping prefill are much larger than these overheads, so PC provides net performance gains.
> For small models (e.g., 7B):
> The prefill phase is extremely lightweight (very few FLOPs).
> The PC overhead becomes comparable to—or even larger than—the actual prefill computation cost.
> As a result, the net effect is performance degradation.
>
> **2. RLLM workloads are decode-dominated (Long CoT)**
>
> A key property of RLLMs is their very long chain-of-thought (CoT) generation.
> RLLM inference time is dominated by the decode phase, which is memory-bound.
> Prefix caching only accelerates prefill, not decode.
> The decode bottleneck remains unchanged, while PC adds extra GPU-side memory and metadata operations.
> Thus, PC increases system complexity without accelerating the dominant computation stage.
>
> **3. Low cache hit rate in our workload**
>
> As noted in the experimental setup:
> Most benchmark queries are single-turn prompts.
> Prefix reuse is therefore low.
> In such low hit-rate scenarios, the cost of maintaining the cache exceeds the benefits.
>
> ### **Conclusion**
>
> Prefix caching helps large RLLMs because their prefill cost is large and benefits outweigh overhead. For small RLLMs, however, PC adds overhead but does not meaningfully reduce runtime, and may even worsen it due to increased memory-management complexity.
> Therefore, unless the workload has high prefix-reuse (e.g., multi-turn dialogs or shared/system prompts), we recommend disabling prefix caching for small RLLMs.
>
> ----
>
> **Table-10 to 17 are in the following.**
>
>
> ----

---

> ### Author Response · Authors · 2025-11-25
> **Author Rebuttal Part (4/8)**
>
> |   Model   |   Method   |  Dataset  | Accuracy | E2E Time |   TPS   |  TTFVT  | Output Tokens |
> | :-------: | :--------: | :-------: | :------: | :------: | :-----: | :-----: | :-----------: |
> | 1.5B RLLM |      /     |   GSM8K   |   78.67  |   81.8   |  1746.7 |   0.93  |     126351    |
> | 1.5B RLLM |      /     |  MATH-500 |   34.67  |   543.2  |  1151.5 |   7.66  |     601202    |
> | 1.5B RLLM |      /     | AIME-2024 |   10.00  |   171.4  |  2100.8 |   16.3  |     352712    |
> | 1.5B RLLM |      /     |    GPQA   |   7.00   |   602.5  |  1972.3 |   15.0  |    1152348    |
> | 1.5B RLLM | w/o Prefix |   GSM8K   |   75.33  |   84.51  | 1715.38 |  0.9577 |     128352    |
> | 1.5B RLLM | w/o Prefix |  MATH-500 |   42.00  |  541.61  | 1130.27 |  8.0090 |     587791    |
> | 1.5B RLLM | w/o Prefix | AIME-2024 |   16.67  |  170.91  | 2096.83 | 19.2384 |     350815    |
> | 1.5B RLLM | w/o Prefix |    GPQA   |   5.67   |  600.57  | 1974.81 | 13.6450 |    1149970    |
>
> **Table-10 1B RLLM without prefix caching**
>
> |  Model  |   Method   |  Dataset  | Accuracy | E2E Time |   TPS  |  TTFT  | Output Tokens |
> | :-----: | :--------: | :-------: | :------: | :------: | :----: | :----: | :-----------: |
> | 3B RLLM |      /     |   GSM8K   |   70.33  |  174.99  | 646.19 | 0.1236 |     96453     |
> | 3B RLLM |      /     |  MATH-500 |   35.00  |  493.92  | 494.95 | 0.1565 |     220089    |
> | 3B RLLM |      /     | AIME-2024 |   4.44   |  237.26  | 627.77 | 0.1762 |     141384    |
> | 3B RLLM |      /     |    GPQA   |   11.67  |  729.24  | 640.77 | 0.3535 |     431229    |
> | 3B RLLM | w/o Prefix |   GSM8K   |   74.67  |  172.62  | 646.13 | 0.1317 |     94910     |
> | 3B RLLM | w/o Prefix |  MATH-500 |   34.67  |  450.28  | 509.68 | 0.1981 |     205122    |
> | 3B RLLM | w/o Prefix | AIME-2024 |   6.67   |  203.48  | 825.89 | 0.2102 |     160493    |
> | 3B RLLM | w/o Prefix |    GPQA   |   7.33   |  601.27  | 736.82 | 0.2872 |     406986    |
>
> **Table-11 3B RLLM without prefix caching**
>
> |  Model  |   Method   |  Dataset  | Accuracy | E2E Time |   TPS  |  TTFT  | Output Tokens |
> | :-----: | :--------: | :-------: | :------: | :------: | :----: | :----: | :-----------: |
> | 4B RLLM |      /     |   GSM8K   |   87.00  |  606.39  | 448.15 | 0.1517 |     255127    |
> | 4B RLLM |      /     |  MATH-500 |   37.00  |  1143.54 | 513.00 | 0.3683 |     562267    |
> | 4B RLLM |      /     | AIME-2024 |   23.33  |  493.63  | 729.90 | 0.1963 |     352738    |
> | 4B RLLM |      /     |    GPQA   |   6.67   |  1655.89 | 706.87 | 0.6480 |    1134449    |
> | 4B RLLM | w/o Prefix |   GSM8K   |   84.33  |  643.99  | 437.34 | 0.3544 |     265022    |
> | 4B RLLM | w/o Prefix |  MATH-500 |   39.00  |  1158.26 | 502.03 | 0.5133 |     557106    |
> | 4B RLLM | w/o Prefix | AIME-2024 |   26.67  |  496.64  | 716.82 | 0.5513 |     348441    |
> | 4B RLLM | w/o Prefix |    GPQA   |   8.33   |  1667.94 | 699.65 | 0.7679 |    1130926    |
>
> **Table-12 4B RLLM without prefix caching**
>
> |  Model  |   Method   |  Dataset  | Accuracy | E2E Time |   TPS  |  TTFVT  | Output Tokens |
> | :-----: | :--------: | :-------: | :------: | :------: | :----: | :-----: | :-----------: |
> | 8B RLLM |      /     |   GSM8K   |   71.67  |  312.38  | 486.23 |  4.1972 |     135267    |
> | 8B RLLM |      /     |  MATH-500 |   46.00  |  1475.65 | 430.24 | 26.6753 |     610506    |
> | 8B RLLM |      /     | AIME-2024 |   22.22  |  565.46  | 621.69 | 63.2468 |     343983    |
> | 8B RLLM |      /     |    GPQA   |   10.33  |  1904.99 | 610.99 | 50.1914 |    1127883    |
> | 8B RLLM | w/o Prefix |   GSM8K   |   71.33  |  264.27  | 572.27 |  3.2143 |     134612    |
> | 8B RLLM | w/o Prefix |  MATH-500 |   48.00  |  1195.59 | 527.13 | 20.2698 |     605859    |
> | 8B RLLM | w/o Prefix | AIME-2024 |   13.33  |  428.79  | 847.52 | 53.3377 |     355845    |
> | 8B RLLM | w/o Prefix |    GPQA   |   8.00   |  1455.82 | 807.04 | 40.7557 |    1138860    |
>
> **Table-13 8B RLLM without prefix caching**
>
> |   Model  |   Method   |  Dataset  | Accuracy | E2E Time |   TPS   |  TTFT  | Output Tokens |
> | :------: | :--------: | :-------: | :------: | :------: | :-----: | :----: | :-----------: |
> | 1.5B LLM |      /     |   GSM8K   |   43.00  |  229.20  |  983.57 | 0.0464 |     208813    |
> | 1.5B LLM |      /     |  MATH-500 |   2.33   |  269.58  | 1362.62 | 0.0383 |     342955    |
> | 1.5B LLM |      /     | AIME-2024 |   3.33   |   77.88  | 1612.82 | 0.0397 |     118053    |
> | 1.5B LLM |      /     |    GPQA   |   4.00   |  271.69  | 1469.04 | 0.0543 |     363076    |
> | 1.5B LLM | w/o Prefix |   GSM8K   |   34.67  |  239.65  | 1005.01 | 0.0578 |     224224    |
> | 1.5B LLM | w/o Prefix |  MATH-500 |   5.67   |  268.40  | 1321.34 | 0.0580 |     330269    |
> | 1.5B LLM | w/o Prefix | AIME-2024 |    0.0   |   77.77  | 1714.10 | 0.0607 |     125741    |
> | 1.5B LLM | w/o Prefix |    GPQA   |   5.67   |  264.41  | 1450.04 | 0.0829 |     347353    |
>
> **Table-14 1B LLM without prefix caching**

---

> ### Author Response · Authors · 2025-11-25
> **Author Rebuttal Part (5/8)**
>
> |  Model |   Method   |  Dataset  | Accuracy | E2E Time |   TPS  |  TTFT  | Output Tokens |
> | :----: | :--------: | :-------: | :------: | :------: | :----: | :----: | :-----------: |
> | 3B LLM |      /     |   GSM8K   |   71.33  |   96.09  | 735.03 | 0.1336 |     54009     |
> | 3B LLM |      /     |  MATH-500 |   42.33  |  403.87  | 450.49 | 0.1572 |     157568    |
> | 3B LLM |      /     | AIME-2024 |   2.22   |  243.30  | 626.80 | 0.1746 |     144940    |
> | 3B LLM |      /     |    GPQA   |   25.00  |  594.20  | 518.77 | 0.3193 |     272210    |
> | 3B LLM | w/o Prefix |   GSM8K   |   68.67  |   86.37  | 823.13 | 0.1674 |     54468     |
> | 3B LLM | w/o Prefix |  MATH-500 |   44.00  |  338.06  | 528.22 | 0.2129 |     154197    |
> | 3B LLM | w/o Prefix | AIME-2024 |   6.67   |  188.33  | 717.12 | 0.2192 |     127496    |
> | 3B LLM | w/o Prefix |    GPQA   |   26.00  |  480.12  | 556.83 | 0.3029 |     231298    |
>
> **Table-15 3B LLM without prefix caching**
>
> |  Model |   Method   |  Dataset  | Accuracy | E2E Time |   TPS  |  TTFT  | Output Tokens |
> | :----: | :--------: | :-------: | :------: | :------: | :----: | :----: | :-----------: |
> | 4B LLM |      /     |   GSM8K   |   58.33  |  327.20  | 329.30 | 0.0849 |     91124     |
> | 4B LLM |      /     |  MATH-500 |   20.67  |  825.03  | 444.95 | 0.0863 |     342727    |
> | 4B LLM |      /     | AIME-2024 |   0.00   |  291.15  | 780.29 | 0.1031 |     219620    |
> | 4B LLM |      /     |    GPQA   |   12.33  |  596.03  | 403.09 | 0.1792 |     204211    |
> | 4B LLM | w/o Prefix |   GSM8K   |   57.67  |  184.00  | 450.33 | 0.3010 |     66238     |
> | 4B LLM | w/o Prefix |  MATH-500 |   22.33  |  787.06  | 428.68 | 0.4810 |     313024    |
> | 4B LLM | w/o Prefix | AIME-2024 |   1.11   |  339.31  | 508.50 | 0.5265 |     164978    |
> | 4B LLM | w/o Prefix |    GPQA   |   14.00  |  627.28  | 379.10 | 0.7312 |     201756    |
>
> **Table-16 4B LLM without prefix caching**
>
> |  Model |   Method   |  Dataset  | Accuracy | E2E Time |   TPS  |  TTFT  | Output Tokens |
> | :----: | :--------: | :-------: | :------: | :------: | :----: | :----: | :-----------: |
> | 8B LLM |      /     |   GSM8K   |   10.67  |  1132.57 | 384.20 | 0.1439 |     418505    |
> | 8B LLM |      /     |  MATH-500 |   1.67   |  986.09  | 260.40 | 0.1214 |     232406    |
> | 8B LLM |      /     | AIME-2024 |   0.00   |  338.52  | 435.69 | 0.1330 |     139929    |
> | 8B LLM |      /     |    GPQA   |   16.94  |  590.85  | 228.47 | 0.2394 |     98948     |
> | 8B LLM | w/o Prefix |   GSM8K   |   11.00  |  1342.13 | 344.13 | 0.4413 |     445235    |
> | 8B LLM | w/o Prefix |  MATH-500 |   1.67   |  1035.80 | 227.53 | 0.6856 |     211306    |
> | 8B LLM | w/o Prefix | AIME-2024 |   1.11   |  366.74  | 302.76 | 0.7516 |     103473    |
> | 8B LLM | w/o Prefix |    GPQA   |   12.62  |  657.74  | 198.44 | 1.0510 |     94475     |
>
> **Table-17 6B LLM without prefix caching**
>
> ----

---

> ### Author Response · Authors · 2025-11-25
> **Author Rebuttal Part (6/8)**
>
> > **Q2**: Missing comparison with standard LLMs in Section 5. Section 5 focuses on evaluating the effectiveness of several optimization techniques—such as quantization, speculative decoding, prefix caching, and KV-cache quantization—on reasoning LLMs. However, it does not include any baseline results for LLMs under the same experimental setup. Without this comparison, it is difficult to determine whether the reported behaviors (e.g., prefix caching being ineffective for 7B models, speculative decoding offering limited speedup, or quantization showing inconsistent gains) are unique challenges of reasoning models or common limitations of large language models in general.
>
> Thank you for raising this essential point. We fully agree that including standard LLM baselines is crucial for determining whether the observed behaviors are **unique challenges of RLLMs** or **common limitations across all LLMs**.
>
> While our original intention in Section 5 was to evaluate how well existing optimization techniques—originally designed for standard LLMs—transfer to the emerging class of RLLMs, we have supplemented  experiments with **direct comparisons against their corresponding base LLMs** in Q1. These extended results reveal several important insights:
>
> ----
>
> ### **1. KV-Cache Quantization: A Model-Architecture Issue, Not RLLM-Specific**
>
> As discussed in our response to Q1, we compared each RLLM with its corresponding base LLM.
>
> *   **Experimental finding:**\
>     The severe performance collapse after enabling KV-cache quantization is **not unique to RLLMs**.\
>     Models based on the *Qwen-2.5-Math* architecture exhibit the same failure case even in their base LLM form, while LLaMA-based LLMs and RLLMs remain stable .
> *   **Interpretation:**\
>     Small reasoning models have lower parameter redundancy and are therefore more sensitive to quantization-induced precision loss.\
>     When combined with architecture-specific operator incompatibilities in current inference engines (e.g., vLLM’s handling of Qwen-Math KV-cache kernels), this sensitivity can amplify into catastrophic accuracy degradation.
> *   **Conclusion:**\
>     KV-cache quantization must be applied **with extreme caution** for small RLLMs, and architecture-specific validation is necessary before deployment.
>
> ***
>
> ### **2. Prefix Caching: A General Bottleneck for Small Models, but More Harmful for RLLMs**
>
> We further compared small **base LLMs vs. RLLMs** under PC on/off settings.
>
> *   **Experimental finding:**\
>     Both small LLMs and small RLLMs experience reduced throughput and increased E2E latency when PC is enabled.\
>     This indicates the issue is **general to small models**, not exclusive to RLLMs.
> *   **General cause (shared by LLM and RLLM):**\
>     For small models, the **Prefill phase is extremely cheap**.\
>     The overhead of PC—hashing, block lookup, metadata management, cache maintenance—can exceed the time saved by skipping prefill, resulting in system slowdown.
> *   **RLLM-specific amplification:**\
>     RLLMs additionally suffer because:
>
>     *   Their inference time is dominated by a **very long decode phase** (long CoT), which PC does *not* accelerate.
>     *   PC consumes extra memory for hash tables and cache blocks, reducing available memory for long CoT sequences.
>     *   This increases memory fragmentation and may trigger more frequent swapping, further harming performance.
>
> ***
>
> ### **Summary**
>
> Our extended comparison shows that:
>
> *   Some issues (e.g., PC inefficiency on small models) are **general small-model limitations**.
> *   However, RLLM-specific characteristics—especially **long CoTs and decode-heavy workloads**—further **exacerbate** these limitations.
> *   As a result, **optimizations designed for standard LLMs cannot be directly applied to small RLLMs** without careful redesign.
>
> We believe these findings reinforce the paper’s motivation: **RLLMs require their own serving optimizations rather than naïvely reusing LLM-centric techniques.**

---

> ### Author Response · Authors · 2025-11-25
> **Author Rebuttal Part (7/8)**
>
> > Q3: Unclear implications for system optimization. While the paper presents many interesting empirical patterns, it does not clearly explain how these observations could guide future optimization of RLLM inference systems. The paper stops at observation without turning the results into actionable guidance for improving RLLM
>
> Thank you for the helpful comment. We would like to clarify that the primary goal of this paper is to establish that RLLMs exhibit **distinct serving behaviors **compared with standard LLMs, and to provide **directional guidance** for future system optimization, rather than proposing complete new system designs within this work.
>
> That said, the empirical observations we present naturally lead to several **actionable implications** for designing future RLLM-oriented inference systems. We summarize these optimization directions below.
>
> ***
>
> ### **1. Memory and KV-Cache Management Optimizations**
>
> Our results show that RLLMs incur **higher and more volatile memory usage** due to their long chain-of-thought generation.
>
> **Implications for system design:**
>
> *   Develop \*\*finer-grained memory and KV-cache management policies \*\*tailored to long reasoning chains.
> *   Explore more adaptive **re-prefill / KV reload** mechanisms.
> *   Improve **cache lifetime management** to avoid large oscillations at CoT boundaries.
> *   Investigate **selective KV swapping/offloading** to mitigate GPU memory pressure under long decoding sequences.
>
> These insights suggest that memory management—largely sufficient for standard LLMs—needs to be substantially reconsidered for RLLMs.
>
> ***
>
> ### **2. Straggler-Aware Scheduling**
>
> We observe strong **straggler effects** within batches due to large variations in reasoning length. Hard queries dominate batch completion time, reducing GPU utilization toward the tail.
>
> **Implications for system design:**
>
> *   Incorporate **difficulty-aware** or **runtime-adaptive scheduling**, dynamically prioritizing or reordering requests.
> *   In multi-instance deployments, use **straggler routing** or **instance-level isolation** to reduce interference between short and long requests.
> *   Explore **preemption policies** that reduce tail latency for heterogeneous workloads.
>
> This suggests that batch scheduling strategies effective for standard LLMs may be insufficient for decode-heavy RLLM workloads.
>
> ***
>
> ### **3. Asymmetric Prefill–Decode Resource Allocation**
>
> Compared with standard LLMs, RLLMs are **much more decode-heavy**, and their runtime varies significantly with task difficulty rather than input length.
>
> **Implications for system design:**
>
> *   Leverage **asymmetric prefill–decode disaggregation**, assigning more resources to decode workers.
> *   Adapt existing PD-disaggregation frameworks to RLLM workloads to improve utilization and throughput.
> *   Design resource-allocation policies that explicitly account for **task-dependent decode variance**.
>
> This highlights opportunities to customize PD-disaggregation specifically for reasoning workloads.
>
> ***
>
> ### **4. Co-Design Opportunities with Hardware**
>
> RLLMs generate substantially longer sequences, leading to **higher bandwidth demand and KV-access intensity** than standard LLMs.
>
> **Implications for system design:**
>
> *   Explore hardware–software co-design for **KV-access–optimized accelerators**.
> *   Develop runtime systems that better exploit memory hierarchy or **optimize KV-cache bandwidth bottlenecks**.
> *   Consider architectural changes that reduce decode-phase latency for long-sequence workloads.
>
> These observations motivate the possibility of future RLLM-specialized hardware.
>
> ***
>
> ### **5. Multi-Model Cooperative Serving**
>
> RLLMs and standard LLMs differ greatly in runtime characteristics and cost profiles.
>
> **Implications for system design:**
>
> *   Adopt **hybrid serving pipelines**, where a router selects standard LLMs or RLLMs based on estimated task difficulty.
> *   Enable cost-efficient deployments where **RLLMs are used only when hard reasoning task is required**, while cheaper LLMs handle routine requests.
>
> This workload-aware routing can reduce cost and improve throughput in production systems.
>
> ***
>
> ### **Summary**
>
> Although this paper focuses on empirical characterization rather than proposing new mechanisms, each observation we report directly translates into **actionable and concrete system-design opportunities**. We hope these insights serve as a foundation for future work on building efficient, RLLM-specialized inference systems.

---

> ### Author Response · Authors · 2025-11-25
> **Author Rebuttal Part (8/8)**
>
> We hope our clarifications adequately resolve the raised concerns, and we appreciate the opportunity to elaborate further if necessary.

---

### Official Review · Reviewer_hqmG · 2025-11-02

**Soundness:** 3
**Presentation:** 2
**Contribution:** 3
**Rating:** 6
**Confidence:** 2

**Summary:**

In this paper, as indicated by the title, the authors empirically investigates how reasoning LLMs (RLLMs) behave under ``inference serving`` and how well standard serving optimizations transfer from vanilla LLMs. The authors first motivate that RLLMs, due to long CoT generation, place qualitatively different demands on serving systems and pricing models (token budgets). They pose the central question: `` Is there any distinct difference in serving behaviors between LLM and RLLM?``

Then the authors introduce ASU, a three-part assessment that considers Accuracy, Service-end, and User-end metrics. They also provide an evaluation suite  and run controlled studies across model sizes (7B–70B), two engines (vLLM and SGLang), and datasets emphasizing math/knowledge tasks (GSM8K, MATH-500, AIME-2024, GPQA).

The authors have some good findings:

- KV-cache usage:  ``RLLM exhibits significant KV Cache fluctuations and usage.``

- Straggler requests: ``long tail distribution of requests running
time caused by slow requests.``

- Adaptive running time: ``RLLM solves different difficulty level problems with adaptive
running time.``

- Domain preference: `` RLLM excels LLM on math reasoning while on-par on knowledge intensive tasks.``

The paper then investigates common serving optimisations for RLLMs, including
- Weight quantization
- KV-cache quantization
- Prefix caching
- Speculative decoding

And the authors got further meaningful observations:

- `` MWQ methods exert differing impacts on various metrics of RLLM inference ``

- `` KV Cache quantization can improve running efficiency for sufficient large RLLM. ``

- `` PC can accelerate larger RLLMs (14B and above) without performance degrade. ``

- `` SD improves the running time of RLLMs and deteriorates metrics like TPS. ``

I have to admit that I'm not an expert in LLM serving. It appears to me that this paper makes meaningful contributions.

**Strengths:**

``S1``: As long-CoT reasoning models become mainstream with the emergence of many prevalent deep reasoning models, their serving behavior is a an important systems problem.

``S2``: Good breadth of experiments across engines (vLLM and SGLang), model scales, and several optimizations (weights, KV, prefix cache, speculation).

``S3``: The summary of observations and findings are clear.

**Weaknesses:**

``W1``: The study focuses on GSM8K, MATH-500, AIME-2024, GPQA. Given rapid shifts in eval suites and the benchmark nature of this paper, I’d suggest the authors add SuperGPQA and CommonsenseQA.

``W2``: I would have expected to see some elaborated explanations of some observations in the paper. For example, the authors claim that:

- “We find that for sufficiently large RLLMs (14B and above), prefix caching significantly improves runtime speed and serving metrics without compromising performance. However, for 7B models, prefix caching negatively impacts efficiency, leading to increased latency.”

It’s interesting to see some discussions on the behind reason.

``W3``: Some typos issues such as “emeraged” and “perserved”. The authors should more carefully polish their paper.

Again, I have to admit that I’m not very familiar with LLM serving. I’ll look into the comments of other reviewers who are experts in this area for my final rating.

**Questions:**

``Q1``:  Why do prefix cache and KV-FP8 help large models (14B and above) but harm 7B?

---

> ### Author Response · Authors · 2025-11-25
> **Author Rebuttal Part (1/6)**
>
> Dear Reviewer hqmG:
>
> We appreciate the your time and thoughtful evaluation of our work. We have carefully considered each comment and address them point-by-point in the following rebuttal.
>
> ----
>
> | RLLM  Type |              RLLM Name              | LLM Type |          LLM name          |
> | :-------------: | :---------------------------------------: | :------------: | :------------------------------: |
> |     1B RLLM     | deepseek-ai/DeepSeek-R1-Distill-Qwen-1.5B |     1B LLM     |      Qwen/Qwen2.5-Math-1.5B      |
> |     3B RLLM     |  suayptalha/DeepSeek-R1-Distill-Llama-3B  |     3B LLM     | meta-llama/Llama-3.2-3B-Instruct |
> |     4B RLLM     |       microsoft/Phi-4-mini-reasoning      |     4B LLM     |   microsoft/Phi-4-mini-instruct  |
> |     8B RLLM     |  deepseek-ai/DeepSeek-R1-Distill-Llama-8B |     8B LLM     |      meta-llama/Llama-3.1-8B     |
>
> **Table-1 Employed models in our rebuttal.**
>
>
> ----
>
> > Q1: The study focuses on GSM8K, MATH-500, AIME-2024, GPQA. Given rapid shifts in eval suites and the benchmark nature of this paper, I’d suggest the authors add SuperGPQA and CommonsenseQA.
>
> Thank you for the suggestion. We fully agree that **SuperGPQA** and **CommonsenseQA** are important and increasingly recognized benchmarks, and we plan to include evaluation results on these datasets in future revisions of the paper.
>
> Below we further clarify why the current version focuses on **GSM8K**, **MATH-500**, **AIME-2024**, and **GPQA**:
>
> 1.  **These datasets cover the core capability range of RLLMs.**\
>     As discussed in the paper, RLLMs exhibit their primary advantages on complex reasoning tasks such as mathematics and code, while their performance on knowledge-intensive tasks typically remains similar to that of the underlying base LLM—a pattern also confirmed by our experiments.
>
>     *   **GSM8K**, **MATH-500**, and **AIME-2024** correspond to mathematical reasoning tasks at *easy → medium → difficult* levels, enabling a systematic analysis of RLLMs’ reasoning depth.
>     *   **GPQA** serves as a knowledge-reasoning benchmark, facilitating comparison between RLLMs and LLMs in knowledge-heavy scenarios.
> 2.  **These datasets are widely adopted as mainstream benchmarks in both academia and industry.**\
>     As noted in Section 3.1, these four datasets are used extensively in recent work on RLLM evaluation, including multiple representative models released in 2024–2025\[1]\[2]\[3]\[4]. They therefore provide strong generality and comparability.
>
> In summary, our current dataset selection sufficiently captures the key reasoning capabilities of RLLMs while remaining aligned with existing research practices. We appreciate the reviewer’s recommendation and will include additional datasets in future versions to further broaden evaluation coverage.
>
> > **Q2** Some typos issues such as “emeraged” and “perserved”. The authors should more carefully polish their paper.
>
> Thank you for pointing this out. We will carefully revise the manuscript and thoroughly check for typos and other writing issues in our submission.

---

> ### Author Response · Authors · 2025-11-25
> **Author Rebuttal (Part 2/6)**
>
> > **Q3**  I would have expected to see some elaborated explanations of some observations in the paper. For example, the authors claim that:
> >
> > *   “We find that for sufficiently large RLLMs (14B and above), prefix caching significantly improves runtime speed and serving metrics without compromising performance. However, for 7B models, prefix caching negatively impacts efficiency, leading to increased latency.”
> >
> > It’s interesting to see some discussions on the behind reason.
>
> Thank you for raising this important point. We agree that providing a deeper analysis would strengthen the paper, and following the reviewer’s suggestion, we conducted additional controlled experiments across a broader range of model sizes and architectures (listed in Table-1) to understand why small RLLMs respond poorly to KV-cache quantization.
>
> ***
>
> Our expanded analysis leads to several plausible and experimentally supported hypotheses:
>
> ### **1. Why KV-Cache Quantization Harms Small RLLMs ？**
>
> In Observation 5.2, we reported that enabling KV-cache quantization causes drastic accuracy drops for small RLLMs (e.g., 7B), whereas large models remain stable. Our new experiments across 1.5B, 3B, 4B, and 8B models reveal the following insights （Table 2 - Table 9）:
>
> #### **(a) High sensitivity of small models to quantization noise**
>
> Across all tested RLLM and LLM, larger models demonstrate robust performance under KV-cache quantization, but smaller models show substantial variance or collapse.
>
> We hypothesize that this stems from small models having:
>
> *   Lower parameter redundancy,
> *   Higher reliance on precise attention scores, and
> *   Weaker ability to tolerate perturbations in multi-step CoT reasoning.
>
> This makes their reasoning chains particularly susceptible to small errors introduced by low-precision KV states.
>
> #### **(b) Architecture–engine compatibility issues**
>
> Performance collapse is highly **architecture-dependent**:
>
> *   Failures are concentrated in **Qwen-2.5-Math** 1.5B and 7B models.
> *   Other small models (e.g., 3B, 4B) remain relatively stable after quantization.
>
> This suggests that part of the degradation arises from **incomplete model support** for specific architectures in the current serving engine (e.g. vLLM), rather than from model scale.
>
> #### **(c) Quantization appears to change reasoning behavior**
>
> We also observed that KV-cache quantization substantially changes the **number of outputed tokens** (i.e., reasoning chain length) for 1B/3B/4B RLLM models (also 4B LLM). This indicates that quantization noise may disrupt the model’s \*\* step-by-step reasoning trajectory\*\*, which aligns with the hypothesis that long CoT requires high-precision attention dynamics.
>
> #### **(d) Small RLLMs are affected far more than small LLMs**
>
> Interestingly, the accuracy degradation is significantly larger for \*\*small RLLMs \*\*than for **small non-reasoning LLMs**, suggesting that long CoT reasoning is inherently more sensitive to KV-cache perturbations.
>
> **Summary:**  Our deeper analysis indicates that the severe degradation observed in small RLLMs under KV-cache quantization arises from the combination of three factors: 1) their low parameter redundancy makes them highly sensitive to quantization noise, 2) their long chain-of-thought reasoning requires precise attention dynamics, and 3) architecture–kernel mismatches in the current vLLM implementation further amplify these vulnerabilities. Therefore,  we explicitly recommend applying KV-cache quantization to small RLLMs only with great caution.
>
> |   Model   |  Method  |  Dataset  | Accuracy | E2E Time |   TPS   | TTFVT | Output Tokens |
> | :-------: | :------: | :-------: | :------: | :------: | :-----: | :---: | :-----------: |
> | 1.5B RLLM |     /    |   GSM8K   |   78.67  |   81.8   |  1746.7 |  0.93 |     126351    |
> | 1.5B RLLM |     /    |  MATH-500 |   34.67  |   543.2  |  1151.5 |  7.66 |     601202    |
> | 1.5B RLLM |     /    | AIME-2024 |   10.00  |   171.4  |  2100.8 |  16.3 |     352712    |
> | 1.5B RLLM |     /    |    GPQA   |   7.00   |   602.5  |  1972.3 |  15.0 |    1152348    |
> | 1.5B RLLM | KV-Quant |   GSM8K   |   1.33   |  645.27  | 1636.84 |  1.77 |    1039590    |
> | 1.5B RLLM | KV-Quant |  MATH-500 |   1.33   |  683.57  | 1793.17 | 19.50 |    1201382    |
> | 1.5B RLLM | KV-Quant | AIME-2024 |   0.00   |  203.76  | 1845.88 |  1.49 |     368566    |
> | 1.5B RLLM | KV-Quant |    GPQA   |   0.33   |  684.12  | 1823.20 | 15.96 |    1211232    |
>
> **Table-2 1B RLLM KV Cache Quantization**

---

> ### Author Response · Authors · 2025-11-25
> **Author Rebuttal (Part 3/6)**
>
> |  Model  |  Method  |  Dataset  | Accuracy | E2E Time |   TPS  |  TTFT  | Output Tokens |
> | :-----: | :------: | :-------: | :------: | :------: | :----: | :----: | :-----------: |
> | 3B RLLM |     /    |   GSM8K   |   70.33  |  174.99  | 646.19 | 0.1236 |     96453     |
> | 3B RLLM |     /    |  MATH-500 |   35.00  |  493.92  | 494.95 | 0.1565 |     220089    |
> | 3B RLLM |     /    | AIME-2024 |   4.44   |  237.26  | 627.77 | 0.1762 |     141384    |
> | 3B RLLM |     /    |    GPQA   |   11.67  |  729.24  | 640.77 | 0.3535 |     431229    |
> | 3B RLLM | KV-Quant |   GSM8K   |   70.33  |  113.69  | 886.52 | 0.1620 |     84167     |
> | 3B RLLM | KV-Quant |  MATH-500 |   35.33  |  529.98  | 485.45 | 0.2157 |     232903    |
> | 3B RLLM | KV-Quant | AIME-2024 |   3.33   |  209.76  | 771.79 | 0.2341 |     154332    |
> | 3B RLLM | KV-Quant |    GPQA   |   12.00  |  632.22  | 731.24 | 0.3161 |     426256    |
>
> **Table-3 3B RLLM KV Cache Quantization**
>
> |  Model  |  Method  |  Dataset  | Accuracy | E2E Time |   TPS   |  TTFT  | Output Tokens |
> | :-----: | :------: | :-------: | :------: | :------: | :-----: | :----: | :-----------: |
> | 4B RLLM |     /    |   GSM8K   |   87.00  |  606.39  |  448.15 | 0.1517 |     255127    |
> | 4B RLLM |     /    |  MATH-500 |   37.00  |  1143.54 |  513.00 | 0.3683 |     562267    |
> | 4B RLLM |     /    | AIME-2024 |   23.33  |  493.63  |  729.90 | 0.1963 |     352738    |
> | 4B RLLM |     /    |    GPQA   |   6.67   |  1655.89 |  706.87 | 0.6480 |    1134449    |
> | 4B RLLM | KV-Quant |   GSM8K   |   84.05  |  617.30  |  489.17 | 0.1948 |     285334    |
> | 4B RLLM | KV-Quant |  MATH-500 |   44.67  |  916.27  |  654.39 | 0.2614 |     575223    |
> | 4B RLLM | KV-Quant | AIME-2024 |   18.89  |  352.53  | 1026.07 | 0.2811 |     351464    |
> | 4B RLLM | KV-Quant |    GPQA   |   8.00   |  1164.06 |  990.93 | 0.4428 |    1117451    |
>
> **Table-4 4B RLLM KV Cache Quantization**
>
> |  Model  |  Method  |  Dataset  | Accuracy | E2E Time |   TPS  |  TTFVT  | Output Tokens |
> | :-----: | :------: | :-------: | :------: | :------: | :----: | :-----: | :-----------: |
> | 8B RLLM |     /    |   GSM8K   |   71.67  |  312.38  | 486.23 |  4.1972 |     135267    |
> | 8B RLLM |     /    |  MATH-500 |   46.00  |  1475.65 | 430.24 | 26.6753 |     610506    |
> | 8B RLLM |     /    | AIME-2024 |   22.22  |  565.46  | 621.69 | 63.2468 |     343983    |
> | 8B RLLM |     /    |    GPQA   |   10.33  |  1904.99 | 610.99 | 50.1914 |    1127883    |
> | 8B RLLM | KV-Quant |   GSM8K   |   71.33  |  294.56  | 523.36 |  3.3801 |     136574    |
> | 8B RLLM | KV-Quant |  MATH-500 |   45.33  |  1230.57 | 521.34 | 21.3810 |     617166    |
> | 8B RLLM | KV-Quant | AIME-2024 |   20.00  |  425.85  | 821.24 | 46.3206 |     342163    |
> | 8B RLLM | KV-Quant |    GPQA   |   8.33   |  1467.07 | 803.51 | 42.3785 |    1142755    |
>
> **Table-5 8B RLLM KV Cache Quantization**
>
> |   Model  |  Method  |  Dataset  | Accuracy | E2E Time |   TPS   |  TTFT  | Output Tokens |
> | :------: | :------: | :-------: | :------: | :------: | :-----: | :----: | :-----------: |
> | 1.5B LLM |     /    |   GSM8K   |   43.00  |  229.20  |  983.57 | 0.0464 |     208813    |
> | 1.5B LLM |     /    |  MATH-500 |   2.33   |  269.58  | 1362.62 | 0.0383 |     342955    |
> | 1.5B LLM |     /    | AIME-2024 |   3.33   |   77.88  | 1612.82 | 0.0397 |     118053    |
> | 1.5B LLM |     /    |    GPQA   |   4.00   |  271.69  | 1469.04 | 0.0543 |     363076    |
> | 1.5B LLM | KV-Quant |   GSM8K   |   11.30  |  271.26  | 1451.46 | 0.0765 |     377097    |
> | 1.5B LLM | KV-Quant |  MATH-500 |   2.00   |  274.17  | 1585.54 | 0.0862 |     410326    |
> | 1.5B LLM | KV-Quant | AIME-2024 |   0.00   |   82.58  | 1961.33 | 0.0892 |     154408    |
> | 1.5B LLM | KV-Quant |    GPQA   |   4.33   |  281.86  | 1585.92 | 0.1199 |     410969    |
>
> **Table-6 1B LLM KV Cache Quantization**
>
>
> |  Model |  Method  |  Dataset  | Accuracy | E2E Time |   TPS  |  TTFT  | Output Tokens |
> | :----: | :------: | :-------: | :------: | :------: | :----: | :----: | :-----------: |
> | 3B LLM |     /    |   GSM8K   |   71.33  |   96.09  | 735.03 | 0.1336 |     54009     |
> | 3B LLM |     /    |  MATH-500 |   42.33  |  403.87  | 450.49 | 0.1572 |     157568    |
> | 3B LLM |     /    | AIME-2024 |   2.22   |  243.30  | 626.80 | 0.1746 |     144940    |
> | 3B LLM |     /    |    GPQA   |   25.00  |  594.20  | 518.77 | 0.3193 |     272210    |
> | 3B LLM | KV-Quant |   GSM8K   |   66.67  |   96.61  | 813.14 | 0.2074 |     53805     |
> | 3B LLM | KV-Quant |  MATH-500 |   41.67  |  362.41  | 471.08 | 0.2505 |     146349    |
> | 3B LLM | KV-Quant | AIME-2024 |   2.22   |  205.04  | 746.91 | 0.2651 |     145582    |
> | 3B LLM | KV-Quant |    GPQA   |   25.67  |  420.76  | 590.33 | 0.3950 |     212342    |
>
> **Table-7 3B RLLM KV Cache Quantization**

---

> ### Author Response · Authors · 2025-11-25
> **Author Rebuttal Part (4/6)**
>
> |  Model |  Method  |  Dataset  | Accuracy | E2E Time |   TPS  |  TTFT  | Output Tokens |
> | :----: | :------: | :-------: | :------: | :------: | :----: | :----: | :-----------: |
> | 4B LLM |     /    |   GSM8K   |   58.33  |  327.20  | 329.30 | 0.0849 |     91124     |
> | 4B LLM |     /    |  MATH-500 |   20.67  |  825.03  | 444.95 | 0.0863 |     342727    |
> | 4B LLM |     /    | AIME-2024 |   0.00   |  291.15  | 780.29 | 0.1031 |     219620    |
> | 4B LLM |     /    |    GPQA   |   12.33  |  596.03  | 403.09 | 0.1792 |     204211    |
> | 4B LLM | KV-Quant |   GSM8K   |   57.00  |  120.82  | 665.70 | 0.1722 |     63809     |
> | 4B LLM | KV-Quant |  MATH-500 |   22.33  |  639.76  | 504.21 | 0.2299 |     298200    |
> | 4B LLM | KV-Quant | AIME-2024 |   0.00   |  253.91  | 799.20 | 0.2492 |     195362    |
> | 4B LLM | KV-Quant |    GPQA   |   12.00  |  554.87  | 449.49 | 0.4071 |     213366    |
>
> **Table-8 4B RLLM KV Cache Quantization**
>
> |  Model |  Method  |  Dataset  | Accuracy | E2E Time |   TPS  |  TTFT  | Output Tokens |
> | :----: | :------: | :-------: | :------: | :------: | :----: | :----: | :-----------: |
> | 8B LLM |     /    |   GSM8K   |   10.67  |  1132.57 | 384.20 | 0.1439 |     418505    |
> | 8B LLM |     /    |  MATH-500 |   1.67   |  986.09  | 260.40 | 0.1214 |     232406    |
> | 8B LLM |     /    | AIME-2024 |   0.00   |  338.52  | 435.69 | 0.1330 |     139929    |
> | 8B LLM |     /    |    GPQA   |   16.94  |  590.85  | 228.47 | 0.2394 |     98948     |
> | 8B LLM | KV-Quant |   GSM8K   |   9.00   |  1312.40 | 330.81 | 0.4438 |     417527    |
> | 8B LLM | KV-Quant |  MATH-500 |   3.00   |  1083.71 | 249.87 | 0.7026 |     246408    |
> | 8B LLM | KV-Quant | AIME-2024 |   0.00   |  369.53  | 310.85 | 0.7762 |     107308    |
> | 8B LLM | KV-Quant |    GPQA   |   12.67  |  726.95  | 200.91 | 1.3001 |     110003    |
>
> **Table-9 8B RLLM KV Cache Quantization**
>
> ----
>
> > **Q4** Why do prefix cache and KV-FP8 help large models (14B and above) but harm 7B?
>
> Thank you for raising this important point. We agree that providing a deeper analysis would strengthen the paper, and following the reviewer’s suggestion, we conducted additional controlled experiments across a broader range of model sizes and architectures (listed in Table-1) to understand why small RLLMs respond poorly to prefix caching.
>
> ## **Why Prefix Cache (PC) Slows Down Small RLLMs ?**
>
> Our paper reports that prefix caching improves throughput for large RLLMs but unexpectedly degrades efficiency for small ones. We performed additional analysis and now provide a  system-level explanation based on the overhead–benefit trade-off.
>
> **1. Overhead outweighs benefits for small language models**
>
> Prefix caching introduces non-trivial runtime overhead, including:
> + Hashing and prefix matching
> + Cache block lookup
> + Metadata and memory management
>
> For large models, the prefill phase is computationally expensive. The savings from skipping prefill are much larger than these overheads, so PC provides net performance gains.
> For small models (e.g., 7B):
> The prefill phase is extremely lightweight (very few FLOPs).
> The PC overhead becomes comparable to—or even larger than—the actual prefill computation cost.
> As a result, the net effect is performance degradation.
>
> **2. RLLM workloads are decode-dominated (Long CoT)**
>
> A key property of RLLMs is their very long chain-of-thought (CoT) generation.
> RLLM inference time is dominated by the decode phase, which is memory-bound.
> Prefix caching only accelerates prefill, not decode.
> The decode bottleneck remains unchanged, while PC adds extra GPU-side memory and metadata operations.
> Thus, PC increases system complexity without accelerating the dominant computation stage.
>
> **3. Low cache hit rate in our workload**
>
> As noted in the experimental setup:
> Most benchmark queries are single-turn prompts.
> Prefix reuse is therefore low.
> In such low hit-rate scenarios, the cost of maintaining the cache exceeds the benefits.
>
> ### **Conclusion**
>
> Prefix caching helps large RLLMs because their prefill cost is large and benefits outweigh overhead. For small RLLMs, however, PC adds overhead but does not meaningfully reduce runtime, and may even worsen it due to increased memory-management complexity.

---

> ### Author Response · Authors · 2025-11-25
> **Author Rebuttal Part (5/6)**
>
> |   Model   |   Method   |  Dataset  | Accuracy | E2E Time |   TPS   |  TTFVT  | Output Tokens |
> | :-------: | :--------: | :-------: | :------: | :------: | :-----: | :-----: | :-----------: |
> | 1.5B RLLM |      /     |   GSM8K   |   78.67  |   81.8   |  1746.7 |   0.93  |     126351    |
> | 1.5B RLLM |      /     |  MATH-500 |   34.67  |   543.2  |  1151.5 |   7.66  |     601202    |
> | 1.5B RLLM |      /     | AIME-2024 |   10.00  |   171.4  |  2100.8 |   16.3  |     352712    |
> | 1.5B RLLM |      /     |    GPQA   |   7.00   |   602.5  |  1972.3 |   15.0  |    1152348    |
> | 1.5B RLLM | w/o Prefix |   GSM8K   |   75.33  |   84.51  | 1715.38 |  0.9577 |     128352    |
> | 1.5B RLLM | w/o Prefix |  MATH-500 |   42.00  |  541.61  | 1130.27 |  8.0090 |     587791    |
> | 1.5B RLLM | w/o Prefix | AIME-2024 |   16.67  |  170.91  | 2096.83 | 19.2384 |     350815    |
> | 1.5B RLLM | w/o Prefix |    GPQA   |   5.67   |  600.57  | 1974.81 | 13.6450 |    1149970    |
>
> **Table-10 1B RLLM without prefix caching**
>
> |  Model  |   Method   |  Dataset  | Accuracy | E2E Time |   TPS  |  TTFT  | Output Tokens |
> | :-----: | :--------: | :-------: | :------: | :------: | :----: | :----: | :-----------: |
> | 3B RLLM |      /     |   GSM8K   |   70.33  |  174.99  | 646.19 | 0.1236 |     96453     |
> | 3B RLLM |      /     |  MATH-500 |   35.00  |  493.92  | 494.95 | 0.1565 |     220089    |
> | 3B RLLM |      /     | AIME-2024 |   4.44   |  237.26  | 627.77 | 0.1762 |     141384    |
> | 3B RLLM |      /     |    GPQA   |   11.67  |  729.24  | 640.77 | 0.3535 |     431229    |
> | 3B RLLM | w/o Prefix |   GSM8K   |   74.67  |  172.62  | 646.13 | 0.1317 |     94910     |
> | 3B RLLM | w/o Prefix |  MATH-500 |   34.67  |  450.28  | 509.68 | 0.1981 |     205122    |
> | 3B RLLM | w/o Prefix | AIME-2024 |   6.67   |  203.48  | 825.89 | 0.2102 |     160493    |
> | 3B RLLM | w/o Prefix |    GPQA   |   7.33   |  601.27  | 736.82 | 0.2872 |     406986    |
>
> **Table-11 3B RLLM without prefix caching**
>
> |  Model  |   Method   |  Dataset  | Accuracy | E2E Time |   TPS  |  TTFT  | Output Tokens |
> | :-----: | :--------: | :-------: | :------: | :------: | :----: | :----: | :-----------: |
> | 4B RLLM |      /     |   GSM8K   |   87.00  |  606.39  | 448.15 | 0.1517 |     255127    |
> | 4B RLLM |      /     |  MATH-500 |   37.00  |  1143.54 | 513.00 | 0.3683 |     562267    |
> | 4B RLLM |      /     | AIME-2024 |   23.33  |  493.63  | 729.90 | 0.1963 |     352738    |
> | 4B RLLM |      /     |    GPQA   |   6.67   |  1655.89 | 706.87 | 0.6480 |    1134449    |
> | 4B RLLM | w/o Prefix |   GSM8K   |   84.33  |  643.99  | 437.34 | 0.3544 |     265022    |
> | 4B RLLM | w/o Prefix |  MATH-500 |   39.00  |  1158.26 | 502.03 | 0.5133 |     557106    |
> | 4B RLLM | w/o Prefix | AIME-2024 |   26.67  |  496.64  | 716.82 | 0.5513 |     348441    |
> | 4B RLLM | w/o Prefix |    GPQA   |   8.33   |  1667.94 | 699.65 | 0.7679 |    1130926    |
>
> **Table-12 4B RLLM without prefix caching**
>
> |  Model  |   Method   |  Dataset  | Accuracy | E2E Time |   TPS  |  TTFVT  | Output Tokens |
> | :-----: | :--------: | :-------: | :------: | :------: | :----: | :-----: | :-----------: |
> | 8B RLLM |      /     |   GSM8K   |   71.67  |  312.38  | 486.23 |  4.1972 |     135267    |
> | 8B RLLM |      /     |  MATH-500 |   46.00  |  1475.65 | 430.24 | 26.6753 |     610506    |
> | 8B RLLM |      /     | AIME-2024 |   22.22  |  565.46  | 621.69 | 63.2468 |     343983    |
> | 8B RLLM |      /     |    GPQA   |   10.33  |  1904.99 | 610.99 | 50.1914 |    1127883    |
> | 8B RLLM | w/o Prefix |   GSM8K   |   71.33  |  264.27  | 572.27 |  3.2143 |     134612    |
> | 8B RLLM | w/o Prefix |  MATH-500 |   48.00  |  1195.59 | 527.13 | 20.2698 |     605859    |
> | 8B RLLM | w/o Prefix | AIME-2024 |   13.33  |  428.79  | 847.52 | 53.3377 |     355845    |
> | 8B RLLM | w/o Prefix |    GPQA   |   8.00   |  1455.82 | 807.04 | 40.7557 |    1138860    |
>
> **Table-13 8B RLLM without prefix caching**
>
> |   Model  |   Method   |  Dataset  | Accuracy | E2E Time |   TPS   |  TTFT  | Output Tokens |
> | :------: | :--------: | :-------: | :------: | :------: | :-----: | :----: | :-----------: |
> | 1.5B LLM |      /     |   GSM8K   |   43.00  |  229.20  |  983.57 | 0.0464 |     208813    |
> | 1.5B LLM |      /     |  MATH-500 |   2.33   |  269.58  | 1362.62 | 0.0383 |     342955    |
> | 1.5B LLM |      /     | AIME-2024 |   3.33   |   77.88  | 1612.82 | 0.0397 |     118053    |
> | 1.5B LLM |      /     |    GPQA   |   4.00   |  271.69  | 1469.04 | 0.0543 |     363076    |
> | 1.5B LLM | w/o Prefix |   GSM8K   |   34.67  |  239.65  | 1005.01 | 0.0578 |     224224    |
> | 1.5B LLM | w/o Prefix |  MATH-500 |   5.67   |  268.40  | 1321.34 | 0.0580 |     330269    |
> | 1.5B LLM | w/o Prefix | AIME-2024 |    0.0   |   77.77  | 1714.10 | 0.0607 |     125741    |
> | 1.5B LLM | w/o Prefix |    GPQA   |   5.67   |  264.41  | 1450.04 | 0.0829 |     347353    |
>
> **Table-14 1B LLM without prefix caching**

---

> ### Author Response · Authors · 2025-11-25
> **Author Rebuttal Part (6/6)**
>
> |  Model |   Method   |  Dataset  | Accuracy | E2E Time |   TPS  |  TTFT  | Output Tokens |
> | :----: | :--------: | :-------: | :------: | :------: | :----: | :----: | :-----------: |
> | 3B LLM |      /     |   GSM8K   |   71.33  |   96.09  | 735.03 | 0.1336 |     54009     |
> | 3B LLM |      /     |  MATH-500 |   42.33  |  403.87  | 450.49 | 0.1572 |     157568    |
> | 3B LLM |      /     | AIME-2024 |   2.22   |  243.30  | 626.80 | 0.1746 |     144940    |
> | 3B LLM |      /     |    GPQA   |   25.00  |  594.20  | 518.77 | 0.3193 |     272210    |
> | 3B LLM | w/o Prefix |   GSM8K   |   68.67  |   86.37  | 823.13 | 0.1674 |     54468     |
> | 3B LLM | w/o Prefix |  MATH-500 |   44.00  |  338.06  | 528.22 | 0.2129 |     154197    |
> | 3B LLM | w/o Prefix | AIME-2024 |   6.67   |  188.33  | 717.12 | 0.2192 |     127496    |
> | 3B LLM | w/o Prefix |    GPQA   |   26.00  |  480.12  | 556.83 | 0.3029 |     231298    |
>
> **Table-15 3B LLM without prefix caching**
>
> |  Model |   Method   |  Dataset  | Accuracy | E2E Time |   TPS  |  TTFT  | Output Tokens |
> | :----: | :--------: | :-------: | :------: | :------: | :----: | :----: | :-----------: |
> | 4B LLM |      /     |   GSM8K   |   58.33  |  327.20  | 329.30 | 0.0849 |     91124     |
> | 4B LLM |      /     |  MATH-500 |   20.67  |  825.03  | 444.95 | 0.0863 |     342727    |
> | 4B LLM |      /     | AIME-2024 |   0.00   |  291.15  | 780.29 | 0.1031 |     219620    |
> | 4B LLM |      /     |    GPQA   |   12.33  |  596.03  | 403.09 | 0.1792 |     204211    |
> | 4B LLM | w/o Prefix |   GSM8K   |   57.67  |  184.00  | 450.33 | 0.3010 |     66238     |
> | 4B LLM | w/o Prefix |  MATH-500 |   22.33  |  787.06  | 428.68 | 0.4810 |     313024    |
> | 4B LLM | w/o Prefix | AIME-2024 |   1.11   |  339.31  | 508.50 | 0.5265 |     164978    |
> | 4B LLM | w/o Prefix |    GPQA   |   14.00  |  627.28  | 379.10 | 0.7312 |     201756    |
>
> **Table-16 4B LLM without prefix caching**
>
> |  Model |   Method   |  Dataset  | Accuracy | E2E Time |   TPS  |  TTFT  | Output Tokens |
> | :----: | :--------: | :-------: | :------: | :------: | :----: | :----: | :-----------: |
> | 8B LLM |      /     |   GSM8K   |   10.67  |  1132.57 | 384.20 | 0.1439 |     418505    |
> | 8B LLM |      /     |  MATH-500 |   1.67   |  986.09  | 260.40 | 0.1214 |     232406    |
> | 8B LLM |      /     | AIME-2024 |   0.00   |  338.52  | 435.69 | 0.1330 |     139929    |
> | 8B LLM |      /     |    GPQA   |   16.94  |  590.85  | 228.47 | 0.2394 |     98948     |
> | 8B LLM | w/o Prefix |   GSM8K   |   11.00  |  1342.13 | 344.13 | 0.4413 |     445235    |
> | 8B LLM | w/o Prefix |  MATH-500 |   1.67   |  1035.80 | 227.53 | 0.6856 |     211306    |
> | 8B LLM | w/o Prefix | AIME-2024 |   1.11   |  366.74  | 302.76 | 0.7516 |     103473    |
> | 8B LLM | w/o Prefix |    GPQA   |   12.62  |  657.74  | 198.44 | 1.0510 |     94475     |
>
> **Table-17 8B LLM without prefix caching**
>
> ----
>
> We hope our detailed responses help clarify  concerns. Please feel free to request any additional information.
>
> ----
>
> ## &#x20;Reference
>
> \[1]  Yang A, Li A, Yang B, et al. Qwen3 technical report\[J]. arXiv preprint arXiv:2505.09388, 2025.
>
> \[2]  Guo D, Yang D, Zhang H, et al. Deepseek-r1 incentivizes reasoning in llms through reinforcement learning\[J]. Nature, 2025, 645(8081): 633-638.
>
> \[3]  Xu H, Peng B, Awadalla H, et al. Phi-4-mini-reasoning: Exploring the limits of small reasoning language models in math\[J]. arXiv preprint arXiv:2504.21233, 2025.
>
> \[4]  Team K, Bai Y, Bao Y, et al. Kimi k2: Open agentic intelligence\[J]. arXiv preprint arXiv:2507.20534, 2025.

---

### Author Response · Authors · 2025-12-01
**General Response and Summary**

Dear AC, Reviewers:

We thank the all the reviewers for their careful reading and constructive feedback.

We are glad to see  **all of reviewers**  agree that our paper studies an important and timely problem: how to efficiently serve reasoning language models. They highlight that our work is among the **first** systematic studies of this topic, with broad experiments across  various settings. They also find our observations, findings, and analysis are  **valuable both for researchers and system practitioners**.  In the rebuttal period, we have tried our best to address the reviewers' comments and concerns in individual responses to each reviewer. These reviews allowed us to improve our draft.

----

In the following part, we would like to provide  a summary of rebuttals to questions that most of reviewers concern.

> Q1:  Why do KV Cache quantization help large models but harm small models?

**Reply**: Experiments on 1.5B–8B models (Table 2-9 in the rebuttal for reviewer hqmG, d8ud, ewEa) show that small models are far more sensitive to quantization noise: they have less parameter redundancy, depend more on precise attention scores, and are worse at tolerating perturbations in multi-step chain-of-thought reasoning, so small errors in low-precision KV states can derail their reasoning. The degradation is also architecture-dependent: failures are concentrated in Qwen-2.5-Math 1.5B and 7B, while other small models like 3B and 4B remain relatively stable, suggesting that incomplete support for certain architectures in the serving engine (e.g., vLLM) contributes to the problem. Overall, small RLLMs are less robust than similarly sized non-reasoning LLMs, concluding that KV-cache quantization should be applied to small RLLMs only with great caution.

> Q2: Why do prefix cache (PC) help large models but harm small models?

**Reply** : PC introduces extra overhead for hashing and prefix matching, cache lookups, and metadata/memory management. At the same time, RLLM workloads are typically dominated by decoding, which is memory-bound. PC only helps with prefill, not decode, so it does nothing to relieve the main bottleneck while adding extra GPU-side memory and metadata operations. Besides, PC is more helpful in multi-turn dialogue or long context. In our evaluation settings, the outputs of RLLMs share less prefix, which leads to a low prefix hit rate, so the cost of maintaining the cache outweighs its benefits.

> Q3:  Insights and implications for RLLM serving system optimization.

**Reply** :  Based on RLLM-specific characteristics and the behaviors observed in our study, we believe these below techniques can be employed to optimize RLLM serving:
+ **Recompute vs. Swap–based preemption.**
Current engines typically rely on recompute-based preemption to avoid explicit memory-management overhead.   However, under RLLM workloads with much longer reasoning chains, re-computation becomes a larger portion of total latency.   This suggests that **selective KV-cache swapping** may be a promising alternative for RLLM-oriented inference.
+ **Length-aware scheduling (e.g., approximate SJF).**
Building on prior LLM serving work, we believe that leveraging early signals—such as logits from the first generated token—to predict the eventual reasoning length may be enable approximate Shortest-Job-First or related policies, potentially reducing overall turnaround time and alleviating queue congestion.
+ **Token-budget–based preemption and multi-level queues.**
Treating “generated token count” as an analogue to CPU time slices enables token-budget–based preemption.   Incorporating token counts as aging/degradation signals in multi-level feedback queues (MLFQ) could help maintain responsiveness for short queries while gently throttling extremely long reasoning chains.
+ **Long-running request isolation in multi-instance deployment.**
In multi-instance settings, isolating unusually long-running requests into dedicated instances or resource pools—using lightweight redirection—may reduce interference with short requests and improve overall quality of service.
We hope that these directions provide initial inspiration for future research efforts on efficient RLLM serving.

----

In this rebuttal, we focused on addressing the main concerns raised by the reviewers.  We hope these clarifications complement the strengths already noted by the reviewers.  We sincerely thank the AC and reviewers for their time and constructive feedback, and we respectfully ask the AC to take these strengths and our clarifications into account when making the final decision.


Best Regrads,

Author of Paper  7783

---

### Meta-Review · Area_Chair_KfDq · 2026-01-06

**Summary:**

This paper presents one of the first systematic empirical studies of reasoning large language model (RLLM) inference serving, highlighting how long CoT reasoning fundamentally alters serving behavior compared to standard/smaller LLMs. Through extensive experiments across model sizes, engines, datasets, and optimization techniques, the authors identify distinctive phenomena such as memory volatility, straggler requests, adaptive runtimes, and domain-specific advantages. The work further evaluates common serving optimizations and shows that techniques effective for large RLLMs may degrade performance or accuracy for smaller ones. Overall, the paper provides timely empirical evidence and concrete system-level insights relevant to both researchers and practitioners.

**Reviewer Concerns:**

Reviewer hqmG
The reviewer’s primary concerns centered on insufficient explanation for why prefix caching and KV-cache quantization benefit large RLLMs but harm smaller ones, as well as limited benchmark coverage. Minor presentation issues and typos were also noted, though the reviewer generally viewed the contributions as meaningful and timely.

Reviewer d8ud
This reviewer emphasized that several empirical observations—especially the negative effects of prefix caching on 7B RLLMs—were initially under-explained, making the results feel descriptive rather than analytical. They also noted the lack of direct LLM baselines in the optimization section and asked for clearer, more actionable system-design implications.

Reviewer soWx
The reviewer felt the paper was overly empirical and treated models largely as black boxes, with insufficient depth of analysis to explain several observed phenomena. They also questioned whether some evaluations overlapped with prior work, asked for clarification on unexpected accuracy variations (e.g., batch size and prefix caching), and requested explanations for why certain quantization methods worsened latency.

Reviewer ewEa
This reviewer argued that several findings, such as higher memory usage and long-tail latency, were expected consequences of chain-of-thought reasoning and would benefit from stronger analytical grounding. They also criticized the lack of proposed mitigation strategies or concrete solutions, particularly for straggler requests and optimization failures in small RLLMs, noting that the analysis stopped short of actionable system design.

**Reviewer Scores:**

Most reviewers questioned details of experiments and asked for more explanation. I view this as positive signals for the community to be interested in the presented observations. With more experiments presented in the rebuttal, I think most reviewers would lean towards a more positive outcome.

---

### Decision · Program_Chairs · 2026-01-26

Accept (Poster)